# Fast Solvers for Discrete Diffusion Models: Theory and Applications of High-Order Algorithms

**Yinuo Ren**[1,*]  **Haoxuan Chen**[1,*,†]  **Yuchen Zhu**[2,*]  **Wei Guo**[2,*]
**Yongxin Chen**[2]  **Grant M. Rotskoff**[1]  **Molei Tao**[2]  **Lexing Ying**[1]
[1]Stanford University  [2]Georgia Institute of Technology
{yinuoren, haoxuanc, rotskoff, lexing}@stanford.edu
{yzhu738, wei.guo, yongchen, mtao}@gatech.edu

## Abstract

Discrete diffusion models have emerged as a powerful generative modeling framework for discrete data with successful applications spanning from text generation to image synthesis. However, their deployment faces challenges due to the high dimensionality of the state space, necessitating the development of efficient inference algorithms. Current inference approaches mainly fall into two categories: exact simulation and approximate methods such as $\tau$-leaping. While exact methods suffer from unpredictable inference time and redundant function evaluations, $\tau$-leaping is limited by its first-order accuracy. In this work, we advance the latter category by tailoring the first extension of high-order numerical inference schemes to discrete diffusion models, enabling larger step sizes while reducing error. We rigorously analyze the proposed schemes and establish the second-order accuracy of the $\theta$-Trapezoidal method in KL divergence. Empirical evaluations on GSM8K-level math-reasoning, GPT-2-level text, and ImageNet-level image generation tasks demonstrate that our method achieves superior sample quality compared to existing approaches under equivalent computational constraints, with consistent performance gains across models ranging from 200M to 8B. Our code is available at https://github.com/yuchen-zhu-zyc/DiscreteFastSolver.

## 1 Introduction

Diffusion and flow-based models on discrete spaces [1–10] have emerged as a cornerstone of modern generative modeling for categorical data, offering unique advantages in domains where continuity assumptions fail. Unlike their continuous counterparts, discrete diffusion models inherently accommodate data with discrete structures, *e.g.*, language tokens, molecular sequences, tokenized images, and graphs, enabling principled generation and inference in combinatorially complex spaces. These models have exerted a large impact on numerous applications, from the design of molecules [11], proteins [12], and DNA sequences [13, 14] under biophysical constraints, to the generation of high-fidelity text [15] and images [16] via autoregressive or masked transitions, *etc.*. Beyond standalone tasks, discrete diffusion models also synergize with methodologies, ranging from tensor networks [17] to guidance mechanisms [18–20].

Discrete diffusion models, despite their broad applicability, face a critical bottleneck: *inference inefficiency*. Current inference methods include: (1) exact simulation methods [21], which ensure unbiased sampling from the pre-trained model but suffer from unpredictable inference time and redundant score evaluations, resulting in poor scaling w.r.t. dimensionality; and (2) approximate

---

[*]Equal contribution

[†] Corresponding author

methods such as $\tau$-leaping [22], which offer simple and parallelizable implementation but, due to their first-order accuracy, requires small step sizes to control discretization error, forcing a stringent trade-off between speed and sample quality.

To address these limitations in possibly computationally constrained environments, we develop high-order numerical schemes tailored for discrete diffusion model inference. Drawing inspirations from acceleration techniques developed for ordinary differential equations (ODEs) [23], stochastic differential equations (SDEs) [24, 25], chemical reaction simulations [26], and most recently continuous diffusion [27–29], our work represents the *first successful adaptation of high-order numerical schemes to the discrete diffusion domain*. Through careful design, these high-order schemes provide unprecedented efficient and versatile solutions for discrete diffusion model inference.

**Our Contributions.** The main contributions of this paper are summarized as follows:

- We introduce the *first high-order numerical solvers* for discrete diffusion model inference, namely the $\theta$-Runge-Kutta-2 ($\theta$-RK-2) method and the $\theta$-Trapezoidal method;

- We rigorously establish the theoretical properties of both methods, proving *second-order convergence* of the $\theta$-Trapezoidal method and *conditional second-order convergence* of the $\theta$-RK-2 method;

- We empirically validate our theoretical results and demonstrate the *superior performance* of the $\theta$-Trapezoidal method through comprehensive evaluations on large-scale text and image generation benchmarks.

## 1.1 Related Works

Here we briefly review related works and defer a more detailed discussion to App. A.

**Discrete Diffusion Models.** Since their introduction, discrete diffusion models have undergone significant refinements, including the development of score-entropy loss [30] and flow-matching formulation [31, 32]. These models generally fall into two categories based on their noise distribution: uniform [30, 20] and masked (absorbing state) [33–35, 21], each offering unique advantages in modeling discrete distributions. Recent theoretical advances have emerged through studies [36–38].

**High-Order Scheme for Continuous Diffusion Models.** The development of high-order numerical schemes for solving ODEs and SDEs represents decades of research, as comprehensively reviewed in [23, 39, 40]. These schemes have recently been adapted to accelerate continuous diffusion model inference, encompassing approaches such as the exponential integrators [41–43], Adams-Bashforth methods [29, 44, 45], Taylor methods [27, 46] and (stochastic) Runge-Kutta methods [47, 28, 48–51].

**High-Order Scheme for Chemical Reaction Systems.** Regarding approximate methods for simulating compound Poisson processes and chemical reaction systems with state-dependent intensities, efforts have been made on the $\tau$-leaping method [52], and its extensions [53, 54, 26, 55, 56]. For a quick review of the problem setting and these methods, one may refer to [57, 58]. The adaptation of these methods to discrete diffusion models presents unique challenges due to the presence of both time and state-inhomogeneous intensities in the underlying Poisson processes.

## 2 Preliminaries

In this subsection, we review several basic concepts and previous error analysis results of discrete diffusion models.

## 2.1 Discrete Diffusion Models

In discrete diffusion models, one considers a continuous-time Markov chain (CTMC) $(\boldsymbol{x}_t)_{0 \le t \le T}$ on a finite space $\mathbb{X}$ as the *forward process*. We represent the distribution of $\boldsymbol{x}_t$ by a vector $\boldsymbol{p}_t \in \Delta^{|\mathbb{X}|}$, where $\Delta^{|\mathbb{X}|}$ denotes the probability simplex in $\mathbb{R}^{|\mathbb{X}|}$. Given a target distribution $\boldsymbol{p}_0$, the CTMC

satisfies the following equation:

$$\frac{d\boldsymbol{p}_t}{dt} = \boldsymbol{Q}_t \boldsymbol{p}_t, \quad \text{where } \boldsymbol{Q}_t = (Q_t(y,x))_{x,y \in \mathbb{X}} \tag{2.1}$$

is the rate matrix at time $t$ satisfying

$$\text{(i) } Q_t(x,x) = -\sum_{y \neq x} Q_t(y,x), \ \forall x \in \mathbb{X}; \ \text{(ii) } Q_t(x,y) \geq 0, \ \forall x \neq y \in \mathbb{X}.$$

Below we will use the notation $\boldsymbol{Q}_t^0 = \boldsymbol{Q}_t - \text{diag}\,\boldsymbol{Q}_t$. It can be shown that the corresponding backward process is of the same form but with a different rate matrix [59]:

$$\frac{d\overleftarrow{\boldsymbol{p}}_s}{ds} = \overline{\boldsymbol{Q}}_s \overleftarrow{\boldsymbol{p}}_s, \quad \text{where } \overline{Q}_s(y,x) = \begin{cases} \frac{\overleftarrow{p}_s(y)}{\overleftarrow{p}_s(x)} \overleftarrow{Q}_s(x,y), & \forall x \neq y \in \mathbb{X}, \\ -\sum_{y' \neq x} \overline{Q}_s(y',x), & \forall x = y \in \mathbb{X}. \end{cases} \tag{2.2}$$

is the rate matrix and $\overleftarrow{*}_s$ denotes $*_{T-s}$. The rate matrix $\boldsymbol{Q}_t$ is often chosen to possess certain sparse structures such that the forward process converges to a simple distribution that is easy to sample from. Popular choices include the uniform and absorbing state cases [30], where the forward process (2.1) converges to the uniform distribution on $\mathbb{X}$ and a Dirac distribution, respectively.

Common training practice is to define the score function (or the score vector) as $\boldsymbol{s}_t(x) = (s_t(x,y))_{y \in \mathbb{X}} := \frac{\boldsymbol{p}_t}{p_t(x)}$ for any $x \in \mathbb{X}$, $t \in [0,T]$ and estimate it by a neural network $\widehat{\boldsymbol{s}}_t^\phi(x)$, where the parameters $\phi$ are trained by minimizing the score entropy [30, 60] for some weights $\psi_t \geq 0$ as:

$$\min_\phi \int_0^T \psi_t \mathbb{E}_{x_t \sim p_t} \left[ \sum_{y \neq x_t} Q_t(x_t, y) \left( s_t(x_t, y) \log \frac{s_t(x_t,y)}{\widehat{s}_t^\phi(x_t,y)} - s_t(x_t,y) + \widehat{s}_t^\phi(x_t,y) \right) \right] dt. \tag{2.3}$$

Similar to the continuous case, the backward process is approximated by another CTMC $\frac{d\boldsymbol{q}_s}{ds} = \widehat{\overline{\boldsymbol{Q}}}_s^\phi \boldsymbol{q}_s$, with $\boldsymbol{q}_0 = \boldsymbol{p}_\infty$ and rate matrix $\widehat{\overline{\boldsymbol{Q}}}_s^\phi$, where $\widehat{\overline{Q}}_s^\phi(y,x) = \overleftarrow{\widehat{s}}_s^\phi(x,y) \overleftarrow{Q}_s(x,y)$ for any $x \neq y \in \mathbb{X}$. The inference is done by first sampling from $\boldsymbol{p}_\infty$ and then evolving the CTMC accordingly. For simplicity, we drop the superscript $\phi$ hereafter.

## 2.2 Stochastic Integral Formulation of Discrete Diffusion Models

Discrete diffusion models can also be formulated as stochastic integrals, which is especially useful for their theoretical analysis [38]. In this section, we briefly recapitulate the relevant results and refer to App. B for the mathematical details. Below, we work on the probability space $(\Omega, \mathcal{B}, \mathbb{P})$ and denote the pairwise difference set of the state space $\mathbb{X}$ by $\mathbb{D} := \{x - y : x \neq y \in \mathbb{X}\}$. In this work, we focus on the case where $\mathbb{X} = [S]^d$ with $d$ data dimensions and $S$ sites along each dimension.

We first introduce the Poisson random measure, a key concept in the formulation.

**Definition 2.1** (Informal Definition of Poisson Random Measure). *The random measure $N[\lambda](dt, d\nu)$ on $\mathbb{R}^+ \times \mathbb{D}$ is called a* Poisson random measure *with* evolving intensity $\lambda$ *w.r.t. a measure $\gamma$ on $\mathbb{D}$ if, roughly speaking, the number of jumps of magnitude $\nu$ during the infinitesimal time interval $(t, t+dt]$ is Poisson distributed with mean $\lambda_t(\nu)\gamma(d\nu)dt$.*

The forward process (2.1) can thus be represented by the following stochastic integral:

$$x_t = x_0 + \int_0^t \int_\mathbb{D} \nu N[\lambda](ds, d\nu),$$

where the intensity $\lambda$ is defined as $\lambda_t(\nu, \omega) = Q_t^0(x_{t^-}(\omega) + \nu, x_{t^-}(\omega))$ if $x_{t^-}(\omega) + \nu \in \mathbb{X}$ and 0 otherwise. Here, the outcome $\omega \in \Omega$ and $x_{t^-}$ denotes the left limit of the càdlàg process $x_t$ at time $t$ with $x_{0^-} = x_0$. We will also omit the variable $\omega$, should it be clear from context. The backward process in discrete diffusion models (2.2) can also be represented similarly as:

$$y_s = y_0 + \int_0^s \int_\mathbb{D} \nu N[\mu](ds, d\nu), \tag{2.4}$$

where the intensity $\mu$ is defined as $\mu_s(\nu, \omega) = \overleftarrow{s}_s(y_{s^-}, y_{s^-} + \nu) \overleftarrow{Q}_s^0(y_{s^-}, y_{s^-} + \nu)$ if $y_{s^-} + \nu \in \mathbb{X}$ and 0 otherwise. During inference, $\widehat{y}_s = \widehat{y}_0 + \int_0^s \int_\mathbb{D} \nu N[\widehat{\mu}](ds, d\nu)$ is used instead of (2.4), where the estimated intensity $\widehat{\mu}$ is defined by replacing the true score $\boldsymbol{s}_t$ with the neural network estimated score $\widehat{\boldsymbol{s}}_t$ in $\mu_s(\nu, \omega)$. . In the following, we also denote the intensity $\mu_s(\nu, \omega)$ at time $s$ by $\mu_s(\nu, y_{s^-})$ with slight abuse of terminology to emphasize its dependency on $\omega$ through $y_{s^-}(\omega)$.

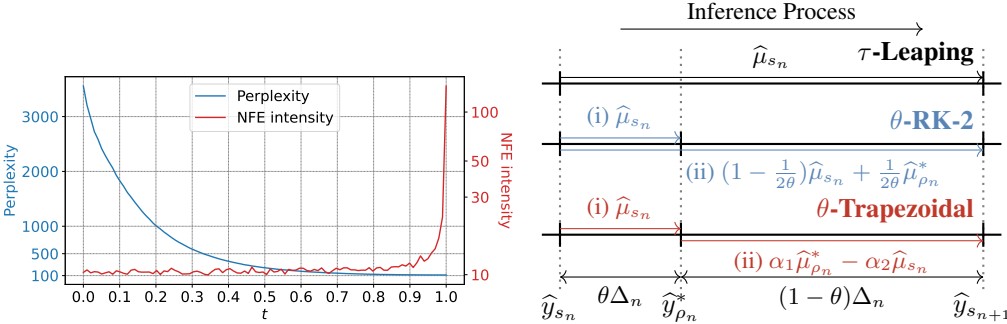

Figure 1: **Left:** Application of the uniformization algorithm to discrete diffusion models for text generation. The $x$-axis denotes the time of the backward process, and the $y$-axis denotes the frequency of jumps (NFE). Perplexity convergence occurs before the NFE grows unbounded. **Right:** Comparison between $\tau$-leaping and the proposed second-order schemes ($\theta$-RK-2 and $\theta$-Trapezoidal).

## 3 Numerical Schemes for Discrete Diffusion Model Inference

Before introducing the proposed numerical schemes, we first review existing numerical schemes for discrete diffusion models, including exact simulation methods and the $\tau$-leaping method, and discuss their merits and limitations.

### 3.1 Exact Simulation Methods

Unlike in continuous diffusion models, where exact simulation is infeasible, discrete diffusion models permit inference without discretization error. Notable examples of unbiased samplers include uniformization [36] for the uniform state case and the First-Hitting Sampler (FHS) [21] for the absorbing state case. The main idea behind these methods is to first sample the next jump time and then the jump itself. Theoretical analysis [38] reveals that such schemes *lack guarantees with finite computation budget*, since the number of required jumps (and thus the inference time) follows a random distribution with expectation $\Omega(d)$. This computational restriction may be less favorable for high-dimensional applications, such as generative modeling of DNA or protein sequences.

Furthermore, *the absence of discretization error does not necessarily translate to superior sample quality*, given the inherent estimation errors in neural network-based score functions. This limitation is further amplified by the *highly skewed distribution* of jumps, with a concentration occurring during the terminal phase of the backward process, when the neural network-based score function exhibits the highest estimation error. This phenomenon stems from the potential singularity of the target distribution $\boldsymbol{p}_0$, which induces singularities in the score function, making accurate neural network estimation particularly challenging during that phase (*cf.* Assump. 4.4 [38]).

The left figure in Fig. 1 illustrates an application of the uniformization algorithm to discrete diffusion inference for text generation, with detailed experimental parameters presented in Sec. 6.3 and App. D.3. As the process approaches the target distribution ($t \to T$), the number of jumps (in terms of the number of score function evaluations, NFE) grows unbounded, while perplexity improvements become negligible. This skew in computational effort leads to *redundant function evaluations*. Although early stopping is commonly adopted at $T - \delta$ for some small $\delta \ll 1$ to alleviate this inefficiency, this approach introduces challenges in its selection, particularly under computational constraints or when efficiency-accuracy trade-offs are desired. Moreover, the variable jump schedules across batch samples complicate parallelization efforts in exact methods, highlighting the need for more adaptable and efficient algorithmic solutions.

### 3.2 Approximate Method: $\tau$-Leaping Method

The $\tau$-leaping method [52, 22] is a widely adopted scheme that effectively addresses both dimensionality scaling and inference-time control challenges. This Euler-type scheme approximates the backward process with time-dependent intensity $\widehat{\mu}_t$ via the following updates:

$$\widehat{y}_{t+\Delta} = \widehat{y}_t + \sum_{\nu \in \mathbb{D}} \nu \mathcal{P}\left(\widehat{\mu}_t(\nu)\Delta\right), \tag{3.1}$$

where $\Delta$ denotes the time step and $\mathcal{P}(\cdot)$ denotes a Poisson random variable. In general, one may design different discretization schemes for $\tau$-leaping, and the summation in (3.1) is parallelizable, underscoring the method's flexibility and efficiency. We refer to Alg. 3 and App. B.2 for a detailed description of the $\tau$-leaping method for discrete diffusion model inference. Regarding convergence properties as the time discretization becomes increasingly refined, theoretical analyses by [22, 38] have established the error bounds of the $\tau$-leaping method, the results of which are summarized in the following theorem. Further discussion can be found in App. B.2.

**Theorem 3.1** (Thm. 4.7 in [38]). *Under a certain discretization scheme and technical assumptions, and given an $\epsilon$-accurate score function, the following error bound holds:*

$$D_{\mathrm{KL}}(p_\delta \| \widehat{q}_{T-\delta}) \lesssim \exp(-T) + \epsilon + \kappa T, \tag{3.2}$$

*where $\delta \ll 1$ is the early stopping time, $\kappa$ controls the step size, and $T$ is the time horizon. The notation $\lesssim$ indicates the inequality holds up to a constant factor as $\kappa \to 0$.*

The error bound (3.2) decouples three error sources of the $\tau$-leaping scheme: the truncation error $\mathcal{O}(e^{-T})$, the score estimation error $\epsilon$, and the discretization error $\mathcal{O}(\kappa T)$. Similar to the case for the Euler method for ODEs and the Euler-Maruyama scheme for SDEs, the $\tau$-leaping method is a first-order scheme in terms of the discretization error $\mathcal{O}(\kappa T)$.

## 4 Algorithms: High-Order Inference Schemes

A natural improvement of $\tau$-leaping is to develop high-order schemes for discrete diffusion models. As a foundational example, consider the second-order Runge-Kutta (RK-2) method with two stages [23] for solving the ODE $\mathrm{d}x_t = f_t(x_t)\mathrm{d}t$. This method represents one of the simplest high-order numerical schemes:

$$\widehat{x}^*_{t+\theta\Delta} = \widehat{x}_t + f_t(\widehat{x}_t)\theta\Delta, \quad \widehat{x}_{t+\Delta} = \widehat{x}_t + \left[(1-\tfrac{1}{2\theta})f_t(\widehat{x}_t) + \tfrac{1}{2\theta}f_{t+\theta\Delta}(\widehat{x}^*_{t+\theta\Delta})\right]\Delta. \tag{4.1}$$

This scheme reduces to the exact midpoint method when $\theta = \frac{1}{2}$ and Heun's method when $\theta = 1$. The underlying intuition stems from the observation that for $f \in C^2(\mathbb{R})$, $\left[\left(1 - \tfrac{1}{2\theta}\right) f(0) + \tfrac{1}{2\theta}f(\theta\Delta)\right]\Delta$ offers a second-order approximation of $\int_0^\Delta f(x)\mathrm{d}x$ in contrast to $f(0)\Delta$, which is only first-order. This approach has been successfully adapted for SDE simulation [24] and continuous diffusion model inference [48, 28, 29, 49, 51]. Notably, these methods enhance sample quality and computational efficiency without requiring additional model training, making the development of high-order schemes for discrete diffusion inference both theoretically appealing and practically viable.

In this section, we propose two high-order solvers for inference in the discrete diffusion model. We will primarily focus on two-stage algorithms aiming for second-order accuracy. Specifically, we will introduce the $\theta$-RK-2 and $\theta$-Trapezoidal methods. Throughout this section, we assume a time discretization scheme $(s_i)_{i\in[0:N]}$ with $0 = s_0 < \cdots < s_N = T - \delta$, where $\delta$ is the early stopping time and use the shorthand notations $*_+ = \max\{0, *\}$. For any $s \in (s_n, s_{n+1}]$ and $n \in [0 : N-1]$, we define $\lfloor s \rfloor = s_n$, $\rho_s = (1-\theta)s_n + \theta s_{n+1}$, $\Delta_n = s_{n+1} - s_n$, and $\theta$-*section points* as $\rho_n = (1-\theta)s_n + \theta s_{n+1}$. We choose $\gamma(\mathrm{d}\nu)$ to be the counting measure on $\mathbb{D}$.

### 4.1 $\theta$-RK-2 Method

We first present the $\theta$-*RK-2 method*, which is simple in design and serves as a natural analog of the second-order RK method for ODEs (4.1) in terms of time and state-dependent Poisson random measures, as a warm-up for the $\theta$-Trapezoidal method. We note that similar methods have been proposed for simulating SDEs driven by

---

**Algorithm 1:** $\theta$-RK-2 Method

**Input:** $\widehat{y}_0 \sim q_0$, $\theta \in (0,1]$, $(s_n, \rho_n)_{n\in[0:N-1]}$, $\widehat{\mu}$, $\widehat{\mu}^*$.
**Output:** A sample $\widehat{y}_{s_N} \sim \widehat{q}^{\mathrm{RK}}_{t_N}$.

  **1 for** $n = 0$ **to** $N-1$ **do**
    **2**   $\widehat{y}^*_{\rho_n} \leftarrow \widehat{y}_{s_n} + \sum_{\nu\in\mathbb{D}} \nu\mathcal{P}\left(\widehat{\mu}_{s_n}(\nu)\theta\Delta_n\right);$
    **3**   $\widehat{y}_{s_{n+1}} \leftarrow \widehat{y}_{s_n} +$
        $\sum_{\nu\in\mathbb{D}} \nu\mathcal{P}\left(\mathbf{1}_{\widehat{\mu}_{s_n}>0}\left[\left(1-\tfrac{1}{2\theta}\right)\widehat{\mu}_{s_n} + \tfrac{1}{2\theta}\widehat{\mu}^*_{\rho_n}\right]_+(\nu)\Delta_n\right);$
  **4 end**

---

Brownian motions or Poisson processes, such as the stochastic [24] and the Poisson [54] RK methods. A summary of this method is given in Alg. 1.

Intuitively, the $\theta$-RK-2 method is a two-stage algorithm that:

(i) Firstly, it runs $\tau$-leaping with step size $\theta\Delta_n$, obtains an *intermediate state* $\widehat{y}^*_{\rho_n}$ at the $\theta$-section point $\rho_n$, and evaluates the intensity $\widehat{\mu}^*_{\rho_n}$ there;

(ii) Then another step of $\tau$-leaping for a full step $\Delta_n$ is run using a weighted sum of the intensities at the current time point $s_n$ and the $\theta$-section point $\rho_n$.

We emphasize that our method differs from the midpoint method proposed in [52] for simulating chemical reactions, in which the Poisson random variable in the first step is replaced by its expected value. Such modification is in light of the lack of continuity and orderliness of the state space.

### 4.2 $\theta$-Trapezoidal Method

As to be shown theoretically and empirically, the conceptually simple $\theta$-RK-2 method may have limitations in terms of both accuracy and efficiency. To this end, we propose the following $\theta$-*Trapezoidal method*, which is developed based on existing methods proposed for simulating SDEs [25] and chemical reactions [26]. Below, we introduce two parameters that will be used extensively later:

$$\alpha_1 = \tfrac{1}{2\theta(1-\theta)} \text{ and } \alpha_2 = \tfrac{(1-\theta)^2+\theta^2}{2\theta(1-\theta)}, \text{ with } \alpha_1 - \alpha_2 = 1.$$

The $\theta$-Trapezoidal method is summarized in Alg. 2. Intuitively, it separates each interval $(s_n, s_{n+1}]$ into two sub-intervals $(s_n, \rho_n]$ and $(\rho_n, s_{n+1}]$, on which simulations are detached with different intensities designed in a balanced way.

Compared to the $\theta$-RK-2 method, the $\theta$-Trapezoidal method is also two-stage with an identical first step. The second step, however, differs in two major aspects:

---

**Algorithm 2: $\theta$-Trapezoidal Method**

**Input:** $\widehat{y}_0 \sim q_0$, $\theta \in (0,1]$, $(s_n, \rho_n)_{n \in [0:N-1]}$, $\widehat{\mu}, \widehat{\mu}^*$.
**Output:** A sample $\widehat{y}_{s_N} \sim \widehat{q}^{\text{trap}}_{t_N}$.

1 **for** $n = 0$ **to** $N - 1$ **do**
2 $\quad \widehat{y}^*_{\rho_n} \leftarrow \widehat{y}_{s_n} + \sum_{\nu \in \mathbb{D}} \nu \mathcal{P}\left(\widehat{\mu}_{s_n}(\nu)\theta\Delta_n\right);$
3 $\quad \widehat{y}_{s_{n+1}} \leftarrow \widehat{y}^*_{\rho_n} +$
$\quad\quad \sum_{\nu \in \mathbb{D}} \nu \mathcal{P}\left(\left(\alpha_1\widehat{\mu}^*_{\rho_n} - \alpha_2\widehat{\mu}_{s_n}\right)_+(\nu)(1-\theta)\Delta_n\right);$
4 **end**

---

(1) The second step starts from the intermediate state $\widehat{y}^*_{\rho_n}$ instead of $\widehat{y}_{s_n}$ and only runs for a fractional step $(1-\theta)\Delta_n$ rather than a full step $\Delta_n$;

(2) The weighted sum is comprised of an altered pair of coefficients $(\alpha_1, -\alpha_2)$, performing an *extrapolation* instead of interpolation with coefficients $(1 - \tfrac{1}{2\theta}, \tfrac{1}{2\theta})$ as in the $\theta$-RK-2 method with $\theta \in [\tfrac{1}{2}, 1]$. This feature will be shown to render the algorithm unconditionally second-order.

Following the common practice in the literature [22], we reject updates with multiple jumps along one dimension in both algorithms, ensuring their well-posedness. A simple analysis shows that rejection only happens with probability $\mathcal{O}(\kappa)$, and we refer to further details in Rmk. C.4. We refer to Props. C.2 and C.3 for the stochastic integral formulations of these two algorithms. We provide a visual comparison between the $\theta$-RK-2 and the $\theta$-Trapezoidal method in the right figure of Fig. 1.

## 5 Theoretical Analysis

In this section, we provide the theoretical results of the $\theta$-Trapezoidal and $\theta$-RK-2 methods. The goal of this section is to show that under certain conditions, both methods are second-order accurate, improving from the first-order accuracy of the $\tau$-leaping method (*cf.* Thm. 3.1). Our theoretical analysis also reveals that the $\theta$-Trapezoidal method is more robust to the choice of $\theta$ than $\theta$-RK-2, to be confirmed by our empirical results in Sec. 6.

### 5.1 Assumptions

For simplicity, we impose a periodic boundary condition on the state space $\mathbb{X} = [S]^d$, *i.e.*, embed the state space in the $d$-dimensional torus $\mathbb{T}^d$, to streamline the proofs (*cf.* Rmk. C.4).

**Assumption 5.1** (Convergence of Forward Process). *The forward process converges to the stationary distribution exponentially fast,* i.e., $D_{\text{KL}}(p_T \| p_\infty) \lesssim \exp(-T)$.

This assumption ensures rapid convergence of the forward process, controlling error when terminated at a sufficiently large time horizon $T$, and is automatically satisfied in the masked state case

and the uniform state case, given sufficient connectivity of the graph (*cf.* [38]). The exponential rate aligns with continuous diffusion models (*cf.* [61]).

**Assumption 5.2** (Regularity of Intensity). *For the true intensity $\mu_s(\nu, y_{s-})$ and the estimated intensity $\widehat{\mu}_s(\nu, y_{s-})$, it holds almost everywhere w.r.t. $\mu_s(\nu, y_{s-})\gamma(\mathrm{d}\nu)\breve{p}_{s-}(\mathrm{d}y_{s-})$ that: (1) Both intensities belong to $C^2([0, T - \delta])$; (2) Both intensities are upper and lower bounded on $[0, T - \delta]$.*

This assumes two key requirements of the scores: (1) the forward process maintains sufficient smoothness, which is achievable through appropriate time reparametrization; and (2) if and only if a state $y \in \mathbb{X}$ is achievable by the forward process and $\nu$ is a permissible jump therefrom, then both its true and estimated intensity are bounded, corresponding to Assumps. 4.3(i), 4.4, and 4.5 [38].

**Assumption 5.3** (Estimation Error). *For all grid points and $\theta$-section points, the estimation error of the neural network-based score is small, i.e., for any $s \in \cup_{n \in [0:N-1]}\{s_n, \rho_n\}$, we have*

*(1)* $\mathbb{E}\left[\int_{\mathbb{D}}\left(\mu_s(\nu)\left(\log\frac{\mu_s(\nu)}{\widehat{\mu}_s(\nu)} - 1\right) + \widehat{\mu}_s(\nu)\right)\gamma(\mathrm{d}\nu)\right] \leq \epsilon_{\mathrm{I}};$

*(2)* $\mathbb{E}\left[\int_{\mathbb{D}}|\mu_s(\nu) - \widehat{\mu}_s(\nu)|\,\gamma(\mathrm{d}\nu)\right] \leq \epsilon_{\mathrm{II}}.$

This assumption quantifies the proximity of the estimated intensity $\widehat{\mu}$ to the true intensity $\mu$ after sufficient training. Compared with [38], the additional $L^\infty$ part in (2) is required for technical reasons, which is similar to [62, 51]. In practice, such additional assumptions may be realized by adding extra penalty terms to the objective function during training.

## 5.2 Convergence Guarantees

The following theorem summarizes our theoretical guarantees for the $\theta$-Trapezoidal method:

**Theorem 5.4** (Second Order Convergence of $\theta$-Trapezoidal Method). *Suppose $\theta \in (0, 1]$ and $\alpha_1\widehat{\mu}^*_{\rho_s} - \alpha_2\widehat{\mu}_{\lfloor s \rfloor} \geq 0$ for all $s \in [0, T - \delta]$, then the following error bound holds for Alg. 2 under Assumps. 5.1 to 5.3:*

$$D_{\mathrm{KL}}(p_\delta \| \widehat{q}^{\mathrm{trap}}_{T-\delta}) \lesssim \exp(-T) + (\epsilon_{\mathrm{I}} + \epsilon_{\mathrm{II}})T + \kappa^2 T,$$

*where $\delta$ is the early stopping time, $\kappa = \max_{n \in [0:N-1]} \Delta_n$, i.e., the largest stepsize, and $\widehat{q}^{\mathrm{trap}}_{T-\delta}$ is the distribution obtained by Alg. 1 as defined in Prop. C.2.*

The complete proof is presented in App. C.2. The outline is to first bound $D_{\mathrm{KL}}(p_\delta \| \widehat{q}^{\mathrm{trap}}_{T-\delta})$ by the KL divergence between the corresponding path measures, as established in Thm. C.5, and then decompose the integral in the log-likelihood and bound respectively, where the primary technique used is *Dynkin's formula* (Thm. C.10). With a term-by-term comparison with Thm. 3.1, we observe a significant improvement in the discretization error term from $\mathcal{O}(\kappa T)$ to $\mathcal{O}(\kappa^2 T)$. This confirms that the $\theta$-Trapezoidal method achieves second-order accuracy given a sufficient time horizon $T$ and accurate score estimation, with empirical validation presented in Sec. 6.

**Theorem 5.5** (Conditional Second-Order Convergence of $\theta$-RK-2 Method). *Suppose $\theta \in (0, \frac{1}{2}]$ and $(1 - \frac{1}{2\theta})\widehat{\mu}_{\lfloor s \rfloor} + \frac{1}{2\theta}\widehat{\mu}^*_{\rho_s} \geq 0$ for all $s \in [0, T - \delta]$, then the following error bound holds for Alg. 1 under Assumps. 5.1 to 5.3:*

$$D_{\mathrm{KL}}(p_\delta \| \widehat{q}^{\mathrm{RK}}_{T-\delta}) \lesssim \exp(-T) + (\epsilon_{\mathrm{I}} + \epsilon_{\mathrm{II}})T + \kappa^2 T,$$

*where $\delta$ is the early stopping time, $\kappa = \max_{n \in [0:N-1]} \Delta_n$, i.e., the largest stepsize, and $\widehat{q}^{\mathrm{RK}}_{T-\delta}$ is the distribution obtained by Alg. 2 as defined in Prop. C.3.*

The proof of the theorem above is provided in App. C.3. The restricted range of $\theta$ is caused by one specific error term (III.4) (C.9) that permits bounding with *Jensen's inequality* only when $\theta \in (0, \frac{1}{2}]$, similar to its counterpart (II.4) (C.11) in the $\theta$-Trapezoidal method. The limitation arises partially because the weighted sum with coefficients $(1 - \frac{1}{2\theta}, \frac{1}{2\theta})$ becomes an *extrapolation* only if $1 - \frac{1}{2\theta} < 0$, a feature that naturally holds for all $\theta \in (0, 1]$ in the $\theta$-Trapezoidal method. These theoretical findings are consistent with the empirical observations in Fig. 6 of App. D.3, where the performance of $\theta$-RK-2 method clearly peaks when $\theta \in (0, \frac{1}{2}]$.

**Remark 5.6** (Comparison between Trapezoidal and RK-2 Methods). *Trapezoidal methods were originally proposed by [25] as a minimal second-order scheme in the weak sense for simulating SDEs. In simulating chemical reaction contexts, [26] claimed that trapezoidal methods also achieve second-order convergence for* covariance error *apart from the weak error, a property not shared by midpoint (RK-2) methods. Our empirical results partly reflect these findings, while we defer theoretical investigation of covariance error convergence in discrete diffusion models to future work.*

**Remark 5.7** (Remark on the Positivity of Extrapolated Intensity). *Due to the nature of extrapolation, both our theorems require an additional assumption on the positivity of the extrapolated intensity, which is classically assumed in [25, 26], and resolving this issue is a long-standing open problem. The best result so far is Prop. 5 [26], claiming clamping the intensity above 0 only causes an error of order $\mathcal{O}(\kappa^p)$, for any large integer $p$. We empirically demonstrate the validity of this assumption Tab. 6 in practice through the text generation task (Sec. 6.2) and find that positivity occurs for both methods with high probability over 95%, approaching 100% with increasing NFE. We refer to further discussion in Rmk. C.6.*

# 6  Experiments

Based on the theoretical analysis, we expect the $\theta$-Trapezoidal method to outperform the $\tau$-leaping method and the $\theta$-RK-2 method in terms of sample quality, given the same number of function evaluations. This section empirically validates the anticipated effectiveness of our proposed $\theta$-Trapezoidal method (Alg. 2) through comprehensive evaluations across text and image generation tasks. Our comparative analysis includes established discrete diffusion samplers as baselines, *e.g.*, the Euler method [33], $\tau$-leaping [22], Tweedie $\tau$-leaping [30], First-Hitting Sampler (FHS) [21], Parallel Decoding [63], and Semi-Autoregressive (Semi-AR) sampler [64]. We benchmark on both uniform and masked discrete diffusion models, with experiment details provided in App. D.

## 6.1  15-State Toy Model

We first evaluate the performance of the $\theta$-Trapezoidal method using a 15-state toy model ($d = 1$, $S = 15$). The target distribution is uniformly generated from $\Delta^{15}$, with rate matrix $\boldsymbol{Q} = \frac{1}{15}\boldsymbol{E} - \boldsymbol{I}$, where $\boldsymbol{E}$ is the all-one and $\boldsymbol{I}$ is the identity matrix. This setup provides analytically available score functions, allowing isolation and quantification of numerical errors introduced by inference algorithms. We apply both the $\theta$-Trapezoidal and the $\theta$-RK-2 method to generate $10^6$ samples and estimate the KL divergence between the true ground truth $\boldsymbol{p}_0$ and the generated distribution $\widehat{\boldsymbol{q}}_T$.

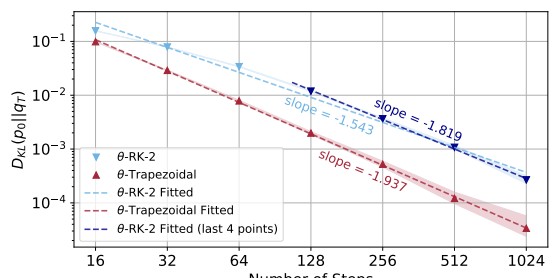

Figure 2: Empirical KL divergence between the true and generated distribution of the toy model vs. number of steps. Data are fitted with linear regression with 95% confidence interval by bootstrapping.

For a fair comparison, we choose $\theta = \frac{1}{2}$ for both methods, and the results are presented in Fig. 2. While both methods exhibit super-linear convergence as the total number of steps grows, the $\theta$-Trapezoidal method outperforms the $\theta$-RK-2 method in terms of both absolute value and convergence rate, while the $\theta$-RK-2 method takes longer to enter the asymptotic regime. Moreover, the fitted line indicates that the $\theta$-Trapezoidal method approximately converges quadratically with respect to the step count, confirming our theoretical results.

## 6.2  Text Generation

For the text generation task, we employ the pre-trained score function from RADD [33] as our base model for benchmarking inference algorithms. RADD is a masked discrete diffusion model with GPT-2-level text generation capabilities [65] and is trained on the OpenWebText dataset [66] with $d = 1024$ and $S = 50258$. Our comparative analysis maintains consistent computational resources across methods, quantified through the number of score function evaluations (NFE), and evaluates the sample quality produced by FHS, the Euler method, $\tau$-leaping, Tweedie $\tau$-leaping, Semi-AR,

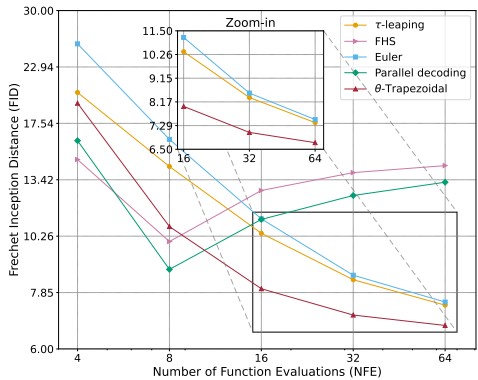

Figure 3: FID of images generated by different sampling algorithms vs. number of function evaluations (NFE). Lower values are better.

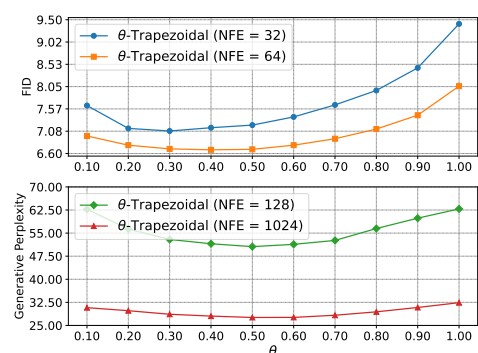

Figure 4: Sampling quality vs. $\theta \in (0, 1]$ in $\theta$-Trapezoid method. **Upper**: Image generation (FID). **Lower**: Text generation (perplexity). Lower is better.

and our proposed $\theta$-Trapezoidal method. We generate text sequences of 1024 tokens and measure their generative perplexity (computed with GPT-2 Large [65]) following the evaluation protocol established in [33].

Tab. 1 presents the results for both low (128) and high (1024) NFEs, with comprehensive results across additional NFE values in Tab. 3. The empirical results demonstrate that the $\theta$-Trapezoidal method consistently produces better samples within a fixed computation budget than existing popular inference algorithms. Notably, it outperforms Euler and Tweedie $\tau$-leaping, two of the best-performing samplers adopted by RADD, by a large margin. It also consistently prevails over FHS, which performs exact simulation at high NFE (1024), supporting again our observations that being free of

Table 1: Generative perplexity (on GPT-2 large) of texts generated by different sampling algorithms. Lower values are better, with the best in **bold**.

| Method | NFE = 128 | NFE = 1024 |
|---|---|---|
| FHS | $\leq 122.732$ | $\leq 109.406$ |
| Euler | $\leq 86.276$ | $\leq 44.686$ |
| Tweedie $\tau$-leap. | $\leq 85.738$ | $\leq 44.257$ |
| $\tau$-leaping | $\leq 52.366$ | $\leq 28.797$ |
| Semi-AR | $\leq 360.793$ | $\leq 147.406$ |
| $\theta$-RK-2 | $\leq 64.317$ | $\leq 36.330$ |
| $\theta$-Trapezoidal | $\leq \mathbf{49.051}$ | $\leq \mathbf{27.553}$ |

discretization error does not necessarily imply better sampling quality. These results validate the practical efficiency and accuracy of Alg. 2. Additional evaluation results, including unigram entropy and generative perplexities evaluated under LLaMA 3 [67], are detailed in App. D.2.

## 6.3 Image Generation

Our experiments on image generation utilize the pre-trained score function from MaskGIT [63, 68] as the base model, which can be converted into a masked discrete diffusion model by introducing a noise schedule (see App. D.3). MaskGIT employs a masked image transformer architecture trained on ImageNet [69] of $256 \times 256$ resolution, where each image amounts to a sequence of 256 discrete image tokens following VQ-GAN tokenization [70] ($d = 256$, $S = 1025$). We evaluate the $\theta$-Trapezoidal method against FHS, the Euler method, $\tau$-leaping, and parallel decoding under equivalent NFE budgets ranging from 4 to 64. Following the setting in [63], we generate $5 \times 10^4$ images and compute their Fréchet Inception Distance (FID) against the ImageNet validation split.

Fig. 3 reveals that $\theta$-Trapezoidal method (Alg. 2) consistently achieves lower (and thus better) FID values compared to both the Euler method and $\tau$-leaping across all NFE values. While FHS and parallel decoding show advantages at extremely low NFE ($\leq 8$), their performance saturates with increased computational resources, making them less favorable compared to our rapidly converging method. Additional results, including generated image samples (Fig. 7), are detailed in App. D.

**Remark 6.1** (Algorithm Hyperparameters). *We evaluate the performance of the $\theta$-Trapezoidal method across various $\theta$ and NFE values for both text and image generation tasks. As illustrated in Fig. 4, we observe that the $\theta$-Trapezoidal method demonstrates robustness to $\theta$, with a flat landscape near the optimal choice. Our empirical analysis suggests that $\theta \in [0.3, 0.5]$ consistently yields competitive performance across different tasks.*

### 6.4 Diffusion Large Language Model and Math Reasoning

To verify the effectiveness of the proposed method on a scale, we additionally benchmark its performance on LLaDA-Instruct [64], an 8B masked diffusion LLM with one of the best language modeling performances among discrete diffusion-based LLMs. We examine its performance on GSM8K [71], a math-reasoning dataset consisting of grade-school-level problems. We compare $\theta$-Trapezoidal to the semi-autoregressive (Semi-AR) sampler therein, with both confidence-based (Conf.) and purely random (Rand.) remasking strategies. For each method, we generate a response of 256 tokens in a zero-shot prompting manner, with NFE ranging from 64 to 256, and report the accuracy in Tab. 2. $\theta$-Trapezoidal outperforms the Semi-AR sampler in the low NFE regime, where NFE is strictly smaller than the sequence length. At high NFE regime, $\theta$-Trapezoidal exhibits a similarly competitive performance as other solvers. This observation accords with our claim that high-order samplers perform better with lower NFE budgets, and that these advantages persist even when the model size is large. Further implementation details are available at App. D.4.

Table 2: Response accuracy on GSM8K with different NFEs. The best results are in **bold**.

| Accuracy (%) | NFE $= 64$ | NFE $= 128$ | NFE $= 256$ |
|---|---|---|---|
| Semi-AR (Conf.) | 33.6 | 32.0 | 39.1 |
| Semi-AR (Rand.) | 33.8 | 34.3 | **40.3** |
| $\theta$-Trapezoidal | **35.1** | **38.4** | 39.7 |

## 7 Conclusion and Future Works

In this work, we introduce the $\theta$-RK-2 and $\theta$-Trapezoidal methods as pioneering high-order numerical schemes tailored for discrete diffusion model inference. Through rigorous analysis based on their stochastic integral formulations, we establish second-order convergence of the $\theta$-Trapezoidal method and that of the $\theta$-RK-2 method under specified conditions. Our analysis indicates that the $\theta$-Trapezoidal method generally provides superior robustness and computational efficiency compared to the $\theta$-RK-2 method. Our empirical evaluations, spanning both a 15-dimensional model with precise score functions and large-scale text and image generation tasks, validate our theoretical findings and demonstrate the superiority performance of our proposed $\theta$-Trapezoidal method over existing samplers in terms of sample quality under equivalent computational constraints. Additionally, we provide a comprehensive analysis of the method's robustness by examining the optimal choice of the parameter $\theta$ in our schemes.

Future research directions include comparative analysis of these schemes and development of more sophisticated numerical approaches for discrete diffusion model inference, potentially developing inference methods of higher order [56] or incorporating adaptive step sizes and parallel sampling methodologies. From the perspective of applications, these methods may also show promise for tasks in computational chemistry and biology, particularly in the design of molecules, proteins, and DNA sequences. Moreover, it would also be interesting to explore the usage of higher-order numerical solvers in other problem settings involving the inference of diffusion models, such as sampling via discrete diffusion models [72–75], inference-time scaling of diffusion models [76–91], improving the reasoning ability of diffusion large language models [92, 64, 93–99], etc.

## Acknowledgments and Disclosure of Funding

YC is supported by the National Science Foundation under Grants No. ECCS-1942523, DMS-2206576, and CMMI-2450378. GMR is supported by a Google Research Scholar Award. YZ and MT are grateful for partial supports by NSF Grants DMS-1847802, DMS-2513699, DOE Grants NA0004261, SC0026274, and Richard Duke Fellowship. YR, HC and LY acknowledge support of the National Science Foundation under Award No. DMS-2208163.

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

# A    Further Discussion on Related Works

In this section, we provide a more detailed literature review of both continuous and discrete diffusion models, as well as several studies on the numerical methods for SDEs and chemical reaction systems, which are highly related to our work.

**Discrete Diffusion Models: Methodology, Theory, and Applications.**    Discrete diffusion and flow-based models [1–4, 6–10, 22] have recently been proposed as generalizations of continuous diffusion models to model discrete distributions.

Such models have been widely used in various areas of science and engineering, including but not limited to modeling retrosynthesis [100], combinatorial optimization [101, 102], solving inverse problems [103, 104] and sampling high-dimensional discrete distributions [90, 72], designing molecules, proteins, and DNA sequences [105, 13, 106, 12, 107–109, 31, 110, 11, 111, 112], image synthesis [113–115], text summarization [15], as well as the generation of graph [116–123], layout [124, 125], motion [126, 127], sound [22, 128], image [16, 129–131], speech [132], electronic health record [133], tabular data [134] and text [135–140, 34, 35, 141, 14]. Inspired by the huge success achieved by discrete diffusion models in practice, researchers have also conducted some studies on the theoretical properties of these models, such as [36–38, 142].

An extensive amount of work has also explored the possibility of making discrete diffusion models more effective from many aspects, such as optimizing the sampling schedule [143], adding correctors [144], developing fast samplers [145], designing correctors based on information learnt by the model [144], simplifying the loss function for training [146], adding editing-based refinements [147], synergizing these models with other techniques and methodologies like distillation [148], Ehrenfest processes [149], Glauber dynamics [150], tensor networks [17], enhanced guidance mechanisms [151, 18–20], structured preferential generation [152], the plan-and-denoise framework [153] and alternative metrics, *e.g.*, the Fisher information metric [154]. However, to the best of our knowledge, existing work on accelerating the inference of discrete diffusion models is relatively sparse compared to the ones we listed above, which makes it a direction worth exploring and serves as one of the main motivations behind this work.

**Numerical Methods for SDEs and Chemical Reaction Systems.**    Below, we review advanced numerical methods proposed for simulating SDEs and chemical reaction systems, which are the main techniques adopted in our work. For the simulation of SDEs driven by Brownian motions, many studies have been performed to design more accurate numerical schemes, which have been widely applied to tackle problems in computational physics, optimization, and Monte Carlo sampling. Examples of such work include the Milstein method [155], explicit methods [156], multistep methods [157], extrapolation-type methods [158, 25], stochastic Runge Kutta methods [24, 159–162], splitting methods [163], methods based on gaussian mixtures [164], randomized midpoint method [165], parallel sampling methods [166, 167] as well as high-order methods for stochastic gradient Markov Chain Monte Carlo [168, 169], underdamped and overdamped Langevin Monte Carlo [170–174]. For a more comprehensive list of related numerical methods, one may refer to [39, 175, 176, 40, 58].

Regarding the simulation of chemical reaction systems, numerical methods can be categorized into two classes. The first class consists of exact simulation methods, which are similar to the Kinetic Monte Carlo (KMC) method [177] developed for simulating spin dynamics and crystal growth in condensed matter physics. Examples of such methods include the Gillespie algorithm (or the Stochastic Simulation Algorithm, a.k.a. SSA) [178, 179] and its variants for multiscale modeling [180–183], the next reaction method and its variants [184, 185], uniformization-based methods [186], etc. The second class of methods are approximate simulation methods, including but not limited to the $\tau$-leaping method [52] and its variants [187, 188, 53, 54, 189–195, 55, 196, 56, 197, 26, 198–201]. For a subset of the methods listed above, numerical analysis has also been performed in many works [202–206] to justify their validity.

**Continuous Diffusion Models: Methodology, Theory, and Acceleration.**    Continuous diffusion and probability flow-based models [207–216] have also been the most popular methods in generative modeling, with a wide range of applications in science and engineering. For a list of related work on the theoretical studies and applications of these models, one may refer to the literature review

conducted in [217, 38]. Here we will only review studies on accelerating the inference of continuous diffusion models, which motivates our work.

An incomplete list of accelerating methods includes approximate mean direction solver [218], restart sampling [219], gaussian mixture solvers [220], self-consistency [221–224], knowledge distillation [225–229], combination with underdamped Langevin dynamics [230], operator learning [231] and more recently ideas from accelerating large language models (LLMs) like caching [232] and speculative decoding [233]. Among all the proposed accelerating methods, one major class of methods are developed based on techniques from numerical analysis like adaptive step sizes [234], exponential integrators [41–43], predictor-corrector solver [235], Adams-Bashforth methods [29, 44, 45], Taylor methods [27, 46], Picard iteration and parallel sampling [236–240, 217], (stochastic) Runge-Kutta methods [47, 28, 48–51] and randomized midpoint method [241, 242]. In contrast, there have been fewer studies on the acceleration of discrete diffusion models via techniques from numerical analysis, which inspires the study undertaken in this paper.

# B  Mathematical Background

In this section, we provide the mathematical background for the stochastic integral formulation of discrete diffusion models, the error analysis of the $\tau$-leaping method, and useful lemmas for the theoretical analysis of high-order schemes for discrete diffusion models.

## B.1  Stochastic Integral Formulation of Discrete Diffusion Models

Throughout this section, we will assume that $(\Omega, \mathcal{F}, \mathbb{P})$ is a probability space, $\mathbb{X}$ is a finite-state space, and denote the pairwise difference set of the state space by $\mathbb{D} := \{x - y : x \neq y \in \mathbb{X}\}$. We also assume that the pairwise difference set $\mathbb{X}$ is equipped with a metric $\|\cdot\|$, a finite measure $\gamma$, and a $\sigma$-algebra $\mathcal{B}$.

As a warm-up, we introduce the definition of the Poisson random measure for a time-homogeneous counting process.

**Definition B.1** (Poisson Random Measure [38, Definition A.1]). *The random measure $N(\mathrm{d}t, \mathrm{d}\nu)$ on $\mathbb{R}^+ \times \mathbb{D}$ is called a* Poisson random measure *w.r.t. measure $\gamma$ if it is a random counting measure satisfying the following properties:*

*(i) For any $B \in \mathcal{B}$ and $0 \leq s < t$,*
$$N((s, t] \times B) \sim \mathcal{P}\left(\gamma(B)(t - s)\right);$$

*(ii) For any $t \geq 0$ and pairwise disjoint sets $\{B_i\}_{i \in [n]} \subset \mathcal{B}$,*
$$\{N_t(B_i) := N((0, t] \times B_i)\}_{i \in [n]}$$
*are independent stochastic processes.*

Then we define the Poisson random measure with evolving intensities. The term "evolving" refers to that the intensity is both time and state-dependent.

**Definition B.2** (Poisson Random Measure with Evolving Intensity [38, Definition A.3]). *Suppose $\lambda_t(y)$ is a non-negative predictable process on $\mathbb{R}^+ \times \mathbb{D} \times \Omega$ satisfying that for any $0 \leq T < \overline{T}$, $\int_0^T \lambda_t(\nu)\mathrm{d}t < \infty$, a.s..*

*The random measure $N[\lambda](\mathrm{d}t, \mathrm{d}\nu)$ on $\mathbb{R}^+ \times \mathbb{D}$ is called a* Poisson random measure *with* evolving intensity $\lambda_t(\nu)$ *w.r.t. measure $\gamma$ if it is a random counting measure satisfying the following properties:*

*(i) For any $B \in \mathcal{B}$ and $0 \leq s < t$,*
$$N[\lambda]((s, t] \times B) \sim \mathcal{P}\left(\int_s^t \int_B \lambda_\tau(\nu)\gamma(\mathrm{d}\nu)\mathrm{d}\tau\right);$$

*(ii) For any $t \geq 0$ and pairwise disjoint sets $\{B_i\}_{i \in [n]} \subset \mathcal{B}$,*
$$\{N_t[\lambda](B_i) := N[\lambda]((0, t] \times B_i)\}_{i \in [n]}$$
*are independent stochastic processes.*

**Remark B.3** (Construction of Poisson Random Measure with Evolving Intensity). *As discussed in Thm. A.4 in [38] and originally proposed by [243], the Poisson random measure with evolving intensity can be constructed in the following way.*

*One first augments the $(\mathbb{X}, \mathcal{B}, \nu)$ measure space to a product space $(\mathbb{D} \times \mathbb{R}, \mathcal{B} \times \mathcal{B}(\mathbb{R}), \gamma \times m)$, where $m$ is the Lebesgue measure on $\mathbb{R}$, and $\mathcal{B}(\mathbb{R})$ is the Borel $\sigma$-algebra on $\mathbb{R}$. The Poisson random measure with evolving intensity $\lambda_t(\nu)$ can be defined in the augmented measure space as*

$$N[\lambda]((s,t] \times B) := \int_s^t \int_B \int_{\mathbb{R}} \mathbf{1}_{0 \le \xi \le \lambda_\tau(\nu)} N(\mathrm{d}\tau, \mathrm{d}\nu, \mathrm{d}\xi), \tag{B.1}$$

*where $N(\mathrm{d}\tau, \mathrm{d}\nu, \mathrm{d}\xi)$ is the Poisson random measure on $\mathbb{R}^+ \times \mathbb{D} \times \mathbb{R}$ w.r.t. measure $\nu(\mathrm{d}y)\mathrm{d}\xi$.*

The following theorem provides the change of measure theorem for Poisson random measure with evolving intensity, which is crucial for the theoretical analysis of numerical schemes for discrete diffusion models.

**Theorem B.4** (Change of Measure for Poisson Random Measure with Evolving Density [38, Thm. 3.3]). *Let $N[\lambda](\mathrm{d}t, \mathrm{d}\nu)$ be a Poisson random measure with evolving intensity $\lambda_t(\nu)$, and $h_t(\nu)$ a positive predictable process on $\mathbb{R}^+ \times \mathbb{D} \times \Omega$. Suppose the following exponential process is a local $\mathcal{F}_t$-martingale:*

$$Z_t[h] := \exp\left( \int_0^t \int_{\mathbb{D}} \log h_t(\nu) N[\lambda](\mathrm{d}t \times \mathrm{d}\nu) - \int_0^t \int_{\mathbb{D}} (h_t(\nu) - 1)\lambda_t(\nu)\gamma(\mathrm{d}\nu) \right), \tag{B.2}$$

*and $\mathbb{Q}$ is another probability measure on $(\Omega, \mathcal{F})$ such that $\mathbb{Q} \ll \mathbb{P}$ with Radon-Nikodym derivative $\mathrm{d}\mathbb{Q}/\mathrm{d}\mathbb{P}|_{\mathcal{F}_t} = Z_t[h]$.*

*Then the Poisson random measure $N[\lambda](\mathrm{d}t, \mathrm{d}\nu)$ under the measure $\mathbb{Q}$ is a Poisson random measure with evolving intensity $\lambda_t(\nu)h_t(\nu)$.*

### B.2 Error Analysis of $\tau$-leaping

The $\tau$-leaping method was originally proposed by [52] and adopted for the inference of discrete diffusion models by [22]. A summary of the algorithm is given in Alg. 3. In this subsection, we provide a sketch for the error analysis of the $\tau$-leaping method when applied to discrete diffusion models, which will be compared with that of high-order schemes later on.

---

**Algorithm 3:** $\tau$-Leaping Method for Discrete Diffusion Model Inference

**Input:** $\widehat{y}_0 \sim q_0$, $\theta \in [0, 1]$, time discretization $(s_n, \rho_n)_{n \in [0:N-1]}$, $\widehat{\mu}, \widehat{\mu}^*$ as defined in Prop. C.2.
**Output:** A sample $\widehat{y}_{s_N} \sim \widehat{q}_{t_N}^{\mathrm{RK}}$.
  1 **for** $n = 0$ **to** $N - 1$ **do**
  2 $\quad \widehat{y}_{s_{n+1}} \leftarrow \widehat{y}_{s_n} + \sum_{\nu \in \mathbb{D}} \nu \mathcal{P}\left( \widehat{\mu}_{s_n}(\nu)\Delta_n \right);$
  3 **end**

---

*Proof of Thm. 3.1.* As we are considering the case where $\mathbb{X} = [S]^d$, *i.e.* the state space is a $d$-dimensional grid with $S$ states along each dimension, we have $\log |\mathbb{X}| = d \log S$. Then we consider a simple time-homogeneous transition matrix $\boldsymbol{Q}_t \equiv \boldsymbol{Q}$ that allows jumps between neighboring states with equal probability. Specifically, we have

$$Q(y, x) = \begin{cases} 1, & \|x - y\|_1 = 1, \\ -2d, & x = y, \end{cases}$$

which can be verified to satisfy Assumption 4.3(i) in [38] with $C = 1$ and $\underline{D} = \overline{D} = 2d$. Assumption 4.3(ii) is also satisfied, as shown in Example B.10 of [38].

Then we may apply Thm. 4.7 in [38] by using the required time discretization scheme according to the properties of the target distribution and plugging in the corresponding values of $C, \underline{D}, \overline{D}$. The result follows by scaling the transition matrix $\boldsymbol{Q}$ by $\frac{1}{d}$, equivalent to scaling the time by $d$. $\qquad \square$

# C Proofs

In this section, we provide the missing proofs in the main text. We will first provide the proofs of the stochastic integral formulations of high-order schemes for discrete diffusion models in App. C.1. Then we will provide the proofs of the main results for the $\theta$-Trapezoidal method in App. C.2 and the $\theta$-RK-2 method in App. C.3. We remark that the proof for the $\theta$-Trapezoidal method requires more techniques and is more involved, to which the proof for the $\theta$-RK-2 method is analogous. In App. C.4, we provide the detailed lemmas and computations omitted in the proofs of Thms. 5.4 and 5.5.

## C.1 Stochastic Integral Formulations of High-Order Schemes

In order to rigorously analyze the $\theta$-RK-2 method, we need the following definition:

**Definition C.1** (Intermediate Process). *We define the intermediate process $\widehat{y}_s^*$ piecewisely on $(s_n, s_{n+1}]$ as follows:*

$$\widehat{y}_s^* = \widehat{y}_{s_n} + \int_{s_n}^{s} \int_{\mathbb{D}} \nu N\left[\widehat{\mu}_{s_n}\right](\mathrm{d}s, \mathrm{d}\nu), \tag{C.1}$$

*where the intensity $\widehat{\mu}_{s_n}$ is given by $\widehat{\mu}_{s_n}(\nu, \widehat{y}_{s_n}) = \overleftarrow{\widehat{s}}_{s_n}(\widehat{y}_{s_n}, \widehat{y}_{s_n} + \nu)\overleftarrow{Q}_{s_n}^0(\widehat{y}_{s_n}, \widehat{y}_{s_n} + \nu)$, i.e., $\widehat{y}_s^*$ is the process obtained by performing $\tau$-leaping from time $s_n$ to $s$ with intensity $\widehat{\mu}$.*

The following proposition provides the stochastic integral formulation of this method.

**Proposition C.2** (Stochastic Integral Formulation of $\theta$-RK-2 Method). *The $\theta$-RK-2 method (Alg. 1) is equivalent to solving the following stochastic integral:*

$$\widehat{y}_s^{\mathrm{RK}} = \widehat{y}_0^{\mathrm{RK}} + \int_0^s \int_{\mathbb{D}} \nu N\left[\widehat{\mu}^{\mathrm{RK}}\right](\mathrm{d}s, \mathrm{d}\nu), \tag{C.2}$$

*in which the intensity $\widehat{\mu}^{\mathrm{RK}}$ is defined as a weighted sum*

$$\widehat{\mu}_s^{\mathrm{RK}}(\nu) = (1 - \tfrac{1}{2\theta})\widehat{\mu}_{\lfloor s \rfloor}(\nu, \widehat{y}_{\lfloor s \rfloor}^{\mathrm{RK}}) + \tfrac{1}{2\theta}\widehat{\mu}_{\rho_s}^*(\nu, \widehat{y}_{\rho_s}^*), \tag{C.3}$$

*and the intermediate intensity $\widehat{\mu}^*$ is defined piecewisely as*

$$\widehat{\mu}_s^*(\nu, \widehat{y}_s^*) = \overleftarrow{\widehat{s}}_s(\widehat{y}_s^*, \widehat{y}_s^* + \nu)\overleftarrow{Q}_s^0(\widehat{y}_s^*, \widehat{y}_s^* + \nu), \tag{C.4}$$

*with the intermediate process $\widehat{y}_s^*$ defined in (C.1) for the corresponding interval. We will call $\widehat{y}_s^{\mathrm{RK}}$ the* interpolating process *of the $\theta$-RK-2 method and denote the distribution of $\widehat{y}_s^{\mathrm{RK}}$ by $\widehat{q}_s^{\mathrm{RK}}$.*

The following proposition establishes the stochastic integral formulation of the $\theta$-Trapezoidal method, whose proof can be found in App. C.1.

**Proposition C.3** (Stochastic Integral Formulation of $\theta$-Trapezoidal Method). *The $\theta$-Trapezoidal method (Alg. 2) is equivalent to solving the following stochastic integral:*

$$\widehat{y}_s^{\mathrm{trap}} = \widehat{y}_0^{\mathrm{trap}} + \int_0^s \int_{\mathbb{D}} N[\widehat{\mu}^{\mathrm{trap}}](\mathrm{d}s, \mathrm{d}\nu) \tag{C.5}$$

*where the intensity $\widehat{\mu}^{\mathrm{trap}}$ is defined piecewisely as*

$$\widehat{\mu}_s^{\mathrm{trap}}(\nu) = \mathbf{1}_{s < \rho_s}\widehat{\mu}_{\lfloor s \rfloor}(\nu, \widehat{y}_{\lfloor s \rfloor}^{\mathrm{trap}}) + \mathbf{1}_{s \geq \rho_s}\left(\alpha_1\widehat{\mu}_{\rho_s}^*(\nu, \widehat{y}_{\rho_s}^*) - \alpha_2\widehat{\mu}_{\lfloor s \rfloor}(\nu, \widehat{y}_{\lfloor s \rfloor}^{\mathrm{trap}})\right)_+. \tag{C.6}$$

*Above, $\mathbf{1}_{(\cdot)}$ denotes the indicator function and the intermediate process $\widehat{y}_s^*$ is defined in (C.1) for the corresponding interval. We will call the process $\widehat{y}_s^{\mathrm{trap}}$ the* interpolating process *of the $\theta$-Trapezoidal method and denote the distribution of $\widehat{y}_s^{\mathrm{trap}}$ by $\widehat{q}_s^{\mathrm{trap}}$.*

*Proof of Props. C.2 and C.3.* Without loss of generality, we give the proof on the interval $(s_n, s_{n+1}]$ for $n \in [0 : N - 1]$, and the generalization to the whole interval $[0, T]$ is straightforward.

Notice that once we condition on the filtration $\mathcal{F}_{s_n}$ and construct the intermediate process $\widehat{y}_s^*$ as specified in (C.1) along the interval $(s_n, s_{n+1}]$, the intermediate intensity $\widehat{\mu}^*$ and the piecewise intensity $\widehat{\mu}_{\lfloor s \rfloor}$ do not evolve with time $s$ or the interpolating processes $\widehat{y}_s^{\mathrm{RK}}$ (or $\widehat{y}_s^{\mathrm{trap}}$, respectively) since it

only depends on the state, the intensity at the beginning of the interval $s_n$ and other randomness that is independent of the interpolating process.

Therefore, the stochastic integral on this interval can be rewritten as for the $\theta$-RK-2 scheme that

$$
\begin{aligned}
\widehat{y}_{s_{n+1}}^{\mathrm{RK}} &= \widehat{y}_{s_n}^{\mathrm{RK}} + \int_{s_n}^{s_{n+1}} \int_{\mathbb{D}} \nu N[\widehat{\mu}^{\mathrm{trap}}](\mathrm{d}s, \mathrm{d}\nu) \\
&= \widehat{y}_{s_n}^{\mathrm{RK}} + \int_{\mathbb{D}} \nu N[\widehat{\mu}^{\mathrm{RK}}]((s_n, s_{n+1}], \mathrm{d}\nu) \\
&= \widehat{y}_{s_n}^{\mathrm{RK}} + \int_{\mathbb{D}} \nu \mathcal{P}(\widehat{\mu}_{s_n}^{\mathrm{RK}}(\nu)(s_{n+1} - s_n))\gamma(\mathrm{d}\nu),
\end{aligned}
$$

and for the $\theta$-Trapezoidal scheme that

$$
\begin{aligned}
\widehat{y}_{s_{n+1}}^{\mathrm{trap}} &= \widehat{y}_{s_n}^{\mathrm{trap}} + \int_{s_n}^{s_{n+1}} \int_{\mathbb{D}} \nu N[\widehat{\mu}^{\mathrm{trap}}](\mathrm{d}s, \mathrm{d}\nu) \\
&= \widehat{y}_{s_n}^{\mathrm{trap}} + \int_{\mathbb{D}} \nu N[\widehat{\mu}^{\mathrm{trap}}]((s_n, s_{n+1}], \mathrm{d}\nu) \\
&= \widehat{y}_{s_n}^{\mathrm{trap}} + \int_{\mathbb{D}} \nu \mathcal{P}(\widehat{\mu}_{s_n}^{\mathrm{trap}}(\nu)(s_{n+1} - s_n))\gamma(\mathrm{d}\nu),
\end{aligned}
$$

and the statement follows by taking $\gamma(\mathrm{d}\nu)$ as the counting measure. $\qquad\square$

**Remark C.4** (Remark on Rejection Sampling and Periodicity Assumption). *The rejection sampling procedure in both algorithms (Algs. 1 and 2) guarantees well-posedness in the rare scenarios where a large drawn value of Poisson random variables or multiple simultaneous jumps in one coordinate would result in an update out of the state space $\mathbb{X} = [S]^d$. To enforce this, we simply allow at most one jump per update across the summation, for example, in the update*

$$
\widehat{y}_{\rho_n}^* \leftarrow \widehat{y}_{s_n} + \sum_{\nu \in \mathbb{D}} \nu \mathcal{P}\left(\widehat{\mu}_{s_n}(\nu)\theta\Delta_n\right),
$$

*as the standard practice in the literature [22, 38]. The indicator function $\mathbf{1}_{\widehat{\mu}_{s_n} > 0}$ in Alg. 1 is also used to ensure that only valid jumps from the current state $\widehat{y}_{s_n}$ are considered, while in Alg. 2, this is implicitly guaranteed by taking the positive part of $\alpha_1\widehat{\mu}_{\rho_n}^* - \alpha_2\widehat{\mu}_{s_n}$, which implies the positivity of $\alpha_1\widehat{\mu}_{\rho_n}^*$ and thus the validity of the jumps $\widehat{y}_{\rho_n}^*$. We point out that the single-jump rule is only a convenient sufficient condition, one should notice that this condition is not necessary for the well-posedness of our algorithms, since our setting of the state space $\mathbb{X}$ carries both orderliness and algebraic structure, and thus one could in principle admit multiple simultaneous jumps without ambiguity.*

*Over the full inference process, the total probability of rejection is at most $\mathcal{O}(\kappa)$. Below, we give a brief justification and we refer to Proposition A.14 in [38] for a complete proof of this claim. During the update aforementioned, the probability of at least two jumps occurring is bounded by*

$$
\mathbb{P}\left(\sum_{\nu \in \mathbb{D}} \mathcal{P}\left(\widehat{\mu}_{s_n}(\nu)\theta\Delta_n\right) > 1\right) = 1 - \mathbb{P}\left(\mathcal{P}\left(\sum_{\nu \in \mathbb{D}} \widehat{\mu}_{s_n}(\nu)\theta\Delta_n\right) \leq 1\right)
$$

$$
= 1 - \exp\left(-\sum_{\nu \in \mathbb{D}} \widehat{\mu}_{s_n}(\nu)\theta\Delta_n\right)\left(1 + \sum_{\nu \in \mathbb{D}} \widehat{\mu}_{s_n}(\nu)\theta\Delta_n\right)
$$

$$
\lesssim \left(\sum_{\nu \in \mathbb{D}} \widehat{\mu}_{s_n}(\nu)\theta\Delta_n\right)^2 \lesssim \Delta_n^2.
$$

*Summing $\mathcal{O}(\Delta_n^2)$ over $N$ steps gives $\sum_{n=0}^{N-1} \Delta_n^2 \lesssim \kappa T$, and an identical argument applies to the second update in each iteration. Hence, the overall rejection rate is at most $\mathcal{O}(\kappa)$.*

*When we impose periodic boundary conditions, $\mathbb{X} = [S]^d$ is equipped with a convenient algebraic structure: addition and scalar multiplication are globally well-defined. In that case, Algs. 1 and 2 match exactly the stochastic integral formulations in Props. C.2 and C.3. This alignment removes the need for per-step rejection, streamlines the application of the change-of-measure argument, and greatly simplifies the convergence proofs of Thms. 5.4 and 5.5. Even without periodicity, those theorems hold with probability at least $1 - \mathcal{O}(\kappa)$, as shown above.*

## C.2 Convergence Analysis of the $\theta$-Trapezoidal Method

**Theorem C.5.** *Let $\breve{p}_{0:T-\delta}$ and $\widehat{q}^{\mathrm{trap}}_{0:T-\delta}$ be the path measures of the backward process with the stochastic integral formulation (2.4) and the interpolating process (C.5) of the $\theta$-Trapezoidal method (Alg. 2), then it holds that*

$$D_{\mathrm{KL}}(\breve{p}_{T-\delta}\|\widehat{q}^{\mathrm{trap}}_{T-\delta}) \leq D_{\mathrm{KL}}(\breve{p}_{0:T-\delta}\|\widehat{q}^{\mathrm{trap}}_{0:T-\delta})$$

$$\leq D_{\mathrm{KL}}(\breve{p}_0\|\widehat{q}_0) + \mathbb{E}\left[\int_0^{T-\delta}\int_{\mathbb{D}}\left(\mu_s(\nu)\log\frac{\mu_s(\nu)}{\widehat{\mu}^{\mathrm{trap}}_s(\nu)} - \mu_s(\nu) + \widehat{\mu}^{\mathrm{trap}}_s(\nu)\right)\gamma(\mathrm{d}\nu)\mathrm{d}s\right], \qquad \text{(C.7)}$$

*where the intensity $\widehat{\mu}^{\mathrm{trap}}$ is defined in (C.5), and the expectation is taken w.r.t. both paths generated by the backward process (2.4) and the randomness of the Poisson random measure used in the first step of each iteration of the algorithm, i.e., the construction of the intermediate process (C.1), which is assumed to be independent of that of the backward process.*

*Proof.* First, we will handle the randomness introduced by the Poisson random measure in the first step of each iteration of the $\theta$-Trapezoidal method. For the ease of presentation, we encode the afore-mentioned randomness as a random variable $\zeta$ and suppose it is still supported on the probability space $(\Omega, \mathcal{F}, \mathbb{P})$ while being independent of the backward process. Then for each realization of $\zeta$, the intermediate process $\widehat{y}^*_s$ is constructed as in (C.1) and the corresponding intensity $\widehat{\mu}^*_s$ is defined in (C.4).

Given the stochastic integral formulation of the backward process (2.4) and the interpolating process of the $\theta$-Trapezoidal method (C.5), we have by Thm. B.4 that this particular realization of the path measure $\widehat{q}^{\mathrm{trap}}_{0:T-\delta}$ can be obtained by changing the path measure $\breve{p}_{0:T-\delta}$ with the Radon-Nikodym derivative

$$Z_t\left[\frac{\widehat{\mu}^{\mathrm{trap}}}{\mu}\right] = \exp\left(-\int_0^t\int_{\mathbb{D}}\log\frac{\mu_s(\nu)}{\widehat{\mu}^{\mathrm{trap}}_s(\nu)}N[\mu](\mathrm{d}s,\mathrm{d}\nu) + \int_0^t\int_{\mathbb{D}}\left(\mu_s(\nu) - \widehat{\mu}^{\mathrm{trap}}_s(\nu)\right)\gamma(\mathrm{d}\nu)\mathrm{d}s\right),$$

*i.e.,*

$$D_{\mathrm{KL}}(\breve{p}_{0:T-\delta}\|\widehat{q}^{\mathrm{trap}}_{0:T-\delta}|\zeta) = \mathbb{E}\left[\log Z^{-1}_{T-\delta}\left[\frac{\widehat{\mu}^{\mathrm{trap}}}{\mu}\right]\right]$$

$$= \mathbb{E}\left[\int_0^{T-\delta}\int_{\mathbb{D}}\left(\mu_s(\nu)\log\frac{\mu_s(\nu)}{\widehat{\mu}^{\mathrm{trap}}_s(\nu)} - \mu_s(\nu) + \widehat{\mu}^{\mathrm{trap}}_s(\nu)\right)\gamma(\mathrm{d}\nu)\mathrm{d}s\right].$$

Then it is easy to see by the data processing inequality and the chain rule of KL divergence that

$$D_{\mathrm{KL}}(\breve{p}_{T-\delta}\|\widehat{q}^{\mathrm{trap}}_{T-\delta}) \leq D_{\mathrm{KL}}(\breve{p}_{0:T-\delta}\|\widehat{q}^{\mathrm{trap}}_{0:T-\delta}) \leq \mathbb{E}\left[D_{\mathrm{KL}}(\breve{p}_{T-\delta}\|\widehat{q}^{\mathrm{trap}}_{T-\delta}|\zeta)\right]$$

$$= D_{\mathrm{KL}}(\breve{p}_0\|\widehat{q}_0) + \mathbb{E}\left[\int_0^{T-\delta}\int_{\mathbb{D}}\left(\mu_s(\nu)\log\frac{\mu_s(\nu)}{\widehat{\mu}^{\mathrm{trap}}_s(\nu)} - \mu_s(\nu) + \widehat{\mu}^{\mathrm{trap}}_s(\nu)\right)\gamma(\mathrm{d}\nu)\mathrm{d}s\right],$$

and the proof is complete. $\square$

In the following, we will provide the outline of the proof of Thm. 5.4, where we leave the proof of several lemmas and detailed calculations to App. C.4 for the clarity of presentation.

*Proof of Thm. 5.4.* Throughout this proof, including the subsequent lemmas and propositions that will be detailed in App. C.4, we will assume that $(y_s)_{s\in[0,T]}$ is a process generated by the path measure $\breve{p}_{0:T}$ of the backward process with the stochastic integral formulation (2.4) and set it as the underlying paths of the expectation in (C.7) as required by Thm. C.5. Especially, $y_s \sim \breve{p}_s$ holds for any $s \in [0,T]$. For simplicity, we will assume that the process $y_s$ is left-continuous at each grid point $s_i$ for $i \in [0:N]$, which happens with probability one.

We first consider the interval $(s_n, s_{n+1}]$ for $n \in [0 : N-1]$, and thus we have $\lfloor s \rfloor = s_n$ and $\rho_s = \rho_n$. Within this interval, we will denote its intermediate process as appeared in (C.1) as $y^*_s$, and the corresponding intermediate intensity as appeared in (C.4) as $\widehat{\mu}^*_s$. In the following discussion, we will assume implicitly that the processes are conditioned on the filtration $\mathcal{F}_{s_n}$.

By the definition of the intensity $\widehat{\mu}^{\mathrm{trap}}(\nu)$ as specified in (C.6)

$$\widehat{\mu}_s^{\mathrm{trap}} = \mathbf{1}_{s<\rho_s}\widehat{\mu}_{\lfloor s \rfloor} + \mathbf{1}_{s\geq\rho_s}\left(\alpha_1\widehat{\mu}_{\rho_s}^* - \alpha_2\widehat{\mu}_{\lfloor s \rfloor}\right)_+,$$

we can rewrite the corresponding part of the integral in (C.7) as

$$\int_{s_n}^{s_{n+1}} \int_{\mathbb{D}} \left( \mu_s(\nu) \log \frac{\mu_s(\nu)}{\widehat{\mu}_s^{\mathrm{trap}}(\nu)} - \mu_s(\nu) + \widehat{\mu}_s^{\mathrm{trap}}(\nu) \right) \gamma(\mathrm{d}\nu)\mathrm{d}s$$

$$= \left( \int_{s_n}^{\rho_n} + \int_{\rho_n}^{s_{n+1}} \right) \int_{\mathbb{D}} \left( \mu_s(\nu) \log \frac{\mu_s(\nu)}{\widehat{\mu}_s^{\mathrm{trap}}(\nu)} - \mu_s(\nu) + \widehat{\mu}_s^{\mathrm{trap}}(\nu) \right) \gamma(\mathrm{d}\nu)\mathrm{d}s$$

$$= \underbrace{\int_{s_n}^{\rho_n} \int_{\mathbb{D}} \left( \mu_s(\nu) \log \frac{\mu_s(\nu)}{\widehat{\mu}_{s_n}(\nu)} - \mu_s(\nu) + \widehat{\mu}_{s_n}(\nu) \right) \gamma(\mathrm{d}\nu)\mathrm{d}s}_{(\mathrm{I})}$$

$$+ \underbrace{\int_{\rho_n}^{s_{n+1}} \int_{\mathbb{D}} \left( \mu_s(\nu) \log \frac{\mu_s(\nu)}{\alpha_1\widehat{\mu}_{\rho_n}^*(\nu) - \alpha_2\widehat{\mu}_{s_n}(\nu)} - \mu_s(\nu) + \alpha_1\widehat{\mu}_{\rho_n}^*(\nu) - \alpha_2\widehat{\mu}_{s_n}(\nu) \right) \gamma(\mathrm{d}\nu)\mathrm{d}s}_{(\mathrm{II})},$$

where the assumption that $\alpha_1\widehat{\mu}_{\rho_s}^* - \alpha_2\widehat{\mu}_{\lfloor s \rfloor} \geq 0$ for all $s \in [0, T-\delta]$ is applied here for the second term (II) above.

**Decomposition of the Integral.** Next, we decompose the integral (I) and (II) into several terms, the magnitudes of which or combinations of which are to be bounded.

(i) The first term is decomposed as

$$(\mathrm{I}) = (\mathrm{I}.1) + (\mathrm{I}.2) + (\mathrm{I}.3) + (\mathrm{I}.4),$$

where each term is defined as

$$(\mathrm{I}.1) = \int_{s_n}^{\rho_n} \int_{\mathbb{D}} \left( \mu_{s_n}(\nu) \log \frac{\mu_{s_n}(\nu)}{\widehat{\mu}_{s_n}(\nu)} - \mu_{s_n}(\nu) + \widehat{\mu}_{s_n}(\nu) \right) \gamma(\mathrm{d}\nu)\mathrm{d}s,$$

$$(\mathrm{I}.2) = \int_{s_n}^{\rho_n} \int_{\mathbb{D}} \left( \mu_s(\nu) \log \mu_s(\nu) - \mu_s(\nu) - \mu_{s_n}(\nu) \log \mu_{s_n}(\nu) + \mu_{s_n}(\nu) \right) \gamma(\mathrm{d}\nu)\mathrm{d}s,$$

$$(\mathrm{I}.3) = \int_{s_n}^{\rho_n} \int_{\mathbb{D}} \left( \mu_s(\nu) - \mu_{s_n}(\nu) \right) \left( \log \left( \alpha_1\widehat{\mu}_{\rho_n}^*(\nu) - \alpha_2\widehat{\mu}_{s_n}(\nu) \right) - \log \widehat{\mu}_{s_n}(\nu) \right) \gamma(\mathrm{d}\nu)\mathrm{d}s,$$

$$(\mathrm{I}.4) = \int_{s_n}^{\rho_n} \int_{\mathbb{D}} \mu_{s_n}(\nu) \log \left( \alpha_1\widehat{\mu}_{\rho_n}^*(\nu) - \alpha_2\widehat{\mu}_{s_n}(\nu) \right) \gamma(\mathrm{d}\nu)\mathrm{d}s$$

$$- \int_{s_n}^{\rho_n} \int_{\mathbb{D}} \mu_s(\nu) \log \left( \alpha_1\widehat{\mu}_{\rho_n}^*(\nu) - \alpha_2\widehat{\mu}_{s_n}(\nu) \right) \gamma(\mathrm{d}\nu)\mathrm{d}s.$$

(ii) The second term is decomposed as

$$(\mathrm{II}) = (\mathrm{II}.1) + (\mathrm{II}.2) + (\mathrm{II}.3) + (\mathrm{II}.4) + (\mathrm{II}.5) + (\mathrm{II}.6),$$

where each term is defined as

$$(\mathrm{II}.1) = \alpha_1 \int_{\rho_n}^{s_{n+1}} \int_{\mathbb{D}} \left( \mu_{\rho_n}(\nu) \log \frac{\mu_{\rho_n}(\nu)}{\widehat{\mu}_{\rho_n}(\nu)} - \mu_{\rho_n}(\nu) + \widehat{\mu}_{\rho_n}(\nu) \right) \gamma(\mathrm{d}\nu)\mathrm{d}s$$

$$- \alpha_2 \int_{\rho_n}^{s_{n+1}} \int_{\mathbb{D}} \left( \mu_{s_n}(\nu) \log \frac{\mu_{s_n}(\nu)}{\widehat{\mu}_{s_n}(\nu)} - \mu_{s_n}(\nu) + \widehat{\mu}_{s_n}(\nu) \right) \gamma(\mathrm{d}\nu)\mathrm{d}s,$$

$$(\mathrm{II}.2) = \int_{\rho_n}^{s_{n+1}} \int_{\mathbb{D}} \left( \mu_s(\nu) \log \mu_s(\nu) - \mu_s(\nu) \right) \gamma(\mathrm{d}\nu)\mathrm{d}s$$

$$- \int_{\rho_n}^{s_{n+1}} \int_{\mathbb{D}} \left( \alpha_1(\mu_{\rho_n}(\nu) \log \mu_{\rho_n}(\nu) - \mu_{\rho_n}(\nu)) - \alpha_2(\mu_{s_n}(\nu) \log \mu_{s_n}(\nu) - \mu_{s_n}(\nu)) \right) \gamma(\mathrm{d}\nu)\mathrm{d}s,$$

$$(\text{II.3}) = \int_{\rho_n}^{s_{n+1}} \int_{\mathbb{D}} \alpha_1 \left( \widehat{\mu}^*_{\rho_n}(\nu) - \widehat{\mu}_{\rho_n}(\nu) \right) \gamma(\mathrm{d}\nu)\mathrm{d}s,$$

$$(\text{II.4}) = \int_{\rho_n}^{s_{n+1}} \int_{\mathbb{D}} \left( \alpha_1 \mu_{\rho_n}(\nu) \log \widehat{\mu}_{\rho_n}(\nu) - \alpha_2 \mu_{s_n}(\nu) \log \widehat{\mu}_{s_n}(\nu) \right) \gamma(\mathrm{d}\nu)\mathrm{d}s$$
$$- \int_{\rho_n}^{s_{n+1}} \int_{\mathbb{D}} \left( \alpha_1 \mu_{\rho_n}(\nu) - \alpha_2 \mu_{s_n}(\nu) \right) \log \left( \alpha_1 \widehat{\mu}_{\rho_n}(\nu) - \alpha_2 \widehat{\mu}_{s_n}(\nu) \right) \gamma(\mathrm{d}\nu)\mathrm{d}s,$$

$$(\text{II.5}) = \int_{\rho_n}^{s_{n+1}} \int_{\mathbb{D}} \left( \alpha_1 \mu_{\rho_n}(\nu) - \alpha_2 \mu_{s_n}(\nu) \right) \log \left( \alpha_1 \widehat{\mu}_{\rho_n}(\nu) - \alpha_2 \widehat{\mu}_{s_n}(\nu) \right) \gamma(\mathrm{d}\nu)\mathrm{d}s$$
$$- \int_{\rho_n}^{s_{n+1}} \int_{\mathbb{D}} \left( \alpha_1 \mu_{\rho_n}(\nu) - \alpha_2 \mu_{s_n}(\nu) \right) \log \left( \alpha_1 \widehat{\mu}^*_{\rho_n}(\nu) - \alpha_2 \widehat{\mu}_{s_n}(\nu) \right) \gamma(\mathrm{d}\nu)\mathrm{d}s,$$

$$(\text{II.6}) = \int_{\rho_n}^{s_{n+1}} \int_{\mathbb{D}} \left( \alpha_1 \mu_{\rho_n}(\nu) - \alpha_2 \mu_{s_n}(\nu) \right) \log \left( \alpha_1 \widehat{\mu}^*_{\rho_n}(\nu) - \alpha_2 \widehat{\mu}_{s_n}(\nu) \right) \gamma(\mathrm{d}\nu)\mathrm{d}s$$
$$- \int_{\rho_n}^{s_{n+1}} \int_{\mathbb{D}} \mu_s(\nu) \log \left( \alpha_1 \widehat{\mu}^*_{\rho_n}(\nu) - \alpha_2 \widehat{\mu}_{s_n}(\nu) \right) \gamma(\mathrm{d}\nu)\mathrm{d}s.$$

**Bounding the Error Terms.** Then we briefly summarize the intuitions and related techniques used in the bounds of the terms above, and the detailed calculations and proofs of the lemmas and propositions are deferred to App. C.4.

(i) *Error due to estimation error associated with the intensity:* The terms (I.1) and (II.1) are bounded by the assumption on the estimation error of the intensity $\widehat{\mu}_s$ (Assump. 5.3), as

$$\mathbb{E}\left[(\text{I.1}) + (\text{II.1})\right] \leq \theta \Delta_n \epsilon_{\text{I}} + \alpha_1 (1 - \theta) \Delta_n \epsilon_{\text{I}} = \theta \Delta_n \epsilon_{\text{I}} + \tfrac{1}{2\theta} \Delta_n \epsilon_{\text{I}} \lesssim \Delta_n \epsilon_{\text{I}},$$

for any $\theta \in (0, 1]$.

The term (II.4) is bounded by Prop. C.9, as

$$\mathbb{E}\left[(\text{II.4})\right] \lesssim \Delta_n \epsilon_{\text{II}},$$

where Jensen's inequality is applied here based on the convexity of the loss.

(ii) *Error related to the smoothness of intensity:* By Cor. C.13, the terms (I.2) and (II.2) are bounded by
$$\mathbb{E}\left[(\text{I.2}) + (\text{II.2})\right] \leq \Delta_n^3.$$

By Cor. C.14, the terms (I.4) and (II.6) are bounded by

$$\mathbb{E}\left[(\text{I.4}) + (\text{II.6})\right] \leq \Delta_n^3.$$

Intuitively, the bounds on these terms closely relate to the properties of the jump process and quantify the smoothness assumption on the intensity $\mu_s$ (Assump. 5.2), especially when the intensity does not vary significantly within the interval $(s_n, s_{n+1}]$. The main technique used for bounding these terms is Dynkin's Formula (Thm. C.10). The third-order accuracy here directly follows from the intuition provided in Sec. 4 based on numerical quadrature.

(iii) *Error involving the intermediate process:* The terms (II.3) and (II.5) are bounded by Prop. C.18 and Cor. C.19 respectively as follows

$$\mathbb{E}\left[(\text{II.3})\right] \lesssim \Delta_n^3 + \Delta_n^2 \epsilon_{\text{II}}, \quad \text{and} \quad \mathbb{E}\left[(\text{II.5})\right] \lesssim \Delta_n^3 + \Delta_n^2 \epsilon_{\text{II}},$$

The term (I.3) is bounded by Prop. C.20 as below

$$\mathbb{E}\left[(\text{I.3})\right] \lesssim \Delta_n^3.$$

The three terms above all involve the intermediate process $y_s^*$ and the corresponding intermediate density $\widehat{\mu}_s^*$.

In conclusion, by summing up all these terms, we have

$$\int_{s_n}^{s_{n+1}} \int_{\mathbb{D}} \left( \mu_s(\nu) \log \frac{\mu_s(\nu)}{\widehat{\mu}_s^{\mathrm{trap}}(\nu)} - \mu_s(\nu) + \widehat{\mu}_s^{\mathrm{trap}}(\nu) \right) \gamma(\mathrm{d}\nu)\mathrm{d}s$$
$$\lesssim \Delta_n(\epsilon_{\mathrm{I}} + \epsilon_{\mathrm{II}}) + \Delta_n^3 + \Delta_n^2 \epsilon_{\mathrm{II}} \lesssim \Delta_n(\epsilon_{\mathrm{I}} + \epsilon_{\mathrm{II}}) + \Delta_n^3.$$

Therefore, the overall error is bounded by first applying Thm. C.5 and then the upper bound derived above to each interval $(s_n, s_{n+1}]$, which yields

$$D_{\mathrm{KL}}(\breve{p}_{T-\delta} \| \widehat{q}_{T-\delta}^{\mathrm{trap}})$$
$$\leq D_{\mathrm{KL}}(\breve{p}_0 \| \widehat{q}_0) + \mathbb{E} \left[ \int_0^{T-\delta} \int_{\mathbb{D}} \left( \mu_s(\nu) \log \frac{\mu_s(\nu)}{\widehat{\mu}_s^{\mathrm{trap}}(\nu)} - \mu_s(\nu) + \widehat{\mu}_s^{\mathrm{trap}}(\nu) \right) \gamma(\mathrm{d}\nu)\mathrm{d}s \right]$$
$$\lesssim D_{\mathrm{KL}}(\breve{p}_0 \| \widehat{q}_0) + \sum_{n=0}^{N-1} \left( \Delta_n(\epsilon_{\mathrm{I}} + \epsilon_{\mathrm{II}}) + \Delta_n^3 \right)$$
$$\lesssim \exp(-T) + T(\epsilon_{\mathrm{I}} + \epsilon_{\mathrm{II}}) + \kappa^2 T,$$

as desired. $\qquad\square$

**Remark C.6** (Discussion on the Positivity Assumption). *In the following, we will take the positivity assumption in Thm. 5.4 as an example, and the case of the $\theta$-RK-2 method is similar. In the statement of Thm. 5.4, we have assumed that*

$$\alpha_1 \widehat{\mu}_{\rho_s}^*(\nu) - \alpha_2 \widehat{\mu}_{\lfloor s \rfloor}(\nu) \geq 0$$

*in (C.6) for all $s \in [0, T-\delta]$, which allows us to replace $\left( \alpha_1 \widehat{\mu}_{\rho_s}^*(\nu) - \alpha_2 \widehat{\mu}_{\lfloor s \rfloor}(\nu) \right)_+$ by the difference itself. [25] showed that this approximation is at most of $\mathcal{O}(\Delta_n^3)$ within the corresponding interval, and [26] further proved that for any order $p \geq 1$, there exists a sufficiently small step size $\Delta$ such that this approximation is at least $p$-th order, i.e., of order $\mathcal{O}(\Delta^p)$ for that step.*

*We give a brief justification of this assumption here. We consider the expectation of the difference itself, which is given by*

$$\mathbb{E}\left[ \alpha_1 \widehat{\mu}_{\rho_s}^*(\nu) - \alpha_2 \widehat{\mu}_{\lfloor s \rfloor}(\nu) \right] = \mathbb{E}\left[ \widehat{\mu}_{\lfloor s \rfloor}(\nu) + \alpha_1 \left( \widehat{\mu}_{\rho_s}^*(\nu) - \widehat{\mu}_{\rho_s}(\nu) \right) + \alpha_1 \left( \widehat{\mu}_{\rho_s}(\nu) - \widehat{\mu}_{\lfloor s \rfloor}(\nu) \right) \right]$$
$$\gtrsim 1 - \alpha_1(\kappa \epsilon_{\mathrm{II}} + \kappa) = 1 - \mathcal{O}(\kappa),$$

*where we used $\mathbb{E}\left[ |\widehat{\mu}_{\rho_s}^*(\nu) - \widehat{\mu}_{\rho_s}(\nu)| \right] \lesssim \kappa \epsilon_{\mathrm{II}}$, as established in (C.17) and $\mathbb{E}\left[ |\widehat{\mu}_{\rho_s}(\nu) - \widehat{\mu}_{\lfloor s \rfloor}(\nu)| \right] \lesssim \kappa$, as shown in (C.18). Therefore, as long as the step sizes $\Delta_n$ are sufficiently small, the positivity assumption is valid in the sense that the expectation of the difference is at least $1 - \mathcal{O}(\kappa)$.*

## C.3 Convergence Analysis of the $\theta$-RK-2 Method

Here we may again apply the data processing inequality and the chain rule of KL divergence to upper bound the error associated with the $\theta$-RK-2 method. A statement of the upper bound is provided in Thm. C.7 below, whose proof is omitted here since it is similar to that of Thm. C.5 above.

**Theorem C.7.** *Let $\breve{p}_{0:T-\delta}$ and $\widehat{q}_{0:T-\delta}^{\mathrm{RK}}$ be the path measures of the backward process with the stochastic integral formulation (2.4) and the interpolating process (C.2) of the $\theta$-RK-2 method ( Alg. 1), then it holds that*

$$D_{\mathrm{KL}}(\breve{p}_{T-\delta} \| \widehat{q}_{T-\delta}^{\mathrm{RK}}) \leq D_{\mathrm{KL}}(\breve{p}_{0:T-\delta} \| \widehat{q}_{0:T-\delta}^{\mathrm{RK}})$$
$$\leq D_{\mathrm{KL}}(\breve{p}_0 \| \widehat{q}_0) + \mathbb{E}\left[ \int_0^{T-\delta} \int_{\mathbb{D}} \left( \mu_s(\nu) \log \frac{\mu_s(\nu)}{\widehat{\mu}_s^{\mathrm{RK}}(\nu)} - \mu_s(\nu) + \widehat{\mu}_s^{\mathrm{RK}}(\nu) \right) \gamma(\mathrm{d}\nu)\mathrm{d}s \right], \qquad \text{(C.8)}$$

*where the intensity $\widehat{\mu}^{\mathrm{RK}}$ is defined in (C.2), and the expectation is taken w.r.t. both paths generated by the backward process (2.4) and the randomness of the Poisson random measure used in the first step of each iteration of the algorithm, i.e., the construction of the intermediate process (C.1), which is assumed to be independent of that of the backward process.*

Following the same flow as in the proof of Thm. 5.4, we will first provide an outline of the proof of Thm. 5.5, and defer the proof of several key lemmas and detailed calculations to App. C.4 for the clarity of presentation. We will also comment on the differences that may lead to the less desirable numerical properties of the $\theta$-RK-2 method.

*Proof of Thm. 5.5.* In the following proof sketch, we will be using the same notation as in the proof of Thm. 5.4, and we will assume that the process $y_s$ is left-continuous at each grid point $s_i$ for $i \in [0 : N]$. We also start by taking a closer look at the integral within each interval $(s_n, s_{n+1}]$ for $n \in [0 : N-1]$, and denote the intermediate process as appeared in (C.1) as $y_s^*$ and the corresponding intermediate intensity as appeared in (C.4) as $\widehat{\mu}_s^*$.

As defined in (C.3), the intensity $\widehat{\mu}_s^{\mathrm{RK}}(\nu)$ is given by

$$\widehat{\mu}_s^{\mathrm{RK}}(\nu) = \left(1 - \tfrac{1}{2\theta}\right) \widehat{\mu}_{\lfloor s \rfloor}(\nu) + \tfrac{1}{2\theta} \widehat{\mu}_{\rho_s}^*(\nu),$$

which helps us rewrite the corresponding part of the integral in (C.8) as

$$\int_{s_n}^{s_{n+1}} \int_{\mathbb{D}} \left( \mu_s(\nu) \log \frac{\mu_s(\nu)}{\widehat{\mu}_s^{\mathrm{RK}}(\nu)} - \mu_s(\nu) + \widehat{\mu}_s^{\mathrm{RK}}(\nu) \right) \gamma(\mathrm{d}\nu)\mathrm{d}s$$

$$= \int_{s_n}^{s_{n+1}} \int_{\mathbb{D}}$$

$$\underbrace{\left( \mu_s(\nu) \log \frac{\mu_s(\nu)}{(1 - \tfrac{1}{2\theta})\widehat{\mu}_{s_n}(\nu) + \tfrac{1}{2\theta}\widehat{\mu}_{\rho_n}^*(\nu)} - \mu_s(\nu) + \left(1 - \tfrac{1}{2\theta}\right) \widehat{\mu}_{s_n}(\nu) + \tfrac{1}{2\theta}\widehat{\mu}_{\rho_n}^*(\nu) \right) \gamma(\mathrm{d}\nu)\mathrm{d}s}_{\text{(III)}} .$$

Above we again use the positivity assumption that $(1 - \tfrac{1}{2\theta})\widehat{\mu}_{\lfloor s \rfloor} + \tfrac{1}{2\theta}\widehat{\mu}_{\rho_s}^* \geq 0$ for the term (III) above, just as what we have done in the proof and discussion of Thm. 5.4 above.

**Decomposition of the Integral.** Then we perform a similar decomposition of the integral as in the proof of Thm. 5.4 as follows:

$$\text{(III)} = \text{(III.1)} + \text{(III.2)} + \text{(III.3)} + \text{(III.4)} + \text{(III.5)} + \text{(III.6)},$$

where each term is defined as

$$\text{(III.1)} = \left(1 - \tfrac{1}{2\theta}\right) \int_{s_n}^{s_{n+1}} \int_{\mathbb{D}} \left( \mu_{s_n}(\nu) \log \left( \frac{\mu_{s_n}(\nu)}{\widehat{\mu}_{s_n}(\nu)} \right) - \mu_{s_n}(\nu) + \widehat{\mu}_{s_n}(\nu) \right) \gamma(\mathrm{d}\nu)\mathrm{d}s$$

$$+ \tfrac{1}{2\theta} \int_{s_n}^{s_{n+1}} \int_{\mathbb{D}} \left( \mu_{\rho_n}(\nu) \log \left( \frac{\mu_{\rho_n}(\nu)}{\widehat{\mu}_{\rho_n}(\nu)} \right) - \mu_{\rho_n}(\nu) + \widehat{\mu}_{\rho_n}(\nu) \right) \gamma(\mathrm{d}\nu)\mathrm{d}s,$$

$$\text{(III.2)} = \int_{s_n}^{s_{n+1}} \int_{\mathbb{D}} \left( \mu_s(\nu) \log \mu_s(\nu) - \mu_s(\nu) \right) \gamma(\mathrm{d}\nu)\mathrm{d}s$$

$$- \int_{s_n}^{s_{n+1}} \int_{\mathbb{D}} \left( \left(1 - \tfrac{1}{2\theta}\right) \left( \mu_{s_n}(\nu) \log \mu_{s_n}(\nu) - \mu_{s_n} \right) + \tfrac{1}{2\theta} \left( \mu_{\rho_n}(\nu) \log \mu_{\rho_n}(\nu) - \mu_{\rho_n}(\nu) \right) \right) \gamma(\mathrm{d}\nu)\mathrm{d}s,$$

$$\text{(III.3)} = \int_{s_n}^{s_{n+1}} \int_{\mathbb{D}} \tfrac{1}{2\theta} \left( \widehat{\mu}_{\rho_n}^*(\nu) - \widehat{\mu}_{\rho_n}(\nu) \right) \gamma(\mathrm{d}\nu)\mathrm{d}s,$$

$$\text{(III.4)} = \int_{s_n}^{s_{n+1}} \int_{\mathbb{D}} \left( \left(1 - \tfrac{1}{2\theta}\right) \mu_{s_n}(\nu) \log \widehat{\mu}_{s_n}(\nu) + \tfrac{1}{2\theta} \mu_{\rho_n}(\nu) \log \widehat{\mu}_{\rho_n}(\nu) \right) \gamma(\mathrm{d}\nu)\mathrm{d}s$$

$$- \int_{s_n}^{s_{n+1}} \int_{\mathbb{D}} \left( \left(1 - \tfrac{1}{2\theta}\right) \mu_{s_n}(\nu) + \tfrac{1}{2\theta} \mu_{\rho_n}(\nu) \right) \log \left( \left(1 - \tfrac{1}{2\theta}\right) \widehat{\mu}_{s_n}(\nu) + \tfrac{1}{2\theta} \widehat{\mu}_{\rho_n}(\nu) \right) \gamma(\mathrm{d}\nu)\mathrm{d}s,$$

$$\text{(III.5)} = \int_{s_n}^{s_{n+1}} \int_{\mathbb{D}} \left( \left(1 - \tfrac{1}{2\theta}\right) \mu_{s_n}(\nu) + \tfrac{1}{2\theta} \mu_{\rho_n}(\nu) \right) \log \left( \left(1 - \tfrac{1}{2\theta}\right) \widehat{\mu}_{s_n}(\nu) + \tfrac{1}{2\theta} \widehat{\mu}_{\rho_n}(\nu) \right) \gamma(\mathrm{d}\nu)\mathrm{d}s$$

$$- \int_{s_n}^{s_{n+1}} \int_{\mathbb{D}} \left( \left(1 - \tfrac{1}{2\theta}\right) \mu_{s_n}(\nu) + \tfrac{1}{2\theta} \mu_{\rho_n}(\nu) \right) \log \left( \left(1 - \tfrac{1}{2\theta}\right) \widehat{\mu}_{s_n}(\nu) + \tfrac{1}{2\theta} \widehat{\mu}_{\rho_n}^*(\nu) \right) \gamma(\mathrm{d}\nu)\mathrm{d}s,$$

$$\text{(III.6)} = \int_{s_n}^{s_{n+1}} \int_{\mathbb{D}} \left( \left(1 - \tfrac{1}{2\theta}\right) \mu_{s_n}(\nu) + \tfrac{1}{2\theta} \mu_{\rho_n}(\nu) \right) \log \left( \left(1 - \tfrac{1}{2\theta}\right) \widehat{\mu}_{s_n}(\nu) + \tfrac{1}{2\theta} \widehat{\mu}_{\rho_n}^*(\nu) \right) \gamma(\mathrm{d}\nu)\mathrm{d}s$$

$$- \int_{s_n}^{s_{n+1}} \int_{\mathbb{D}} \mu_s(\nu) \log \left( \left(1 - \tfrac{1}{2\theta}\right) \widehat{\mu}_{s_n}(\nu) + \tfrac{1}{2\theta} \widehat{\mu}_{\rho_n}^*(\nu) \right) \gamma(\mathrm{d}\nu) \mathrm{d}s.$$

**Bounding the Error Terms.** Then we briefly summarize the intuitions and related techniques used in the bound of the terms above,. Detailed calculations and proofs of the lemmas and propositions used here are deferred to App. C.4.

(i) *Error due to the intensity estimation:* The terms in (III.1) are bounded by the assumption on the estimation error of the intensity $\widehat{\mu}_s$ (Assump. 5.3) as follows

$$\mathbb{E}\left[(\text{III.1})\right] \leq \left(1 - \tfrac{1}{2\theta}\right) \Delta_n \epsilon_{\mathrm{I}} + \tfrac{1}{2\theta} \Delta_n \epsilon_{\mathrm{I}} = \Delta_n \epsilon_{\mathrm{I}},$$

for any $\theta \in (0, 1]$.

(ii) *Error related to the smoothness of intensity:* By Cors. C.16 and C.17, the terms (III.2) and (III.6) are bounded by

$$\mathbb{E}\left[(\text{III.2})\right] \leq \Delta_n^3, \quad \text{and} \quad \mathbb{E}\left[(\text{III.6})\right] \leq \Delta_n^3,$$

respectively.

(iii) *Error involving the intermediate process:* The term (III.3) and (III.5) are bounded in almost the same way as that of Prop. C.18 and Cor. C.19. By simply altering the integral upper limits, we obtain that

$$\mathbb{E}\left[(\text{III.3})\right] \lesssim \Delta_n^3 + \Delta_n^2 \epsilon_{\mathrm{II}}, \quad \mathbb{E}\left[(\text{III.5})\right] \lesssim \Delta_n^3 + \Delta_n^2 \epsilon_{\mathrm{II}}.$$

The only term that cannot be directly bounded based on results in App. C.4 is (III.4), which is given by

$$\mathbb{E}\left[(\text{III.4})\right] = \mathbb{E}\left[ \int_{s_n}^{s_{n+1}} \int_{\mathbb{D}} \left( \left(1 - \tfrac{1}{2\theta}\right) \mu_{s_n}(\nu) \log \widehat{\mu}_{s_n}(\nu) + \tfrac{1}{2\theta} \mu_{\rho_n}(\nu) \log \widehat{\mu}_{\rho_n}(\nu) \right) \gamma(\mathrm{d}\nu) \mathrm{d}s \right.$$
$$\left. - \int_{s_n}^{s_{n+1}} \int_{\mathbb{D}} \left( \left(1 - \tfrac{1}{2\theta}\right) \mu_{s_n}(\nu) + \tfrac{1}{2\theta} \mu_{\rho_n}(\nu) \right) \log \left( \left(1 - \tfrac{1}{2\theta}\right) \widehat{\mu}_{s_n}(\nu) + \tfrac{1}{2\theta} \widehat{\mu}_{\rho_n}(\nu) \right) \gamma(\mathrm{d}\nu) \mathrm{d}s \right]$$
(C.9)

Recall that in the proof of its counterpart (Prop. C.9), we utilized the convexity of the loss function and the extrapolation nature of the second step in the $\theta$-Trapezoidal method (C.6) to bound the error term. However, the same technique cannot be directly applied to the $\theta$-RK-2 method for any $\theta \in [0, 1]$, as the intensity $\widehat{\mu}_s^{\mathrm{RK}}$ is an interpolation of the intensity $\widehat{\mu}_s$ when $\theta \in (\tfrac{1}{2}, 1]$. Therefore, below we will first focus on the case when $\theta \in (0, \tfrac{1}{2}]$.

To be specific, by the assumption on the estimation error (Assump. 5.3), we can reduce (C.9) to

$$\mathbb{E}\left[ \int_{s_n}^{s_{n+1}} \int_{\mathbb{D}} \left( \left(1 - \tfrac{1}{2\theta}\right) \widehat{\mu}_{s_n}(\nu) \log \widehat{\mu}_{s_n}(\nu) + \tfrac{1}{2\theta} \mu_{\rho_n}(\nu) \log \widehat{\mu}_{\rho_n}(\nu) \right) \right.$$
$$\left. - \int_{s_n}^{s_{n+1}} \int_{\mathbb{D}} \left( \left(1 - \tfrac{1}{2\theta}\right) \widehat{\mu}_{s_n}(\nu) + \tfrac{1}{2\theta} \widehat{\mu}_{\rho_n}(\nu) \right) \log \left( \left(1 - \tfrac{1}{2\theta}\right) \widehat{\mu}_{s_n}(\nu) + \tfrac{1}{2\theta} \widehat{\mu}_{\rho_n}(\nu) \right) \gamma(\mathrm{d}\nu) \mathrm{d}s \right],$$
(C.10)

which can then be upper bounded based on Jensen's inequality and the convexity of the loss function for $\theta \in (0, \tfrac{1}{2}]$.

Summing up the bounds of the terms above, we have

$$\int_{s_n}^{s_{n+1}} \int_{\mathbb{D}} \left( \mu_s(\nu) \log \frac{\mu_s(\nu)}{\widehat{\mu}_s^{\mathrm{RK}}(\nu)} - \mu_s(\nu) + \widehat{\mu}_s^{\mathrm{RK}}(\nu) \right) \gamma(\mathrm{d}\nu) \mathrm{d}s$$
$$\lesssim \Delta_n(\epsilon_{\mathrm{I}} + \epsilon_{\mathrm{II}}) + \Delta_n^3 + \Delta_n^2 \epsilon_{\mathrm{II}} \lesssim \Delta_n(\epsilon_{\mathrm{I}} + \epsilon_{\mathrm{II}}) + \Delta_n^3,$$

Consequently, the overall error of the $\theta$-RK-2 method is bounded by

$$D_{\mathrm{KL}}(\breve{p}_{T-\delta}\|\widehat{q}_{T-\delta}^{\mathrm{RK}})$$

$$\leq D_{\mathrm{KL}}(\breve{p}_0\|\widehat{q}_0) + \mathbb{E}\left[\int_0^{T-\delta}\int_{\mathbb{D}}\left(\mu_s(\nu)\log\frac{\mu_s(\nu)}{\widehat{\mu}_s^{\mathrm{RK}}(\nu)} - \mu_s(\nu) + \widehat{\mu}_s^{\mathrm{RK}}(\nu)\right)\gamma(\mathrm{d}\nu)\mathrm{d}s\right]$$

$$\lesssim D_{\mathrm{KL}}(\breve{p}_0\|\widehat{q}_0) + \sum_{n=0}^{N-1}\left(\Delta_n(\epsilon_{\mathrm{I}} + \epsilon_{\mathrm{II}}) + \Delta_n^3\right)$$

$$\lesssim \exp(-T) + T(\epsilon_{\mathrm{I}} + \epsilon_{\mathrm{II}}) + \kappa^2 T,$$

which suggests that the $\theta$-RK-2 is also of second order when $\theta \in (0, \frac{1}{2}]$. For the other case when $\theta \in (\frac{1}{2}, 1]$, we will provide a brief discussion in the remark below. $\qquad\square$

**Remark C.8** (Discussions on the case when $\theta \in (\frac{1}{2}, 1]$). *For $\theta \in (\frac{1}{2}, 1]$, the term (C.10) is positive and thus not necessarily bounded. One may wonder if, despite being positive, this term is still of at least second order. However, the answer seems negative. By applying the Dynkin's formula (Thm. C.10 and Cor. C.11) to $\mu_s \log \widehat{\mu}_s$ in the term (III.4), we have that the first integral in (C.9) can be expanded as follows*

$$\mathbb{E}\left[\int_{s_n}^{s_{n+1}}\int_{\mathbb{D}}\left(\left(1 - \tfrac{1}{2\theta}\right)\mu_{s_n}(\nu)\log\widehat{\mu}_{s_n}(\nu) + \tfrac{1}{2\theta}\mu_{\rho_n}(\nu)\log\widehat{\mu}_{\rho_n}(\nu)\right)\gamma(\mathrm{d}\nu)\mathrm{d}s\right]$$

$$=\tfrac{1}{2\theta}\int_{s_n}^{s_{n+1}}\int_{\mathbb{D}}\left(\mu_{s_n}(\nu)\log\widehat{\mu}_{s_n}(\nu) + \theta\Delta_n\mathcal{L}\left(\mu_{s_n}(\nu)\log\widehat{\mu}_{s_n}(\nu)\right)\right)\gamma(\mathrm{d}\nu)\mathrm{d}s$$

$$+\left(1 - \tfrac{1}{2\theta}\right)\int_{s_n}^{s_{n+1}}\int_{\mathbb{D}}\mu_{s_n}(\nu)\log\widehat{\mu}_{s_n}(\nu)\gamma(\mathrm{d}\nu)\mathrm{d}s + \mathcal{O}(\Delta_n^2)$$

$$=\Delta_n\int_{\mathbb{D}}\mu_{s_n}(\nu)\log\widehat{\mu}_{s_n}(\nu)\gamma(\mathrm{d}\nu) + \frac{1}{2}\Delta_n^2\int_{\mathbb{D}}\mathcal{L}\left(\mu_{s_n}(\nu)\log\widehat{\mu}_{s_n}(\nu)\right)\gamma(\mathrm{d}\nu) + \mathcal{O}(\Delta_n^3).$$

*Similarly, applying Dynkin's formula to the following function*

$$G_s(\nu, y_{s-}) = \left(\tfrac{1}{2\theta}\mu_s(\nu, y_{s-}) + \left(1 - \tfrac{1}{2\theta}\right)\mu_{s_n}(\nu, y_{s-})\right)\log\left(\tfrac{1}{2\theta}\widehat{\mu}_s(\nu, y_{s-}) + \left(1 - \tfrac{1}{2\theta}\right)\widehat{\mu}_{s_n}(\nu, y_{s-})\right),$$

*with $G_0(\nu, y_{s_n}) = \mu_{s_n}(\nu, y_{s_n})\log\widehat{\mu}_{s_n}(\nu, y_{s_n})$ allows us to expand the second integral in (C.9) as below*

$$\mathbb{E}\left[\int_{s_n}^{s_{n+1}}\int_{\mathbb{D}}\left(\tfrac{1}{2\theta}\mu_{\rho_n}(\nu) + \left(1 - \tfrac{1}{2\theta}\right)\mu_{s_n}(\nu)\right)\log\left(\tfrac{1}{2\theta}\widehat{\mu}_{\rho_n}(\nu) + \left(1 - \tfrac{1}{2\theta}\right)\widehat{\mu}_{s_n}(\nu)\right)\gamma(\mathrm{d}\nu)\mathrm{d}s\right]$$

$$=\Delta_n\int_{\mathbb{D}}G_{s_n}(y_{s_n})\gamma(\mathrm{d}\nu) + \theta\Delta_n^2\int_{\mathbb{D}}\mathcal{L}G_{s_n}(y_{s_n})\gamma(\mathrm{d}\nu) + \mathcal{O}(\Delta_n^3),$$

*where*

$$\mathcal{L}G_{s_n}(\nu, y_{s_n})$$

$$=\tfrac{1}{2\theta}\partial_s\mu_{s_n}(\nu, y_{s_n})\log\widehat{\mu}_{s_n}(\nu, y_{s_n}) + \tfrac{1}{2\theta}\mu_{s_n}(\nu, y_{s_n})\tfrac{1}{2\theta}\frac{\partial_s\widehat{\mu}_{s_n}(\nu, y_{s_n})}{\widehat{\mu}_{s_n}(\nu, y_{s_n})}$$

$$+\tfrac{1}{2\theta}\int_{\mathbb{D}}\mu_{s_n}(\nu, y_{s_n}+\nu')\log\left(\tfrac{1}{2\theta}\widehat{\mu}_s(\nu, y_{s_n}+\nu') + \left(1-\tfrac{1}{2\theta}\right)\widehat{\mu}_{s_n}(\nu, y_{s_n}+\nu')\right)\gamma(\mathrm{d}\nu')$$

$$-\tfrac{1}{2\theta}\int_{\mathbb{D}}\mu_{s_n}(\nu, y_{s_n})\log\widehat{\mu}_{s_n}(\nu, y_{s_n})\gamma(\mathrm{d}\nu')$$

$$+\left(1-\tfrac{1}{2\theta}\right)\mu_{s_n}(\nu, y_{s_n})\tfrac{1}{2\theta}\frac{\partial_s\widehat{\mu}_{s_n}(\nu, y_{s_n})}{\widehat{\mu}_{s_n}(\nu, y_{s_n})}$$

$$+\left(1-\tfrac{1}{2\theta}\right)\int_{\mathbb{D}}\mu_{s_n}(\nu, y_{s_n}+\nu')\log\left(\tfrac{1}{2\theta}\widehat{\mu}_s(\nu, y_{s_n}+\nu') + \left(1-\tfrac{1}{2\theta}\right)\widehat{\mu}_{s_n}(\nu, y_{s_n}+\nu')\right)\gamma(\mathrm{d}\nu')$$

$$-\left(1-\tfrac{1}{2\theta}\right)\int_{\mathbb{D}}\mu_{s_n}(\nu, y_{s_n})\log\widehat{\mu}_{s_n}(\nu, y_{s_n})\gamma(\mathrm{d}\nu')$$

$$=\tfrac{1}{2\theta}\partial_s\mu_{s_n}(\nu, y_{s_n})\log\widehat{\mu}_{s_n}(\nu, y_{s_n}) + \tfrac{1}{2\theta}\mu_{s_n}(\nu, y_{s_n})\frac{\partial_s\widehat{\mu}_{s_n}(\nu, y_{s_n})}{\widehat{\mu}_{s_n}(\nu, y_{s_n})}$$

$$+\tfrac{1}{2\theta}\int_{\mathbb{D}}\mu_{s_n}(\nu, y_{s_n}+\nu')\log\widehat{\mu}_s(\nu, y_{s_n}+\nu')\gamma(\mathrm{d}\nu')$$

$$+\left(1-\tfrac{1}{2\theta}\right)\int_{\mathbb{D}}\mu_{s_n}(\nu, y_{s_n}+\nu')\log\widehat{\mu}_s(\nu, y_{s_n}+\nu')\gamma(\mathrm{d}\nu')$$

$$-\tfrac{1}{2\theta}\int_{\mathbb{D}}\mu_{s_n}(\nu, y_{s_n})\log\widehat{\mu}_{s_n}(\nu, y_{s_n})\gamma(\mathrm{d}\nu') - \left(1-\tfrac{1}{2\theta}\right)\int_{\mathbb{D}}\mu_{s_n}(\nu, y_{s_n})\log\widehat{\mu}_{s_n}(\nu, y_{s_n})\gamma(\mathrm{d}\nu').$$

*This further implies that*

$$\theta\mathcal{L}G_{s_n}(y_{s_n}) = \frac{1}{2}\mathcal{L}\left(\mu_{s_n}(\nu)\log\widehat{\mu}_{s_n}(\nu)\right)$$

$$+ \tfrac{1}{2\theta}\int_{\mathbb{D}}\left(\mu_{s_n}(\nu, y_{s_n}+\nu')\log\widehat{\mu}_s(\nu, y_{s_n}+\nu') - \mu_{s_n}(\nu, y_{s_n})\log\widehat{\mu}_{s_n}(\nu, y_{s_n})\right)\gamma(\mathrm{d}\nu').$$

*Comparing the first and second order terms in the two expansions of the two integrals in (C.9) above then implies that the term* (III.4) *is of at most second order.*

## C.4  Lemmas and Propositions

In this section, we provide the detailed proofs of the lemmas and propositions omitted in the proof of Thms. 5.4 and 5.5.

**Error due to the Intensity Estimation.**  Apart from the terms (I.1) and (II.1) in the proof of Thm. 5.4 and the term (III.1) in the proof of Thm. 5.5, we also need to bound the error terms (II.4) in terms of the intensity estimation error, which is given by the following proposition. Notably, the following bound also utilizes the convexity of the loss function and the extrapolation nature of the second step in the $\theta$-Trapezoidal method (C.6).

**Proposition C.9.** *For the interval $(s_n, s_{n+1}]$ for $n \in [0 : N-1]$, we have the following error bound:*

$$\mathbb{E}\left[(\text{II.4})\right] = \mathbb{E}\Bigg[\int_{\rho_n}^{s_{n+1}}\int_{\mathbb{D}}\left(\alpha_1\mu_{\rho_n}(\nu)\log\widehat{\mu}_{\rho_n}(\nu) - \alpha_2\mu_{s_n}(\nu)\log\widehat{\mu}_{s_n}(\nu)\right)\gamma(\mathrm{d}\nu)\mathrm{d}s$$

$$- \int_{\rho_n}^{s_{n+1}}\int_{\mathbb{D}}\left(\alpha_1\mu_{\rho_n}(\nu) - \alpha_2\mu_{s_n}(\nu)\right)\log\left(\alpha_1\widehat{\mu}_{\rho_n}(\nu) - \alpha_2\widehat{\mu}_{s_n}(\nu)\right)\gamma(\mathrm{d}\nu)\mathrm{d}s\Bigg] \quad \text{(C.11)}$$

$$\lesssim \Delta_n\epsilon_{\text{II}}.$$

*Proof.* We first define and bound three error terms (II.4.1), (II.4.2), and (II.4.3) with score estimation error (Assump. 5.3) as follows:

$$
\begin{aligned}
\mathbb{E}\left[|(\text{II.4.1})|\right] =& \mathbb{E}\left[\left|\int_{\rho_n}^{s_{n+1}}\int_{\mathbb{D}} \alpha_1\left(\mu_{\rho_n}(\nu)\log\widehat{\mu}_{\rho_n}(\nu) - \widehat{\mu}_{\rho_n}(\nu)\log\widehat{\mu}_{\rho_n}(\nu)\right)\gamma(\mathrm{d}\nu)\mathrm{d}s\right|\right] \\
\leq& \alpha_1\mathbb{E}\left[\int_{\rho_n}^{s_{n+1}}\int_{\mathbb{D}} \left|\mu_{\rho_n}(\nu) - \widehat{\mu}_{\rho_n}(\nu)\right|\left|\log\widehat{\mu}_{\rho_n}(\nu)\right|\gamma(\mathrm{d}\nu)\mathrm{d}s\right] \\
\lesssim& \mathbb{E}\left[\int_{\rho_n}^{s_{n+1}}\int_{\mathbb{D}} \left|\mu_{\rho_n}(\nu) - \widehat{\mu}_{\rho_n}(\nu)\right|\gamma(\mathrm{d}\nu)\mathrm{d}s\right] \lesssim \Delta_n\epsilon_{\mathrm{II}},
\end{aligned}
$$

Similarly, we also have

$$
\mathbb{E}\left[|(\text{II.4.2})|\right] = \mathbb{E}\left[\left|\int_{\rho_n}^{s_{n+1}}\int_{\mathbb{D}} \alpha_2\left(\mu_{s_n}(\nu)\log\widehat{\mu}_{s_n}(\nu) - \widehat{\mu}_{s_n}(\nu)\log\widehat{\mu}_{s_n}(\nu)\right)\gamma(\mathrm{d}\nu)\mathrm{d}s\right|\right] \lesssim \Delta_n\epsilon_{\mathrm{II}},
$$

and

$$
\begin{aligned}
\mathbb{E}\left[|(\text{II.4.3})|\right] =& \mathbb{E}\left[\left|\int_{\rho_n}^{s_{n+1}}\int_{\mathbb{D}} \left(\alpha_1\mu_{\rho_n}(\nu) - \alpha_2\mu_{s_n}(\nu)\right)\log\left(\alpha_1\widehat{\mu}_{\rho_n}(\nu) - \alpha_2\widehat{\mu}_{s_n}(\nu)\right)\gamma(\mathrm{d}\nu)\mathrm{d}s\right.\right. \\
&\left.\left.- \int_{\rho_n}^{s_{n+1}}\int_{\mathbb{D}} \left(\alpha_1\widehat{\mu}_{\rho_n}(\nu) - \alpha_2\widehat{\mu}_{s_n}(\nu)\right)\log\left(\alpha_1\widehat{\mu}_{\rho_n}(\nu) - \alpha_2\widehat{\mu}_{s_n}(\nu)\right)\gamma(\mathrm{d}\nu)\mathrm{d}s\right|\right] \\
\lesssim& \Delta_n\epsilon_{\mathrm{II}}.
\end{aligned}
$$

The remaining term $(\text{II.4.4}) = (\text{II.4}) - (\text{II.4.1}) - (\text{II.4.2}) - (\text{II.4.3})$ is then given by

$$
\begin{aligned}
(\text{II.4.4}) =& \int_{\rho_n}^{s_{n+1}}\int_{\mathbb{D}} \left(\alpha_1\widehat{\mu}_{\rho_n}(\nu)\log\widehat{\mu}_{\rho_n}(\nu) - \alpha_2\widehat{\mu}_{s_n}(\nu)\log\widehat{\mu}_{s_n}(\nu)\right)\gamma(\mathrm{d}\nu)\mathrm{d}s \\
&- \int_{\rho_n}^{s_{n+1}}\int_{\mathbb{D}} \left(\alpha_1\widehat{\mu}_{\rho_n}(\nu) - \alpha_2\widehat{\mu}_{s_n}(\nu)\right)\log\left(\alpha_1\widehat{\mu}_{\rho_n}(\nu) - \alpha_2\widehat{\mu}_{s_n}(\nu)\right)\gamma(\mathrm{d}\nu)\mathrm{d}s \leq 0,
\end{aligned}
$$

where the last inequality follows from Jensen's inequality, *i.e.,*

$$
\alpha_1 x\log x - \alpha_2 y\log y \leq (\alpha_1 x - \alpha_2 y)\log(\alpha_1 x - \alpha_2 y),
$$

for $\alpha_1, \alpha_2 \geq 0$ and $\alpha_1 - \alpha_2 = 1$. Therefore, by summing up the terms above, we have

$$
\mathbb{E}\left[(\text{II.4})\right] \leq \mathbb{E}\left[(\text{II.4.1}) + (\text{II.4.2}) + (\text{II.4.3}) + (\text{II.4.4})\right] \lesssim \Delta_n\epsilon_{\mathrm{II}},
$$

and the proof is complete. $\qquad\square$

**Error Related to the Smoothness of Intensity.** Below we first present the Dynkin's formula, which is the most essential tool for the proof of the error related to the smoothness of the intensity.

**Theorem C.10** (Dynkin's Formula). *Let $(y_t)_{t\in[0,\tau]}$ be the following process:*

$$
y_t = y_0 + \int_0^t\int_{\mathbb{D}} \nu N[\mu](\mathrm{d}s, \mathrm{d}\nu),
$$

*where $N[\mu](\mathrm{d}s, \mathrm{d}\nu)$ is a Poisson random measure with intensity $\mu$ of the form $\mu_s(\nu, y_{s^-})$. For any $f \in C^1([0,\tau]\times\mathbb{X})$, we define the generator of the process $(y_t)_{t\in[0,\tau]}$ as below*

$$
\mathcal{L}f_t(y) = \lim_{\tau\to 0^+}\left[\left.\frac{f_{t+\tau}(y_{t+\tau}) - f_t(y_t)}{\tau}\right|y_t = y\right] = \partial_t f_t(y) + \int_{\mathbb{D}}\left(f_t(y+\nu) - f_t(y)\right)\mu_t(\nu, y)\gamma(\mathrm{d}\nu).
\tag{C.12}
$$

*Then we have that*

$$
\mathbb{E}\left[f_t(y_t)\right] = f_0(y_0) + \mathbb{E}\left[\int_0^t \mathcal{L}f_s(y_s)\mathrm{d}s\right].
$$

*Proof.* The definition and the form of the generator $\mathcal{L}$, as well as the Dynkin's formula are all well-known in the literature of jump processes. We refer readers to detailed discussions on these topics in [244].

Here we take $X(t) = (t, y_t)$, $z = (\nu, \xi)$, $\alpha(t, X(t)) = 0$, $\sigma(t, X(t)) = 0$, $\gamma(t, X(t^-), z) = \nu \mathbf{1}_{0 \le \xi \le \mu_t(\nu, y_{t^-})}$ in the statement of Thm. 1.19 in [244] and replace the compensated Poisson random measure $\widetilde{N}(dt, dz)$ with the Poisson random measure $N(ds, d\nu, d\xi)$ defined as Rmk. B.3. Then we are allowed to use the ordinary Poisson random measure instead of the compensated one since we are working with a finite measure $\gamma(d\nu)$.

From Thm. 1.22 in [244], we have that

$$\mathcal{L}f_t(y) = \partial_t f_t(y) + \int_{\mathbb{D}} \int_{\mathbb{R}} \left( f_t(y + \nu \mathbf{1}_{0 \le \xi \le \mu_t(\nu, y)}) - f_t(y) \right) \gamma(d\nu) d\xi$$

$$= \partial_t f_t(y) + \int_{\mathbb{D}} \left( f_t(y + \nu) - f_t(y) \right) \mu_t(\nu, y) \gamma(d\nu),$$

and the proof is complete. $\qquad\qquad\square$

In many cases below, we will need the following first-order expansion of the expectation of the function $f_t(y_t)$ by assuming the second-order smoothness of the function $f$.

**Corollary C.11.** *Suppose that the process $(y_t)_{t \in [0, \tau]}$ and the generator $\mathcal{L}$ are defined as in Thm. C.10. If we further assume that $f \in C^2([0, \tau] \times \mathbb{X})$, then it holds that*

$$\mathbb{E}\left[ f_t(y_t) \right] = f_0(y_0) + t\mathcal{L}f_0(y_0) + \mathcal{O}(t^2).$$

*Proof.* We expand the function $f_s(y_s)$ from $t = 0$ as follows

$$\mathbb{E}\left[ f_t(y_t) \right] = f_0(y_0) + \mathbb{E}\left[ \int_0^t \mathcal{L}f_s(y_s) ds \right]$$

$$= f_0(y_0) + \mathbb{E}\left[ \int_0^t \mathcal{L}\left( f_0(y_0) + \int_0^s \mathcal{L}f_\sigma(y_\sigma) d\sigma \right) ds \right]$$

$$= f_0(y_0) + \mathcal{L}f_0(y_0)t + \mathbb{E}\left[ \int_0^t \int_0^s \mathcal{L}^2 f_\sigma(y_\sigma) d\sigma ds \right],$$

where $\mathcal{L}^2$ is the second-order generator of the process $(y_t)_{t \in [0, \tau]}$ defined as follows

$$\mathcal{L}^2 f_\sigma(y) = \mathcal{L}\left( \partial_\sigma f_\sigma(y) + \int_{\mathbb{D}} \left( f_\sigma(y + \nu) - f_\sigma(y) \right) \mu_\sigma(\nu) \gamma(d\nu) \right)$$

$$= \partial_\sigma^2 f_\sigma(y) + 2 \int_{\mathbb{D}} \left( \partial_\sigma f_\sigma(y + \nu) - \partial_\sigma f_\sigma(y) \right) \mu_\sigma(\nu) \gamma(d\nu)$$

$$+ \int_{\mathbb{D}} \left( f_\sigma(y + \nu) - f_\sigma(y) \right) \partial_\sigma \mu_\sigma(\nu) \gamma(d\nu)$$

$$+ \int_{\mathbb{D}} \int_{\mathbb{D}} \left( f_\sigma(y + \nu + \nu') - f_\sigma(y + \nu') - f_\sigma(y + \nu) + f_\sigma(y) \right) \mu_\sigma(\nu) \mu_\sigma(\nu') \gamma(d\nu) \gamma(d\nu'),$$

which is bounded uniformly by a constant based on the assumption on the smoothness of the function $f$ up to the second order and the boundedness of the measure $\gamma(d\nu)$. Therefore, the second-order term above is of magnitude $\mathcal{O}(t^2)$, and the proof is complete. $\qquad\square$

The following lemma provides a general recipe for bounding a combination of errors, which resembles standard analysis performed for numerical quadratures. In fact, the following lemma can be easily proved by Taylor expansion when the process $(y_t)_{t \in [0, \tau]}$ is constant, *i.e.*, $y_t \equiv y$. Cor. C.11 offers an analogous approach to perform the expansion when the process $(y_t)_{t \in [0, \tau]}$ is not constant.

**Lemma C.12.** *For any function* $f \in C^2([0,\tau] \times \mathbb{X})$ *and the true backward process* $(y_t)_{t \in [0,\tau]}$ *defined in (2.4), it holds that*

$$\left| \mathbb{E}\left[ \int_0^{\theta\tau} f_0(y_0)\mathrm{d}s + \int_{\theta\tau}^\tau \left( \alpha_1 f_{\theta\tau}(y_{\theta\tau}) - \alpha_2 f_0(y_0) \right)\mathrm{d}s - \int_0^\tau f_s(y_s)\mathrm{d}s \right] \right| \lesssim \tau^3.$$

*Proof.* Let $\mathcal{L}$ be the generator defined in Thm. C.10. By applying the Dynkin's formula (Thm. C.10 and Cor. C.11) to the function $f_t(y_t)$ and plugging in the expression of the generator $\mathcal{L}$, we have that

$$\mathbb{E}\left[ \int_0^{\theta\tau} f_0(y_0)\mathrm{d}s - \alpha_2 \int_{\theta\tau}^\tau f_0(y_0)\mathrm{d}s + \alpha_1 \int_{\theta\tau}^\tau f_{\theta\tau}(y_{\theta\tau})\mathrm{d}s - \int_0^\tau f_s(y_s)\mathrm{d}s \right]$$

$$= \theta\tau f_0(y_0) - \alpha_2(1-\theta)\tau f_0(y_0) + \alpha_1(1-\theta)\tau \left( f_0(y_0) + \theta\tau \mathcal{L}f_0(y_0) \right)$$

$$- \int_0^\tau \left( f_0(y_0) + s\mathcal{L}f_0(y_0) \right)\mathrm{d}s + \mathcal{O}(\tau^3)$$

$$= \left( \theta - \alpha_2(1-\theta) + \alpha_1(1-\theta) - 1 \right)\tau f_0(y_0) + \alpha_1(1-\theta)\theta\tau^2 \mathcal{L}f_0(y_0) - \frac{\tau^2}{2}\mathcal{L}f_0(y_0) + \mathcal{O}(\tau^3),$$

which is of the order $\mathcal{O}(\tau^3)$ by noticing that

$$\theta - \alpha_2(1-\theta) + \alpha_1(1-\theta) - 1 = \left( \frac{1}{2\theta(1-\theta)} - \frac{\theta^2 + (1-\theta)^2}{2\theta(1-\theta)} \right)(1-\theta) - (1-\theta) = 0$$

$$\alpha_1(1-\theta)\theta - \frac{1}{2} = \frac{1}{2\theta(1-\theta)}(1-\theta)\theta - \frac{1}{2} = 0,$$

and the proof is complete. $\qquad\square$

We remark that in Thm. C.10, Cor. C.11, and Lem. C.12, the smoothness of the function $f$ implies that its derivatives up to the relevant order are bounded by constants independent of the time step $\tau$. This condition is verified in the subsequent proofs.

Then we are ready to bound some of the error terms in the proof of Thm. 5.4 with Lem. C.12.

**Corollary C.13.** *For the interval* $(s_n, s_{n+1}]$ *for* $n \in [0 : N-1]$, *we have the following error bound:*

$$|\mathbb{E}\left[ (\mathrm{I}.2) + (\mathrm{II}.2) \right]|$$

$$= \left| \mathbb{E}\left[ \int_{s_n}^{s_{n+1}} \int_{\mathbb{D}} \left( \mu_s(\nu)\log\mu_s(\nu) - \mu_s(\nu) \right)\gamma(\mathrm{d}\nu)\mathrm{d}s \right. \right.$$

$$- \int_{s_n}^{\rho_n} \int_{\mathbb{D}} \left( \mu_{s_n}(\nu)\log\mu_{s_n}(\nu) + \mu_{s_n}(\nu) \right)\gamma(\mathrm{d}\nu)\mathrm{d}s$$

$$\left. \left. - \int_{\rho_n}^{s_{n+1}} \int_{\mathbb{D}} \left( \alpha_1(\mu_{\rho_n}(\nu)\log\mu_{\rho_n}(\nu) - \mu_{\rho_n}(\nu)) - \alpha_2(\mu_{s_n}(\nu)\log\mu_{s_n}(\nu) - \mu_{s_n}(\nu)) \right)\gamma(\mathrm{d}\nu)\mathrm{d}s \right] \right|$$

$$\lesssim \Delta_n^3.$$

*Proof.* The bound is obtained by applying Lem. C.12 with $f$ being the function

$$f_s(y_s) = \int_{\mathbb{D}} \mu_s(\nu)\log\mu_s(\nu)\gamma(\mathrm{d}\nu),$$

Strictly speaking, $f_s(y_s)$ is actually in the form of $f_s(y_{s^-})$, but the argument can be easily extended to this case by assuming time continuity of the function $f$. $\qquad\square$

**Corollary C.14.** *For the interval* $(s_n, s_{n+1}]$ *for* $n \in [0 : N-1]$, *we have the following error bound:*

$$|\mathbb{E}\left[ (\mathrm{I}.4) + (\mathrm{II}.6) \right]|$$

$$= \left| \mathbb{E}\left[ \int_{s_n}^{\rho_n} \int_{\mathbb{D}} \mu_{s_n}(\nu)\log\left( \alpha_1\widehat{\mu}_{\rho_n}^*(\nu) - \alpha_2\widehat{\mu}_{s_n}(\nu) \right)\gamma(\mathrm{d}\nu)\mathrm{d}s \right. \right.$$

$$+ \int_{\rho_n}^{s_{n+1}} \int_{\mathbb{D}} \left( \alpha_1\mu_{\rho_n}(\nu) - \alpha_2\mu_{s_n}(\nu) \right)\log\left( \alpha_1\widehat{\mu}_{\rho_n}^*(\nu) - \alpha_2\widehat{\mu}_{s_n}(\nu) \right)\gamma(\mathrm{d}\nu)\mathrm{d}s$$

$$\left. \left. - \int_{s_n}^{s_{n+1}} \int_{\mathbb{D}} \mu_s(\nu)\log\left( \alpha_1\widehat{\mu}_{\rho_n}^*(\nu) - \alpha_2\widehat{\mu}_{s_n}(\nu) \right)\gamma(\mathrm{d}\nu)\mathrm{d}s \right] \right| \lesssim \Delta_n^3.$$

*Proof.* Note that the intermediate process $y_s^*$ defined in (C.1) is driven by a Poisson random measure that is independent of the Poisson random measure driving the process $y_s$ within the interval $(s_n, s_{n+1}]$. Therefore, the error bound is obtained by

(1) Taking the expectation w.r.t. the intermediate process $y_s^*$ and thus the intermediate intensity $\widehat{\mu}_s^*$, and

(2) Then applying Lem. C.12 with $f$ being the following function

$$f_s(y_s) = \int_{\mathbb{D}} \mu_s(\nu)\mathbb{E}\left[\log\left(\alpha_1\widehat{\mu}_{\rho_n}^*(\nu) - \alpha_2\widehat{\mu}_{s_n}(\nu)\right)\right]\gamma(\mathrm{d}\nu).$$

The result follows directly. $\qquad\square$

Now we turn to the error term (III.6) in Thm. 5.5, for which we need the following variant of Lem. C.12.

**Lemma C.15.** *For any function $f \in C^2([0, \tau] \times \mathbb{X})$ and the true backward process $(y_t)_{t\in[0,\tau]}$ defined in (2.4), it holds that*

$$\left|\mathbb{E}\left[\int_0^\tau \left(\left(1 - \tfrac{1}{2\theta}\right)f_0(y_0) + \tfrac{1}{2\theta}f_{\theta\tau}(y_{\theta\tau})\right)\mathrm{d}s - \int_0^\tau f_s(y_s)\mathrm{d}s\right]\right| \lesssim \tau^3.$$

*Proof.* The proof is similar to that of Lem. C.12. Specifically, we let $\mathcal{L}$ be the generator defined in Thm. C.10, apply the Dynkin's formula (Thm. C.10 and Cor. C.11) to the function $f_t(y_t)$ and plug in the expression of the generator $\mathcal{L}$, which yields

$$\mathbb{E}\left[\int_0^\tau \left(\left(1 - \tfrac{1}{2\theta}\right)f_0(y_0) + \tfrac{1}{2\theta}f_{\theta\tau}(y_{\theta\tau})\right)\mathrm{d}s - \int_0^\tau f_s(y_s)\mathrm{d}s\right]$$

$$= \left(1 - \tfrac{1}{2\theta}\right)\tau f_0(y_0) + \tfrac{1}{2\theta}\int_0^\tau \left(f_0(y_0) + \theta\tau\mathcal{L}f_0(y_0)\right)\mathrm{d}s - \int_0^\tau \left(f_0(y_0) + s\mathcal{L}f_0(y_0)\right)\mathrm{d}s + \mathcal{O}(\tau^3)$$

$$= \mathcal{O}(\tau^3),$$

as desired. $\qquad\square$

**Corollary C.16.** *For the interval $(s_n, s_{n+1}]$ for $n \in [0 : N-1]$, we have the following error bound:*

$$|\mathbb{E}\left[(\text{III.2})\right]|$$

$$= \left|\mathbb{E}\left[\int_{s_n}^{s_{n+1}}\int_{\mathbb{D}}\left(\mu_s(\nu)\log\mu_s(\nu) - \mu_s(\nu)\right)\gamma(\mathrm{d}\nu)\mathrm{d}s\right.\right.$$

$$\left.\left. - \int_{s_n}^{s_{n+1}}\int_{\mathbb{D}}\left(\left(1 - \tfrac{1}{2\theta}\right)\left(\mu_{s_n}(\nu)\log\mu_{s_n}(\nu) - \mu_{s_n}\right) + \tfrac{1}{2\theta}\left(\mu_{\rho_n}(\nu)\log\mu_{\rho_n}(\nu) - \mu_{\rho_n}(\nu)\right)\right)\gamma(\mathrm{d}\nu)\mathrm{d}s\right]\right|$$

$$\lesssim \Delta_n^3.$$

*Proof.* By applying Lem. C.15 with $f$ being the function

$$f_s(y_s) = \int_{\mathbb{D}}\mu_s(\nu)\log\mu_s(\nu)\gamma(\mathrm{d}\nu),$$

we have that the result follows directly. $\qquad\square$

**Corollary C.17.** *For any $n \in [0 : N-1]$ and the corresponding interval $(s_n, s_{n+1}]$, we have the following error bound:*

$$|\mathbb{E}\left[(\text{III.6})\right]|$$

$$= \left|\mathbb{E}\left[\int_{s_n}^{s_{n+1}}\int_{\mathbb{D}}\left(\left(1 - \tfrac{1}{2\theta}\right)\mu_{s_n}(\nu) + \tfrac{1}{2\theta}\mu_{\rho_n}(\nu)\right)\log\left(\left(1 - \tfrac{1}{2\theta}\right)\widehat{\mu}_{s_n}(\nu) + \tfrac{1}{2\theta}\widehat{\mu}_{\rho_n}^*(\nu)\right)\gamma(\mathrm{d}\nu)\mathrm{d}s\right.\right.$$

$$\left.\left. - \int_{s_n}^{s_{n+1}}\int_{\mathbb{D}}\mu_s(\nu)\log\left(\left(1 - \tfrac{1}{2\theta}\right)\widehat{\mu}_{s_n}(\nu) + \tfrac{1}{2\theta}\widehat{\mu}_{\rho_n}^*(\nu)\right)\gamma(\mathrm{d}\nu)\mathrm{d}s\right]\right| \lesssim \Delta_n^3.$$

*Proof.* Following the arguments in the proof of Cor. C.14, the error bound is obtained by first taking the expectation w.r.t. the intermediate process $y_s^*$ and thus the intermediate intensity $\widehat{\mu}_s^*$, and then applying Lem. C.15 with $f$ being the function

$$f_s(y_s) = \int_{\mathbb{D}} \mu_s(\nu) \mathbb{E}\left[\log\left(\left(1 - \tfrac{1}{2\theta}\right)\widehat{\mu}_{s_n}(\nu) + \tfrac{1}{2\theta}\widehat{\mu}_{\rho_n}^*(\nu)\right)\right]\gamma(\mathrm{d}\nu),$$

as desired. $\qquad\square$

**Error involving the Intermediate Process.**

**Proposition C.18.** *For the interval $(s_n, s_{n+1}]$ with $n \in [0 : N-1]$, we have the following error bound:*

$$\mathbb{E}\left[(\mathrm{II.3})\right] = \mathbb{E}\left[\int_{\rho_n}^{s_{n+1}} \int_{\mathbb{D}}\left(\widehat{\mu}_{\rho_n}^*(\nu) - \widehat{\mu}_{\rho_n}(\nu)\right)\gamma(\mathrm{d}\nu)\mathrm{d}s\right] \lesssim \Delta_n^3 + \Delta_n^2 \epsilon_{\mathrm{II}}.$$

*Proof.* First, we rewrite the error term (II.3) as

$$\begin{aligned}
\mathbb{E}\left[(\mathrm{II.3})\right] &= \mathbb{E}\left[\int_{\rho_n}^{s_{n+1}}\int_{\mathbb{D}}\left(\widehat{\mu}_{\rho_n}^*(\nu) - \widehat{\mu}_{\rho_n}(\nu)\right)\gamma(\mathrm{d}\nu)\mathrm{d}s\right] \\
&\lesssim \int_{\rho_n}^{s_{n+1}}\int_{\mathbb{D}}\left(\mathbb{E}\left[\widehat{\mu}_{\rho_n}^*(\nu)\right] - \mathbb{E}\left[\widehat{\mu}_{\rho_n}(\nu)\right]\right)\gamma(\mathrm{d}\nu)\mathrm{d}s.
\end{aligned} \tag{C.13}$$

Then we expand the integrand by applying the Dynkin's formula (Thm. C.10 and Cor. C.11) to the function $\widehat{\mu}_s(\nu)$ w.r.t. the intermediate process $(y_s^*)_{s\in[s_n,\rho_n]}$ and the process $(y_s)_{s\in[s_n,\rho_n]}$ respectively as follows

$$\begin{aligned}
&\mathbb{E}\left[\widehat{\mu}_{\rho_n}^*(\nu)\right] - \mathbb{E}\left[\widehat{\mu}_{\rho_n}(\nu)\right] \\
&= \mathbb{E}\left[\widehat{\mu}_{s_n}(\nu) + \mathcal{L}^*\widehat{\mu}_{s_n}(\nu)\Delta_n + \mathcal{O}(\Delta_n^2)\right] - \mathbb{E}\left[\widehat{\mu}_{s_n}(\nu) + \mathcal{L}\widehat{\mu}_{s_n}(\nu)\Delta_n + \mathcal{O}(\Delta_n^2)\right] \\
&= \mathbb{E}\left[(\mathcal{L}^* - \mathcal{L})\widehat{\mu}_{s_n}(\nu)\Delta_n\right] + \mathcal{O}(\Delta_n^2),
\end{aligned}$$

where the generators $\mathcal{L}^*$ and $\mathcal{L}$ are defined as in (C.12) w.r.t. the processes $(y_s^*)_{s\in[s_n,\rho_n]}$ and $(y_s)_{s\in[s_n,\rho_n]}$, respectively, *i.e.*, for any function $f \in C^1([s_n,\rho_n]\times\mathbb{X})$, we have

$$\begin{aligned}
\mathcal{L}^* f_s(y) &= \partial_s f_s(y) + \int_{\mathbb{D}}\left(f_s(y+\nu) - f_s(y)\right)\widehat{\mu}_{s_n}(\nu)\gamma(\mathrm{d}\nu), \\
\mathcal{L} f_s(y) &= \partial_s f_s(y) + \int_{\mathbb{D}}\left(f_s(y+\nu) - f_s(y)\right)\mu_{s_n}(\nu)\gamma(\mathrm{d}\nu).
\end{aligned} \tag{C.14}$$

Therefore, for the term $\mathbb{E}\left[|(\mathcal{L}^* - \mathcal{L})\widehat{\mu}_{s_n}(\nu)|\right]$ evaluated at $s = s_n$, we have

$$\begin{aligned}
\mathbb{E}\left[|(\mathcal{L}^* - \mathcal{L})\widehat{\mu}_{s_n}(\nu)|\right] &= \mathbb{E}\left[\left|\int_{\mathbb{D}}\left(\widehat{\mu}_{s_n}(y+\nu) - \widehat{\mu}_{s_n}(y)\right)\left(\widehat{\mu}_{s_n}(\nu) - \mu_{s_n}(\nu)\right)\gamma(\mathrm{d}\nu)\right|\right] \\
&\lesssim \mathbb{E}\left[\int_{\mathbb{D}}|\widehat{\mu}_{s_n}(\nu) - \mu_{s_n}(\nu)|\gamma(\mathrm{d}\nu)\right] \lesssim \epsilon_{\mathrm{II}},
\end{aligned} \tag{C.15}$$

where we used the assumption on the estimation error (Assump. 5.3) in the last inequality. Then we can further reduce (C.13) to

$$\int_{\rho_n}^{s_{n+1}}\int_{\mathbb{D}}\left(\mathbb{E}\left[\widehat{\mu}_{\rho_n}^*(\nu)\right] - \mathbb{E}\left[\widehat{\mu}_{\rho_n}(\nu)\right]\right)\gamma(\mathrm{d}\nu)\mathrm{d}s \lesssim \int_{\rho_n}^{s_{n+1}}\left(\epsilon_{\mathrm{II}}\Delta_n + \mathcal{O}(\Delta_n^2)\right)\mathrm{d}s \lesssim \epsilon_{\mathrm{II}}\Delta_n^2 + \Delta_n^3,$$

and the proof is complete. $\qquad\square$

**Corollary C.19.** *For the interval $(s_n, s_{n+1}]$ for $n \in [0 : N-1]$, we have the following error bound:*

$$\begin{aligned}
\mathbb{E}\left[(\mathrm{II.5})\right] =& \mathbb{E}\Bigg[\int_{\rho_n}^{s_{n+1}}\int_{\mathbb{D}}\left(\alpha_1\mu_{\rho_n}(\nu) - \alpha_2\mu_{s_n}(\nu)\right)\log\left(\alpha_1\widehat{\mu}_{\rho_n}(\nu) - \alpha_2\widehat{\mu}_{s_n}(\nu)\right)\gamma(\mathrm{d}\nu)\mathrm{d}s \\
&- \int_{\rho_n}^{s_{n+1}}\int_{\mathbb{D}}\left(\alpha_1\mu_{\rho_n}(\nu) - \alpha_2\mu_{s_n}(\nu)\right)\log\left(\alpha_1\widehat{\mu}_{\rho_n}^*(\nu) - \alpha_2\widehat{\mu}_{s_n}(\nu)\right)\gamma(\mathrm{d}\nu)\mathrm{d}s\Bigg] \\
\lesssim& \Delta_n^3 + \Delta_n^2\epsilon_{\mathrm{II}}.
\end{aligned}$$

*Proof.* Since the two integrands in (II.5) only differ by replacing $\widehat{\mu}^*_{\rho_n}(\nu)$ with $\widehat{\mu}_{\rho_n}(\nu)$, we have the following upper bound by using the assumption on the boundedness of the intensities (Assump. 5.2 (II))

$$
\begin{aligned}
&\mathbb{E}\left[(\text{II.5})\right]\\
&\lesssim\mathbb{E}\left[\int_{\rho_n}^{s_{n+1}}\int_{\mathbb{D}}|\alpha_1\mu_{\rho_n}(\nu)-\alpha_2\mu_{s_n}(\nu)|\frac{1}{\alpha_1\widehat{\mu}_{\rho_n}(\nu)-\alpha_2\widehat{\mu}_{s_n}(\nu)}\alpha_1\left|\widehat{\mu}_{\rho_n}(\nu)-\widehat{\mu}^*_{\rho_n}(\nu)\right|\gamma(\mathrm{d}\nu)\mathrm{d}s\right]\\
&\lesssim\mathbb{E}\left[\int_{\rho_n}^{s_{n+1}}\int_{\mathbb{D}}\left|\widehat{\mu}_{\rho_n}(\nu)-\widehat{\mu}^*_{\rho_n}(\nu)\right|\gamma(\mathrm{d}\nu)\mathrm{d}s\right]\\
&\lesssim\Delta_n\mathbb{E}\left[\int_{\mathbb{D}}\left|\widehat{\mu}_{\rho_n}(\nu)-\widehat{\mu}^*_{\rho_n}(\nu)\right|\right]\gamma(\mathrm{d}\nu)
\end{aligned}
\tag{C.16}
$$

Applying the same arguments as in Prop. C.18, which uses the generators $\mathcal{L}$ and $\mathcal{L}^*$ defined in (C.14), we can bound the RHS above as follows

$$
\begin{aligned}
&\mathbb{E}\left[\left|\widehat{\mu}^*_{\rho_n}(\nu)-\widehat{\mu}_{\rho_n}(\nu)\right|\right]\\
&=\mathbb{E}\left[\left|\left(\widehat{\mu}_{s_n}(\nu)+\mathcal{L}^*\widehat{\mu}_{s_n}(\nu)\Delta_n+\mathcal{O}(\Delta_n^2)\right)-\left(\widehat{\mu}_{s_n}(\nu)+\mathcal{L}\widehat{\mu}_{s_n}(\nu)\Delta_n+\mathcal{O}(\Delta_n^2)\right)\right|\right]\\
&\lesssim\Delta_n\mathbb{E}\left[|(\mathcal{L}^*-\mathcal{L})\widehat{\mu}_{s_n}(\nu)|\right]+\mathcal{O}(\Delta_n^2)\lesssim\Delta_n\epsilon_{\text{II}}+\mathcal{O}(\Delta_n^2)
\end{aligned}
\tag{C.17}
$$

where the last inequality follows from (C.15). Substituting (C.17) into (C.16) then yields the desired upper bound. $\square$

**Proposition C.20.** *For the interval $(s_n, s_{n+1}]$ with $n \in [0 : N-1]$, we have the following error bound:*

$$
\begin{aligned}
\mathbb{E}\left[(\text{I.3})\right]=&\mathbb{E}\left[\int_{s_n}^{\rho_n}\int_{\mathbb{D}}(\mu_s(\nu)-\mu_{s_n}(\nu))\left(\log\left(\alpha_1\widehat{\mu}^*_{\rho_n}(\nu)-\alpha_2\widehat{\mu}_{s_n}(\nu)\right)-\log\widehat{\mu}_{s_n}(\nu)\right)\gamma(\mathrm{d}\nu)\mathrm{d}s\right]\\
&\lesssim\Delta_n^3.
\end{aligned}
$$

*Proof.* First, we observe by Dynkin's formula (Thm. C.10) that

$$
\mathbb{E}\left[|\mu_s(\nu)-\mu_{s_n}(\nu)|\right]=\mathbb{E}\left[\left|\int_{s_n}^{s}\mathcal{L}\mu_{s_n}\mathrm{d}s+\mathcal{O}(\Delta_n^2)\right|\right]\lesssim\Delta_n,
$$

and also

$$
\mathbb{E}\left[|\widehat{\mu}_s(\nu)-\widehat{\mu}_{s_n}(\nu)|\right]=\mathbb{E}\left[\left|\int_{s_n}^{s}\mathcal{L}^*\widehat{\mu}_{s_n}\mathrm{d}s+\mathcal{O}(\Delta_n^2)\right|\right]\lesssim\Delta_n.
\tag{C.18}
$$

Secondly, applying the given assumption (Assump. 5.2 (II)) on the boundedness of the intensities yields

$$
\begin{aligned}
&\mathbb{E}\left[\left|\log\left(\alpha_1\widehat{\mu}^*_{\rho_n}(\nu)-\alpha_2\widehat{\mu}_{s_n}(\nu)\right)-\log\widehat{\mu}_{s_n}(\nu)\right|\right]\\
&\lesssim\frac{1}{\widehat{\mu}_{s_n}(\nu)}\mathbb{E}\left[\left|\alpha_1\widehat{\mu}^*_{\rho_n}(\nu)-\alpha_2\widehat{\mu}_{s_n}(\nu)-\widehat{\mu}_{s_n}(\nu)\right|\right]\\
&\lesssim\mathbb{E}\left[\left|\alpha_1\widehat{\mu}^*_{\rho_n}(\nu)-\alpha_2\widehat{\mu}_{s_n}(\nu)-\widehat{\mu}_{s_n}(\nu)\right|\right]\\
&\lesssim\mathbb{E}\left[\alpha_1\left|\widehat{\mu}^*_{\rho_n}(\nu)-\widehat{\mu}_{s_n}(\nu)\right|\right]\\
&\lesssim\mathbb{E}\left[\left|\widehat{\mu}^*_{\rho_n}(\nu)-\widehat{\mu}_{\rho_n}(\nu)\right|\right]+\mathbb{E}\left[\left|\widehat{\mu}_{\rho_n}(\nu)-\widehat{\mu}_{s_n}(\nu)\right|\right]\\
&\lesssim\Delta_n+\Delta_n\epsilon_{\text{II}}+\mathcal{O}(\Delta_n^2)\lesssim\Delta_n
\end{aligned}
\tag{C.19}
$$

where the last inequality follows from (C.17) proved above. Therefore, we may further deduce that

$$
\begin{aligned}
&\mathbb{E}\left[(\text{I.3})\right]\\
&\leq\int_{s_n}^{\rho_n}\int_{\mathbb{D}}\mathbb{E}\left[|\mu_s(\nu)-\mu_{s_n}(\nu)|\right]\\
&\qquad\mathbb{E}\left[\left|\log\left(\alpha_1\widehat{\mu}^*_{\rho_n}(\nu)-\alpha_2\widehat{\mu}_{s_n}(\nu)\right)-\log\left(\alpha_1\widehat{\mu}_{\rho_n}(\nu)-\alpha_2\widehat{\mu}_{s_n}(\nu)\right)\right|\right]\gamma(\mathrm{d}\nu)\mathrm{d}s\\
&\lesssim\Delta_n^3,
\end{aligned}
$$

where the first inequality is due to the independency of $y_s$ and $y_s^*$ for $s \in [s_n, \rho_n]$, and the proof is complete. $\square$

# D Details of Numerical Experiments

In Apps. D.1 to D.4, we present additional numerical results for the 15-dimensional toy model, text generation, image generation, and diffusion LLM, respectively.

## D.1 15-Dimensional Toy Model

We first derive the closed-form formula of the marginal distributions $p_t$ in this model. Recall that the state space $\mathbb{X} = \{1, 2, ..., d\}$ with $d = 15$, and the initial distribution is $p_0 \in \Delta^d$. The rate matrix at any time is $Q = \frac{1}{d}E - I$. By solving (2.1), we see that

$$p_t = e^{tQ}p_0 = \left( \frac{1 - e^{-t}}{d}E + e^{-t}I \right) p_0,$$

and therefore $p_t$ converges to the uniform distribution $p_\infty = \frac{1}{d}\mathbf{1}$ as $t \to \infty$. The formula of $p_t$ directly yields the scores $s_t(x) = \frac{p_t}{p_t(x)}$.

During inference, we initialize at the uniform distribution $q_0 = p_\infty$ and run from time 0 to $T = 12$. The truncation error of this choice of time horizon is of the magnitude of $10^{-12}$ reflected by $D_{\mathrm{KL}}(p_T \| p_\infty)$, and therefore negligible. The discrete time points form an arithmetic sequence.

We generate $10^6$ samples for each algorithm and use `np.bincount` to obtain the empirical distribution $\widehat{q}_T$ as the output distribution. Finally, the KL divergence is computed by

$$D_{\mathrm{KL}}(p_0 \| \widehat{q}_T) = \sum_{i=1}^{d} p_0(i) \log \frac{p_0(i)}{\widehat{q}_T(i)}.$$

We also perform bootstrapping for 1000 times to obtain the 95% confidence interval of the KL divergence, the results are shown by the shaded area in Fig. 2. The fitted lines are obtained by standard linear regression on the log-log scale with the slopes marked beside each line in Fig. 2.

## D.2 Text Generation

For text generation, we use the small version of RADD [33] checkpoint[1] trained with $\lambda$-DCE loss. We choose an early stopping time $\delta = 10^{-3}$ for a stable numerical simulation. Since RADD is a masked discrete diffusion model, we can freely choose the noise schedule $\sigma(t)$ used in the inference process. We consider the following log-linear noise schedule used in the model training,

$$\sigma(t) = \frac{1 - \epsilon}{1 - (1 - \epsilon)t}, \quad \bar{\sigma}(t) = \int_0^t \sigma(s)\mathrm{d}s = -\log(1 - (1 - \epsilon)t) \tag{D.1}$$

where we choose $\epsilon = 10^{-3}$.

The score function $s_\theta(x_t, t)$ used for computing the transition rate matrix can be computed from the RADD score model $p_\theta$ using the following formula from [33],

$$s_t^\theta(x_t) = \frac{e^{-\bar{\sigma}(t)}}{1 - e^{-\bar{\sigma}(t)}} p_\theta(x_t), \tag{D.2}$$

where the model $p_\theta$ is trained to approximate the conditional distribution of the masked positions given all unmasked positions. More specifically, let $d$ be the length of the sequence and $\{1, 2, ..., S\}$ be the vocabulary set (not including the mask token). Then given a partially masked sequence $x = (x^1, ..., x^d)$, the model $p_\theta(x)$ outputs a $d \times S$ matrix whose $(\ell, s)$ element approximates $\mathbb{P}_{X \sim p_{\mathrm{data}}}(x^\ell = s | X^{\mathrm{UM}} = x^{\mathrm{UM}})$ when $x^\ell$ is mask, and is $\mathbf{1}_{X^\ell, s}$ if otherwise. Here, $x^{\mathrm{UM}}$ represents the unmasked portion of the sequence $x$.

We adopt a uniform discretization of the time interval $(\delta, 1]$. For $\theta$-RK-2 and $\theta$-Trapezoidal, we pick $\theta = \frac{1}{2}$. We compare our proposed $\theta$-RK-2 and $\theta$-Trapezoidal with the Euler method, Tweedie $\tau$-leaping, $\tau$-leaping, and we present full results across all NFEs ranging from 16 to 1024 in Tabs. 4 and 5. For each method, we generate 1024 samples and compute the average perplexities and unigram entropy on both GPT-2 large[2] and LLaMA 3.[3] All the experiments are run on a single NVIDIA A100 GPU.

Table 3: Generative perplexity of texts generated by different sampling algorithms on GPT-2 large. Lower values are better, with the best in **bold**.

| Method | NFE = 16 | NFE = 32 | NFE = 64 | NFE = 128 | NFE = 256 | NFE = 512 | NFE = 1024 |
|---|---|---|---|---|---|---|---|
| FHS | $\leq 307.425$ | $\leq 186.594$ | $\leq 141.625$ | $\leq 122.732$ | $\leq 113.310$ | $\leq 113.026$ | $\leq 109.406$ |
| Euler | $\leq 277.962$ | $\leq 160.586$ | $\leq 111.597$ | $\leq 86.276$ | $\leq 68.092$ | $\leq 55.622$ | $\leq 44.686$ |
| Tweedie $\tau$-leaping | $\leq 277.133$ | $\leq 160.248$ | $\leq 110.848$ | $\leq 85.738$ | $\leq 70.102$ | $\leq 55.194$ | $\leq 44.257$ |
| $\tau$-leaping | $\leq 126.835$ | $\leq 96.321$ | $\leq 69.226$ | $\leq 52.366$ | $\leq 41.694$ | $\leq 33.789$ | $\leq 28.797$ |
| Semi-AR | $\leq 2857.469$ | $\leq 1543.302$ | $\leq 741.184$ | $\leq 360.793$ | $\leq 222.303$ | $\leq 164.162$ | $\leq 147.406$ |
| $\theta$-RK-2 | $\leq 127.363$ | $\leq 109.351$ | $\leq 86.102$ | $\leq 64.317$ | $\leq 49.816$ | $\leq 40.375$ | $\leq 33.971$ |
| $\theta$-Trapezoidal | $\leq \mathbf{123.585}$ | $\leq \mathbf{89.912}$ | $\leq \mathbf{66.549}$ | $\leq \mathbf{49.051}$ | $\leq \mathbf{39.959}$ | $\leq \mathbf{32.456}$ | $\leq \mathbf{27.553}$ |

Table 4: Generative perplexity of texts generated by different sampling algorithms on LLaMA 3. Lower values are better, with the best in **bold**.

| Method | NFE = 16 | NFE = 32 | NFE = 64 | NFE = 128 | NFE = 256 | NFE = 512 | NFE = 1024 |
|---|---|---|---|---|---|---|---|
| FHS | $\leq 342.498$ | $\leq 210.742$ | $\leq 155.258$ | $\leq 132.135$ | $\leq 127.526$ | $\leq 123.013$ | $\leq 120.791$ |
| Euler | $\leq 318.413$ | $\leq 175.555$ | $\leq 125.955$ | $\leq 91.051$ | $\leq 75.245$ | $\leq 59.971$ | $\leq 49.406$ |
| Tweedie $\tau$-leaping | $\leq 316.744$ | $\leq 172.941$ | $\leq 121.248$ | $\leq 94.253$ | $\leq 75.403$ | $\leq 59.943$ | $\leq 49.239$ |
| $\tau$-leaping | $\leq 152.867$ | $\leq 117.930$ | $\leq 86.980$ | $\leq 68.090$ | $\leq \mathbf{53.664}$ | $\leq 44.676$ | $\leq 38.293$ |
| Semi-AR | $\leq 2696.883$ | $\leq 1684.973$ | $\leq 829.391$ | $\leq 410.177$ | $\leq 251.963$ | $\leq 166.927$ | $\leq 162.093$ |
| $\theta$-RK-2 | $\leq 150.439$ | $\leq 132.090$ | $\leq 107.066$ | $\leq 80.742$ | $\leq 63.277$ | $\leq 52.563$ | $\leq 44.687$ |
| $\theta$-Trapezoidal | $\leq \mathbf{146.027}$ | $\leq \mathbf{113.260}$ | $\leq \mathbf{83.456}$ | $\leq \mathbf{66.071}$ | $\leq 54.307$ | $\leq \mathbf{44.293}$ | $\leq \mathbf{35.524}$ |

From the results in Tabs. 3 to 5, we observe that $\theta$-Trapezoidal almost outperforms all other approaches and generates samplers with better perplexities across almost all NFEs. We also noticed that both the Euler method and Tweedie $\tau$-leaping perform similarly, and are beaten by a large margin by $\theta$-RK-2 and $\tau$-leaping.

In Tab. 6, we present the percentage of positive extrapolated intensities for different algorithms across NFE values. This partially validates the assumption in our theoretical analysis (Thms. 5.4 and 5.5) that the intensity remains positive throughout the sampling process.

### D.3 Image Generation

For the image generation, we use the checkpoint of MaskGIT [63, 68] reproduced in Pytorch[4]. Recall that the MaskGIT is a masked image model which, given a partially masked sequence, outputs the conditional distributions of the masked positions given the unmasked portion, just like the model $p_\theta(\cdot)$ in the aforementioned masked text model, RADD. Therefore, by similarly introducing a time noise schedule $\sigma(t)$ (for which we adopt the same log-linear schedule (D.1) in our experiment), we obtain a masked discrete diffusion model akin to the RADD. The score function can be computed accordingly using the model output as in (D.2).

We choose an early stopping time $\delta = 10^{-3}$, and adopt a uniform discretization of the time interval $(\delta, 1]$ for $\theta$-RK-2, $\theta$-Trapezoidal, $\tau$-leaping and the Euler method. For parallel decoding, we use a linear randomization strategy in the re-masking step and an $\arccos$ masking scheduler, as recommended in [63]. For each method, we generate 50k samples in a class-conditioned way and compute its FID against the validation split of ImageNet. We use classifier-free guidance to enhance generation quality and set the guidance strength to $w = 3$.

We present the full results for NFE ranging from 4 to 64 in Fig. 5. All the experiments are run on 1 NVIDIA A100. Notably, $\theta$-Trapezoidal with $\theta = \frac{1}{3}$ is the best-performing method except for extremely low NFE budgets. While $\theta$-Trapezoidal with $\theta = \frac{1}{2}$ in general demonstrates a less competitive performance, it converges to the same generation quality as $\theta = \frac{1}{3}$ in the high NFE regime. We also noticed that when using extrapolation with $\theta = \frac{1}{3}$, $\theta$-RK-2 beats $\tau$-leaping for NFE larger than 8, which again accords with our theoretical prediction of its competitive performance in $\theta \in (0, \frac{1}{2}]$ regime.

---

[1]`https://huggingface.co/JingyangOu/radd-lambda-dce`

[2]`https://huggingface.co/openai-community/gpt2-large`

[3]`https://huggingface.co/meta-llama/Meta-Llama-3-8B`

[4]`https://github.com/valeoai/Maskgit-pytorch`

Table 5: Unigram entropy of texts generated by different sampling algorithms on LLaMA 3.

| Method | NFE = 16 | NFE = 32 | NFE = 64 | NFE = 128 | NFE = 256 | NFE = 512 | NFE = 1024 |
|---|---|---|---|---|---|---|---|
| FHS | 7.843 | 7.793 | 7.748 | 7.712 | 7.714 | 7.716 | 7.717 |
| Euler | 7.785 | 7.677 | 7.594 | 7.446 | 7.343 | 7.158 | 6.962 |
| Tweedie $\tau$-leaping | 7.786 | 7.675 | 7.564 | 7.453 | 7.345 | 7.151 | 6.970 |
| $\tau$-leaping | 7.048 | 7.122 | 7.016 | 6.890 | 6.706 | 6.537 | 6.407 |
| Semi-AR | 8.019 | 8.056 | 7.994 | 7.908 | 7.836 | 7.771 | 7.810 |
| $\theta$-RK-2 | 6.772 | 7.017 | 7.085 | 7.010 | 6.831 | 6.682 | 6.548 |
| $\theta$-Trapezoidal | 7.126 | 7.163 | 7.033 | 6.919 | 6.740 | 6.532 | 6.412 |

Table 6: Percentage of positive extrapolated intensities for different algorithms across NFE values.

| Method | NFE = 32 | NFE = 64 | NFE = 128 | NFE = 256 | NFE = 512 | NFE = 1024 |
|---|---|---|---|---|---|---|
| $\theta$-RK-2 | $97.21 \pm 3.1$ | $98.31 \pm 2.0$ | $98.01 \pm 1.3$ | $99.27 \pm 0.9$ | $99.44 \pm 0.7$ | $99.52 \pm 0.6$ |
| $\theta$-Trapezoidal | $95.67 \pm 4.8$ | $97.06 \pm 3.6$ | $98.22 \pm 2.4$ | $98.87 \pm 1.6$ | $99.24 \pm 1.1$ | $99.43 \pm 0.9$ |

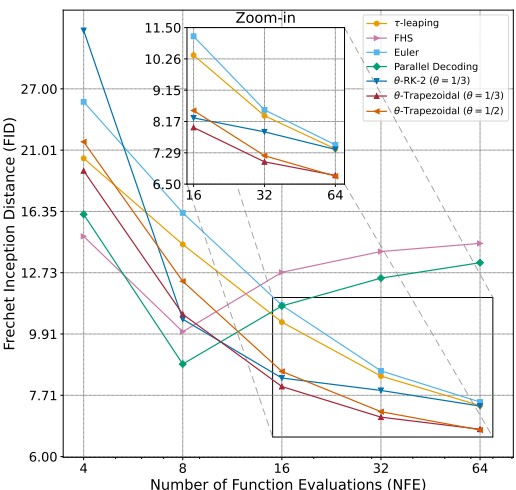

Figure 5: FID of images generated by sampling algorithms vs. number of function evaluations (NFE) with different parameter choices. Lower values are better.

To investigate the robustness of $\theta$-RK-2 with respect to the choice of $\theta$, we also benchmark its performance across multiple choices at NFE 32 and 64, and we present the results in Fig. 6. We observe that the performance of $\theta$-RK-2 has a flat landscape around the optimal $\theta$ choices, which fall in the range $[0.15, 0.4]$. In general, as shown by the curve, the method performs better when extrapolation is used to compute the transition rate matrix, confirming the correctness of our theory (Thm. 5.5) and our discussions. Similar to the behavior of $\theta$-Trapezoidal method in Fig. 4, the performance of $\theta$-RK-2 has a flat landscape around the optimal $\theta$ choices, which typically falls in the range $[0.3, 0.5]$. In general, as shown by the curve, both methods exhibit better performances when extrapolations are deployed, which, once again, certifies the validity of our theoretical results.

Finally, we visualize some images generated with $\theta$-Trapezoidal on 6 different classes in Fig. 7. $\theta$-Trapezoidal consistently generates high-fidelity images that are visually similar to the ground truth ones and well aligned with the concept.

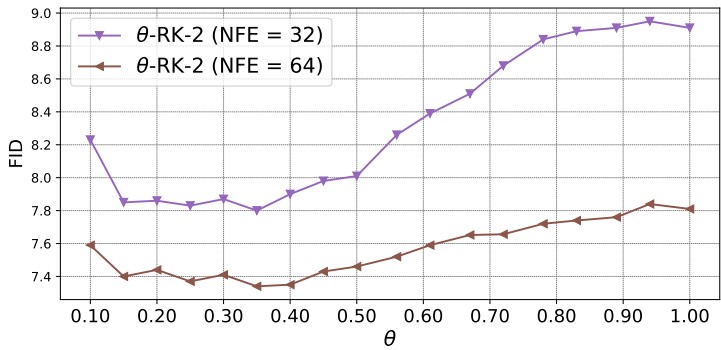

Figure 6: Sampling quality vs. $\theta \in (0, 1]$ in $\theta$-RK-2 algorithm. Sampling quality is quantified through FID.

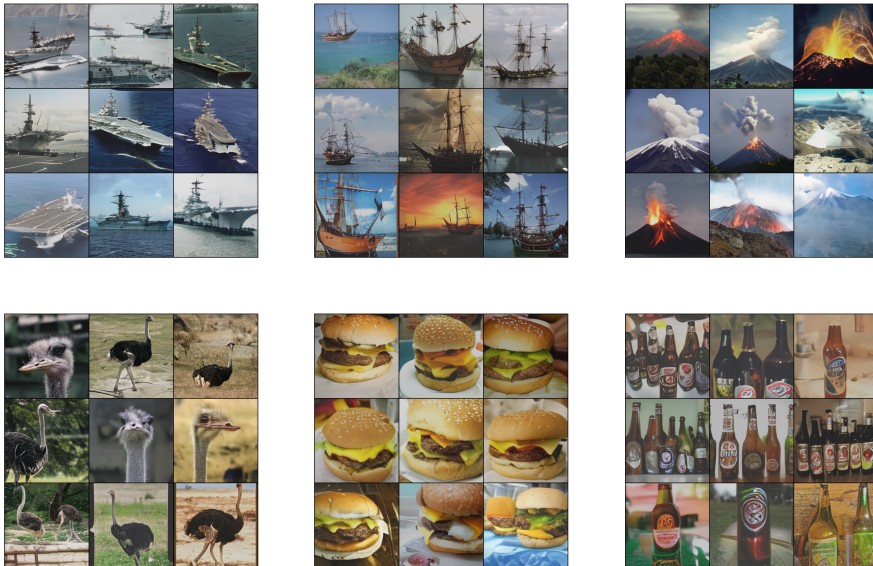

Figure 7: Visualization of samples generated by $\theta$-Trapezoidal. **Upper Left**: Aircraft carrier (ImageNet-1k class: **933**); **Upper Middle**: Pirate (ImageNet-1k class: **724**); **Upper Right**: Volcano (ImageNet-1k class: **980**); **Lower Left**: Ostrich (ImageNet-1k class: **009**); **Lower Middle**: Cheeseburger (ImageNet-1k class: **933**); **Lower Right**: Beer bottle (ImageNet-1k class: **440**).

### D.4 Diffusion Large Language Model and Math Reasoning

For the evaluation of diffusion LLMs on math reasoning datasets, we use the checkpoint of LLaDA [64] after instruction tuning, LLaDA-Instruct 8B[5], one of the state-of-the-art diffusion-based LLMs trained natively based on masked discrete diffusion models [33]. For the math reasoning datasets, we consider GSM8K,[6] a standard benchmark for math reasoning consisting of elementary-school-level word problems. Similar to RADD, since LLaDA is also a masked discrete diffusion model, we can freely choose the noise schedule $\sigma(t)$ used in the inference process. We adopt the same log-linear schedule described in (D.1), and we choose $\epsilon = 10^{-3}$. The computation of the score function using LLaDA follows the same procedure as is depicted in App. D.2.

---

[5]`https://huggingface.co/GSAI-ML/LLaDA-8B-Instruct`
[6]`https://huggingface.co/datasets/openai/gsm8k/viewer/main/train?row=7294`

We choose an early stopping time $\delta = 10^{-3}$, and adopt a uniform discretization of the time interval $(\delta, 1]$ for $\theta$-Trapezoidal, with $\theta = \frac{1}{2}$. For Semi-AR with random remasking or confidence-based remasking strategies, we set the block length to 256, the same as the generation length, to avoid exploiting LLaDA's inherent preference for auto-regressive generation order and ensure a fair comparison with our proposed method. However, we note that our proposed method can also be generalized to a blockwise sampling setting, and we include additional results here for a proof of concept and to demonstrate that LLaDA indeed prefers a left-to-right generation order. We set the sampling temperature to 0.5 for all evaluated inference methods. The evaluation of generated responses follows a standard pipeline with lm-evaluation harness,[7] a standard LLM evaluation kit, following the instruction in the LLaDA repository.[8] Full results are presented in Tab. 7. All experiments are run on 1 NVIDIA A100.

Table 7: Response accuracy on GSM8K with different NFEs.

| Accuracy (%) | NFE = 64 | NFE = 128 | NFE = 256 |
|---|---|---|---|
| Semi-AR (Conf.) | 33.6 | 32.0 | 39.1 |
| Semi-AR (Rand.) | 33.8 | 34.3 | 40.3 |
| $\theta$-Trapezoidal | 35.1 | 38.4 | 39.7 |
| $\theta$-Trapezoidal (Blocksize 8) | 50.4 | 57.4 | 62.5 |

---

[7]https://github.com/EleutherAI/lm-evaluation-harness
[8]https://github.com/ML-GSAI/LLaDA

