# OpenReview forum: "Fast Solvers for Discrete Diffusion Models: Theory and Applications of High-Order Algorithms"
_NeurIPS.cc/2025/Conference — NeurIPS 2025 poster_

### Official Review · Reviewer_5vgu · 2025-07-01

**Clarity:** 3
**Significance:** 2
**Originality:** 2
**Rating:** 4
**Confidence:** 4

**Summary:**

The paper introduces high-order numerical solvers (θ-RK-2 and θ-trapezoidal methods) for discrete diffusion models, aiming to accelerate sampling and inference compared to standard τ-leaping and exact simulation. Theoretical analysis demonstrates second-order convergence in KL divergence under certain conditions. Empirical results on text and image generation show better sample quality for the proposed methods.

**Questions:**

- Can the authors provide evidence that their methods remain effective on larger, more complex benchmarks and with a broader set of evaluation metrics?
- How do the methods perform when the theoretical assumptions are not strictly satisfied? If the authors can address such problem, I am willing to improve my rating.

**Ethical Concerns:**

["NO or VERY MINOR ethics concerns only"]

**Final Justification:**

I have read the rebuttal. The authors gave a very detailed reply that addressed most of my concerns. I raise the rating to borderline accept after the discussion. However, I still suggest the authors conduct evaluation on the GenEval in the future.

**Limitations:**

The main limitations regarding assumptions and empirical scope are discussed by the authors.

**Quality:**

3

**Strengths And Weaknesses:**

Strengths and Weaknesses:

Strengths
- The paper provides a rigorous theoretical analysis, with detailed stochastic integral formulations and error bounds.
- The introduction of high-order schemes to discrete diffusion inference is novel and addresses an under-explored topic.
- Empirical validation is thorough, with experiments on both text and image benchmarks.
- The proposed methods are conceptually simple and easy to incorporate into existing diffusion frameworks.

Weaknesses
- The experimental validation is limited: most results are on toy or small-scale models, with no evaluation on large-scale or widely recognized datasets (e.g., Geneval [1], MMMU [2]). The base models are relatively simple, and only basic baselines are considered.
- Evaluation metrics are primarily FID and NFE, which may not be sufficient to fully support the claimed effectiveness of the methods.
- Theoretical guarantees depend on strong assumptions (regularity, boundedness, positivity, accurate score estimation) that are difficult to ensure in practical scenarios. The robustness of the proposed methods when these conditions are not met is not addressed.

[1] GenEval: An Object-Focused Framework for Evaluating Text-to-Image Alignment

[2] MMMU: A Massive Multi-discipline Multimodal Understanding and Reasoning Benchmark for Expert AGI

---

> ### Author Rebuttal · Authors · 2025-07-31
>
> Thank you for your constructive feedback. We appreciate your recognition of our rigorous theoretical analysis and novelty of introducing high-order schemes to discrete diffusion inference. We address your specific concerns below.
>
> ---
>
> ### Regarding experimental validation on larger benchmarks
>
> We thank the reviewer for this comment. As our work presents a two-fold contribution combining rigorous theoretical analysis with empirical validation, our benchmarks were strategically selected to provide both rigorous theoretical verification and practical performance evaluation: The 15-dimensional toy model allows exact KL divergence to verify theoretical convergence rates, while the GPT-2 text generation and ImageNet image generation represent standard literature benchmarks that enable direct comparison with existing methods.
>
> To enhance the empirical validation of the proposed method, we additionally benchmark its performance on GSM8K (a dataset of grade school math) with LLaDA [4], an 8B paramter masked diffusion language model. For each method, we generate a response of 256 tokens without using demonstration examples in the context, and the answer is evaluated with LM-evaluation-harness, a standard LLM evaluation kit. Due to time limit, we only compare $\theta$-Trapezoidal to the semi-autoregressive sampler therein, considering both confidence-based remasking strategy and purely random remasking strategy. The results are reported below:
>
> | GSM8K Acc                 | NFE = 64  | NFE = 128  | NFE = 256 |
> |:----------------------:|:---------:|:---------:|:--------:|
> | Semi-AR (Confidence)    |33.6    |32.0   |39.1   |
> | Semi-AR  (Random)       |33.8    |34.3   |**40.3**|
> | **$\theta$-Trapezoidal**   |**35.1**|**38.4**|39.7|
>
> $\theta$-Trapezoidal outperforms the Semi-AR sampler in the low NFE regime, where NFE is strictly smaller than the sequence length. At high NFE regime, $\theta$-Trapezoidal exhibits a similarly competitive performance as other solvers. This observation accords with our claim that high-order samplers perform better with lower NFE budgets.
>
> Moreover, the Semi-AR sampler, while performing decently on LLaDA, is a heurstic sampler not universally good-performing. As seen from the results in the reponse below, when apply to RADD (another diffusion LLM), semi-AR significantly underperforms, while our propsoed $\theta$-Trapezoidal demonstrates stable, superior performance across models and NFEs, likely due to its theoretically-grounded effectiveness.
>
> We hope this diminishes your concern on the scalability of our method. Exploring the practical performance on larger-scale text-to-image discrete diffusion models, *e.g.*, GenEval, involves substantial engineering efforts and computational resources beyond this rebuttal period. However, we view scaling to other larger models remains important future work building upon our theoretical foundations.
>
> ### Regarding evaluation metrics
>
> Thank you for pointing this out. To make the reported results more trustworthy, we additionally report the generative perplexity of RADD measured by Llama 3 (8B parameters in contrast to the 700M GPT-2 large), as well as the corresponding unigram entropy. We also include the Semi-AR sampler proposed in the LLaDA paper [4] as an additional benchmark.
>
> Due to time constraints, we only report NFE up to 512, and each metric is computed based on 512 samples. We will include full experiment results in the revision of the paper. For Semi-AR, we follow the orginal implementation proposed in LLaDA and use the random remasking strategy. We found that when choosing confidence-remasking strategy for RADD, the generate samples are consisted of repeated sequences of most frequent tokens, with extremely small entropy, and we thus omit the results in the table.
>
> From the table, we see that our algorithms still performs the best across nearly all NFEs under the new evaluation metric. Other benchmarks such as Semi-AR or FHS either produces texts with higher perplexities or unnatural sample entropy, making them inferior in performance.
>
>
> | Generative ppl with Llama3                 | NFE = 16  | NFE = 32  | NFE = 64 | NFE = 128 | NFE = 256 | NFE = 512 |
> |:----------------------:|:---------:|:---------:|:--------:|:---------:|:---------:|:---------:|
> | FHS                    |342.498    |210.742    |155.258   |132.135    |127.526    |123.013    |
> | Euler                  |318.413    |175.555    |125.955   |91.051     |75.245     |59.971     |
> | Tweedie $\tau$-leaping |316.744    |172.941    |121.248   |94.253     |75.403     |59.943     |
> | $\tau$-leaping         |152.867    |117.930    |86.980    |68.090     |**53.664** |44.676     |
> | **$\theta$-RK-2**          |150.439    |132.090    |107.066   |80.742     |63.277     |52.563     |
> | **$\theta$-Trapezoidal**   |**146.027**|**113.260**|**83.456**|**66.071** |54.307     |**44.293** |
> | Semi-AR                |2696.883   |1684.973   |829.391   |410.177    |251.963    |183.599    |
>
> | Unigram entropy                 | NFE = 16 | NFE = 32 | NFE = 64 | NFE = 128 | NFE = 256 | NFE = 512 |
> |:----------------------:|:--------:|:--------:|:--------:|:---------:|:---------:|:---------:|
> | FHS                    |7.843     |7.793     |7.748     |7.712      |7.714      |7.716      |
> | Euler                  |7.785     |7.677     |7.594     |7.446      |7.343      |7.158      |
> | Tweedie $\tau$-leaping |7.786     |7.675     |7.564     |7.453      |7.345      |7.151      |
> | $\tau$-leaping         |7.048     |7.122     |7.016     |6.890      |6.706      |6.537      |
> | **$\theta$-RK-2**          |6.772     |7.017     |7.085     |7.010      |6.831      |6.682      |
> | **$\theta$-Trapezoidal**  |7.126     |7.163     |7.033     |6.919      |6.740      |6.532      |
> | Semi-AR                |8.019     |8.056     |7.994     |7.908      |7.836      |7.771      |
>
>
> ### Regarding theoretical assumptions and robustness
>
> Thank you for raising these important questions regarding the assumptions underlying our theoretical analysis. We provide detailed responses below addressing the applicability and practical implications of each assumption:
>
> - *Regularity*: Our pursuit of second-order convergence naturally requires second-order continuity of the intensities for rigorous numerical analysis. This smoothness is typically guaranteed under standard forward process choices. However, controlling the Hölder constants involves complex interactions between data properties, time parameterization, and noise schedules—an active research area following empirical studies in the continuous case [1].
> - *Boundedness*: Given our early stopping scheme at time $T-\delta$, boundedness of intensities is ensured under common forward process choices, particularly when the data distribution has a lower bound. This assumption holds in most practical scenarios and can be readily enforced through truncation and appropriate choice of $\delta$.
> - *Positivity*: This assumption is standard in stochastic analysis literature [2], arising from the extrapolation nature of our scheme. The violation probability has been proven asymptotically small to arbitrary order [3]. We explicitly evaluated this probability for discrete diffusion models, with results summarized below:
>
>     | Method | NFE=64 | NFE=128 |  NFE=256 | NFE=512 |
>     |--------|-----------|-----------|-----------|------------|
>     | $\theta$-RK-2 | 98.31 ± 2.0 | 98.01 ± 1.3 | 99.27 ± 0.9 | 99.44 ± 0.7  |
>     | $\theta$-Trapezoidal | 97.06 ± 3.6 | 98.22 ± 2.4 | 98.87 ± 1.6 | 99.24 ± 1.1 |
>
>     These results demonstrate that positivity holds approximately 98-99% of the time. In practical scenarios, violations can be handled through simple cut-off without affecting performance, as confirmed by our empirical results.
>
> - *Score Accuracy*: Like all inference algorithms, our method depends on accurate score estimation. Generated samples contain three error sources: truncation (exponentially small), discretization (which our algorithm minimizes), and estimation (training-dependent). Our observations confirm that more accurate score estimation leads to more effective inference, approaching theoretical second-order convergence.
>
> Our empirical evaluation demonstrates robust performance across different settings, validating the practical effectiveness of our theoretical framework under realistic conditions. We will polish the paper to provide clearer practical guidance for implementation, including truncation strategies for positivity and early stopping for regularity.
>
> ---
>
> We appreciate your detailed feedback and hope our responses address your concerns. Our work establishes rigorous theoretical foundations for high-order methods in discrete diffusion models while demonstrating practical effectiveness through empirical validation. Our theoretical framework advances convergence understanding, while empirical results across multiple domains confirm practical utility. Given these clarifications and additional experiments, we would be grateful if you would reconsider your evaluation.
>
> ---
>
> ### References
>
> [1] Karras, Tero, et al. "Elucidating the design space of diffusion-based generative models." Advances in neural information processing systems 35 (2022): 26565-26577.\
> [2] David F Anderson and Jonathan C Mattingly. A weak trapezoidal method for a class of stochastic differential equations. Communications in Mathematical Sciences, 9(1):301–318, 2011. \
> [3] Hu, Yucheng, Tiejun Li, and Bin Min. "A weak second order tau-leaping method for chemical kinetic systems." The Journal of chemical physics 135.2 (2011).
> [4] Nie, Shen, et al. "Large language diffusion models." arXiv preprint arXiv:2502.09992 (2025).

---

> ### Comment · Reviewer_5vgu · 2025-08-05
> **Response to the Rebuttal.**
>
> I have read the rebuttal. The authors gave a very detailed reply that addressed most of my concerns. I will raise the rating to borderline accept after the discussion. However, I still suggest the authors conduct evaluation on the GenEval in the future.

---

> > ### Author Response · Authors · 2025-08-05
> >
> > Thank you very much for taking the time to review our rebuttal and for your decision to raise the rating to borderline accept. We are delighted to hear that our additional experimental evidence and extended theoretical explanations have successfully addressed most of your concerns.
> >
> > Regarding GenEval, we appreciate your constructive suggestion about evaluation scalability. We would like to assure you that we are actively investigating the necessary engineering integrations and implementations. We will update the paper as soon as we obtain preliminary results along this direction with those we have mentioned in our rebuttal, as we recognize this as important endeavor building upon the theoretical and empirical foundations that this work aimed to lay down.
> >
> > We sincerely thank you for your thoughtful engagement throughout this discussion process. Your feedback has been invaluable in helping us improve our paper and contribute meaningfully to the community.
> >
> > Best regards,\
> > Authors of Paper 13393

---

### Official Review · Reviewer_LqdK · 2025-07-01

**Clarity:** 3
**Significance:** 2
**Originality:** 3
**Rating:** 4
**Confidence:** 4

**Summary:**

This paper addresses the challenge of slow sampling in discrete diffusion models. The authors propose and analyze several high-order sampling algorithms, which are novel extensions of the tau-leaping method. The primary goal is to accelerate the generation process, drawing a parallel to the successful application of high-order solvers for continuous diffusion models.

**Questions:**

1. Regarding Theorem 5.4, the error bound seems to suggest that performance could be further improved by setting the NFE higher than the sequence length (L). Is this a correct interpretation? If so, does it make practical sense to use more sampling steps than the number of tokens to be generated?
2. A critical point of comparison arises at NFE = L. For a simple masking-based sampler like the one in LLaDA, setting NFE = L (i.e., unmasking one token per step) is analogous to autoregressive decoding. This process should, in principle, allow for exact sampling from the model's conditional distributions, resulting in zero sampling error. In this specific and practical regime: What are the theoretical and empirical advantages of using the proposed high-order methods over such a simple, exact-sampling baseline?

**Ethical Concerns:**

["NO or VERY MINOR ethics concerns only"]

**Final Justification:**

I have read the author's response and I am satisfied with the clarifications. I will be keeping my score and recommending acceptance.

**Limitations:**

A key limitation is the lack of comparison to simpler, heuristic-based sampling algorithms (e.g., LLaDA's remasking). The practical benefits of the proposed high-order methods are not clearly established in regimes where these simpler methods might perform optimally (such as when NFE = Sequence Length), which could limit the adoption of this more complex approach.

**Quality:**

3

**Strengths And Weaknesses:**

Strengths:
1. The paper tackles a conceptually important and underexplored problem. While accelerating sampling for continuous diffusion models is a mature field, this work is one of the first to provide a rigorous exploration of high-order methods for the discrete domain.
2. The theoretical analysis is a core strength. The proposed high-order variants of the tau-leaping method are well-motivated, and their derivation and convergence analysis are presented with mathematical rigor.
3. The experiments are comprehensive and sufficiently validate the efficacy of the proposed algorithms. The results clearly demonstrate an acceleration in the sampling process when compared to baseline first-order methods across several tasks.
Weakness:
I think the paper misses a direct comparison to conceptually simpler methods like the confidence-ranked remasking technique proposed in LLaDA (Arxiv: 2502.09992). Such methods, while perhaps less theoretically grounded in diffusion theory, are highly practical and serve as a crucial baseline. A thorough analysis, particularly in the important inference scenario where the Number of Function Evaluations (NFE) equals the sequence length, is needed to demonstrate the practical advantages of this work's more complex approach over the strong, simple heuristics in LLaDA. See also my question below.

---

> ### Author Rebuttal · Authors · 2025-07-31
>
> Thank you for your thorough and insightful review of our paper. We greatly appreciate your recognition of our theoretical contributions and the importance of addressing the underexplored problem of accelerating discrete diffusion model sampling. Your constructive feedback, particularly regarding comparisons with simpler methods and the practical regime where NFE equals sequence length, raises important points that will help strengthen our work.
>
> ---
>
> ### Regarding comparison with confidence-ranked remasking methods (*e.g.*, LLaDA)
>
> Thank you for highlighting this important baseline. We want to clarify that we have indeed included such a comparison in our experiments through the "parallel decoding" baseline, which is conceptually equivalent to the confidence-ranked remasking technique proposed in LLaDA. Specifically, our parallel decoding implementation follows the same principle: at each step, we compute the logits for all masked tokens and then remask the tokens with the lowest confidence scores (see lines 1461-1465 in our paper).
>
> To further resolve this issue, we have carried out additional experiments with LLaDA. We refer the reviewer to our response to Reviewer 5vgu for more results on confidence-ranked methods on RADD and LLaDA. The additional results confirm again the superior performance of our proposed algorithms. These experiments reflects that our proposed high-order solver consistently exhibits competitive performance across tasks of various type and models of various size, while other sampler heuristics such as semi-AR sampler might suffer from instability and have failure cases (such as on RADD).
>
> We will incorporate these new experiments into our revision and add a clarification in the paper on these samplers. We will also expand our discussion to better contextualize how our high-order methods compare to these confidence-based heuristics in different sampling regimes.
>
> ### Regarding NFE higher than sequence length (Theorem 5.4)
>
> Thank you for this astute observation about our theoretical results. You are correct that Theorem 5.4 analyzes uniform discrete diffusion models where using more steps can indeed reduce discretization error. However, it's important to distinguish between two types of discrete diffusion models:
> - For *uniform* discrete diffusion models, the forward process converges to a uniform distribution over the entire state space, and multiple transitions per position are possible. In this setting, using $\mathrm{NFE} > L$ can provide benefits by allowing finer time discretization.
> - For *masked* discrete diffusion models (like those used in our text and image experiments), each position typically transitions only once from masked to unmasked. Therefore, as you correctly note, using $\mathrm{NFE} > L$ provides no additional benefit. Our experiments on text and image generation focus on the practical regime where $\mathrm{NFE} \leq L$, which is why we report results up to $\mathrm{NFE} = 1024$ for sequence length 1024.
>
>
> We will clarify this distinction more prominently in the revision to avoid confusion.
>
> ### Regarding advantages over exact sampling when $\text{NFE} = L$
>
> This is an excellent question that goes to the heart of our contribution. While it's true that methods like LLaDA can perform exact sampling when $\text{NFE} = L$ (unmasking one token per step), there are several important considerations that favor our high-order approach:
>
> 1. The "exact sampling" property assumes perfect score function estimation. In practice, neural networks provide imperfect score estimates, and our high-order methods can better handle these estimation errors through their numerical properties. As discussed in Section 3.1, exact simulation methods suffer from highly skewed computational effort, with most jumps concentrated near the end of the generation process when score estimation errors are highest.
> 2. Our methods provide flexibility in the NFE budget. While $\text{NFE} = L$ might seem natural, practitioners often want to trade off quality for speed ($\text{NFE} < L$) or achieve higher quality with more computation ($\text{NFE} \rightarrow L$). Our theoretical analysis shows that high-order methods achieve better quality-speed trade-offs across this entire spectrum, not just at the specific point $\text{NFE} = L$.
> 3. Our empirical results demonstrate practical advantages even when comparing against confidence-based methods. In Figures 4 and 7, the $\theta$-trapezoidal method consistently outperforms parallel decoding (equivalent to LLaDA-style remasking) across various NFE budgets, including those approaching the sequence length.
>
> We will expand our discussion in Section 3.1 to better articulate these advantages and add a more detailed comparison specifically at the $\text{NFE} = L$ regime in our revision.
>
> ---
>
> Thank you again for your valuable feedback and thought-provoking questions. Your comments have helped us identify areas where we can clarify our contributions and better position our work relative to existing methods. We believe that addressing these points will significantly strengthen our paper and make it more useful to practitioners in the field. We hope our responses have addressed your concerns.

---

> > ### Comment · Reviewer_LqdK · 2025-08-04
> > **post-rebuttal response**
> >
> > I have read the author's response and I am satisfied with the clarifications. I will be keeping my score and recommending acceptance.

---

> ### Author Response · Authors · 2025-08-05
>
> Thank you again for your thorough and insightful review and your recommendation for acceptance. We are pleased to hear that our rebuttal has successfully clarified your questions. Your constructive feedback and thoughtful insights have helped us better articulate the advantages of our high-order methods, and we appreciate your recognition of this work's contribution to the under-explored area of discrete diffusion sampling acceleration.
>
> Best regards,\
> Authors of Paper 13393

---

### Official Review · Reviewer_HxA8 · 2025-07-04

**Clarity:** 4
**Significance:** 3
**Originality:** 3
**Rating:** 5
**Confidence:** 3

**Summary:**

This paper introduces higher order methods for discrete diffusion models, enabling time stepping schemes with higher order convergence when compared to standard schemes. Explicit examples of higher order methods, such as RK-2 and Trapezoidal are given. Computational examples are provided.

**Questions:**

Can the techniques presented here be used to create even higher order methods, such as 3rd or fourth order methods?

Can the authors comment on the influence that the noise schedule has on their method? Usually the constant on the order of a high order scheme has some dependence on the gradient of the function to be integrated. By choice of a clever schedule, would one be able to also help diminish the size of this constant.

Do classical time stepping stability analysis apply? For standard Runge-Kutta methods, we can look at regions of stability, can we do anything similar here?

**Ethical Concerns:**

["NO or VERY MINOR ethics concerns only"]

**Limitations:**

There is no limitations section of this paper, I would recommend that the authors add this to their submission.

**Quality:**

4

**Strengths And Weaknesses:**

Strengths:
This paper is well motivated, it is true in the continuous setting that higher order methods have traditionally given a computational gain. As the discrete time diffusion is still continuous in its temporal component, this is a natural extension that should be explored.

The paper is well layed out and easy to follow. The paper starts with defining discrete diffusion models, shows current stepping methods and presents Theorem 3.1 revealing standard methods are first order. The authors then present their new schemes and Theorem 5.4 and 5.5, natural extensions of Theorem 3.1.

The paper well states its assumptions, with Assumptions 5.1,5.2 and 5.3.  In particular, it is nice that Assumption 5.2 is directly stated, showing that for their proposed higher order scheme to work, they need 2nd order regularity in the rate matrix.

Weaknesses:
I do not see any major weaknesses. It would be nice if some standard techniques from higher order schemes were also presented, such as Buchner tables for the construction of these methods if applicable.

---

> ### Author Rebuttal · Authors · 2025-07-31
>
> Thank you for your thorough and constructive review of our paper. We greatly appreciate your positive assessment of our work as well-motivated and well-laid out. We are particularly pleased that you found our theoretical contributions, including the second-order convergence results and the explicit statement of our assumptions, to be clearly presented. Your insightful questions and suggestions will help us improve the paper further.
>
> ---
>
> ### Regarding standard techniques from higher-order schemes
>
> Thank you for this valuable suggestion. We agree that including discussion of standard techniques such as Butcher tableaux would enhance the paper's completeness. The reviewer's suggestion coincides with a classical work [1] that uses Butcher tableaux to derive various Poisson Runge-Kutta methods. In our revision, we will add a brief discussion in Section 4 explaining how these techniques relate to our methods and how certain modifications can be carried out for the discrete diffusion setting.
>
> ### Regarding higher-order methods ($\geq$ 3rd order)
>
> Thank you for this insightful question. Indeed, constructing even higher-order methods is possible. As shown in the work [2] on numerical analysis of inhomogeneous continuous-time Markov chains, higher-order schemes can be developed. However, there are several considerations, including the smoothness requirement, score estimation accuracy, and computational trade-off, that may require case-by-case empirical studies. We will add a discussion of these points in Section 7 (Future Work) of our revision, noting this as an exciting direction for future research.
>
> ### Regarding the influence of noise schedule
>
> Thank you for this excellent observation. You are absolutely correct that the constant in our convergence bounds depends on the noise schedule and time parametrization. This dependency enters through the smoothness properties of the intensity functions (Assumption 5.2) and affects the constants hidden in our $\mathcal{O}(\kappa^2 T)$ bounds.
>
> As noted in the continuous diffusion literature, particularly the work [3], the choice of noise schedule significantly impacts both training and inference. For discrete diffusion models, this influence is even more pronounced due to the discrete state space. The optimal noise schedule likely depends on the properties of the data distribution and the specific discrete state space structure.
>
> We will add a remark after Theorem 5.4 discussing how the noise schedule affects the practical constants in our convergence bounds and reference relevant empirical studies. This is indeed an active area of research that deserves further investigation.
>
> ### Regarding classical time-stepping stability analysis
>
> Thank you for raising this important theoretical question. To the best of our knowledge, the classical notion of stability regions from deterministic numerical analysis does not directly translate to our stochastic setting. Instead of absolute stability regions, the relevant concepts for stochastic numerical schemes include weak convergence and moment bounds, and our analysis follows the framework established in the stochastic numerical analysis literature, such as the work [3] on tau-leaping methods. We will add a remark in Section 5.2 clarifying this distinction.
>
> ### Regarding the limitations section
>
> Thank you for pointing out this omission. We will add a dedicated Limitations section before Conclusion, discussing our theoretical assumptions, memory requirements for our proposed inference schemes, the dependence of practical performance on score accuracy, and current restriction to second-order methods, *etc..*
>
> ---
>
> We sincerely thank you again for your careful review and valuable feedback. Your questions have helped us identify several areas where we can clarify and strengthen our presentation. We will incorporate all the suggested improvements in our revision and believe these additions will make our paper more complete and accessible to readers.
>
> ---
> ### References
>
> [1] Burrage, Kevin, and Tianhai Tian. "Poisson Runge-Kutta methods for chemical reaction systems." (2004): 82-96.\
> [2] M. Arns, P. Buchholz, A. Panchenko, (2009) On the Numerical Analysis of Inhomogeneous Continuous-Time Markov Chains. INFORMS Journal on Computing 22(3):416-432.\
> [3] Karras, Tero, et al. "Elucidating the design space of diffusion-based generative models." Advances in neural information processing systems 35 (2022): 26565-26577.\
> [4] Hu, Yucheng, Tiejun Li, and Bin Min. "A weak second order tau-leaping method for chemical kinetic systems." The Journal of chemical physics 135.2 (2011).

---

> > ### Comment · Reviewer_HxA8 · 2025-08-06
> >
> > I would like to the thank the authors for their detailed response to my questions. I am very satisfied with the response and will keep my rating as an accept. I am pleased that the authors are including a remark on the influence of the noise schedule.

---

> ### Author Response · Authors · 2025-08-06
> **Response to the Comment**
>
> We sincerely thank the reviewer again for your time and thoughtful review, as well as your recommendation for acceptance. We are glad that our rebuttal addressed your concerns effectively. In case you have any other question, please don't hesitate to let us know.
>
> Best regards,
>
> Authors of Paper 13393

---

### Official Review · Reviewer_61BM · 2025-07-06

**Clarity:** 3
**Significance:** 3
**Originality:** 3
**Rating:** 3
**Confidence:** 4

**Summary:**

This work addresses  computational bottleneck in discrete diffusion models by developing higher-order solvers that significantly reduce the number of function evaluations (NFE) required for sampling. The authors introduce theoretical foundations for higher-order methods in the discrete setting and demonstrate substantial improvements in sampling efficiency for text generation tasks.

**Questions:**

Following Weakness

It would be informative for readers and broader community to see integration of higher order solvers with
- Effect of Temperature/Confidence
- Multi-token unmasking heuristics like in FastDLLMs, etc.
- ReMasking
- Predictor-Corrector Solvers
- How well higher order solvers work with distilled discrete diffusion models based on e.g., SDTT Checkpoints

Also additional metrics like use state-of-the-art language models (LLaMA-3 or alternatives) to evaluate generated text quality independently of the diffusion model.

**References**

1. G. Wang et al. Remasking Discrete Diffusion Models with Inference-Time Scaling (https://arxiv.org/abs/2503.00307)
2. J. Deschenaux et al. Beyond Autoregression: Fast LLMs via Self-Distillation Through Time (https://arxiv.org/html/2410.21035v1)
3. C. Wu et al. Fast-dLLM: Training-free Acceleration of Diffusion LLM by Enabling KV Cache and Parallel Decoding (https://arxiv.org/html/2505.22618v1)

**Ethical Concerns:**

["NO or VERY MINOR ethics concerns only"]

**Final Justification:**

I retain my original rating, while agree on the contribution of formulation of higher order solvers for discrete diffusion.

Higher order integration/ODE solvers is well established literature with close relation to SGD variants too.

While the work is valuable the contribution would be significantly enhanced and useful if we can understand implications of higher order solvers in discrete diffusion setting which authors do not address in anyway and largely adopt Numerical Integration convergence guarantees when applied to discrete diffusion setting

**Quality:**

2

**Strengths And Weaknesses:**

**Strengths**

* The paper provides mathematical foundations for higher-order solvers in discrete diffusion models, extending beyond the first-order methods commonly used in practice.
* The paper demonstrates meaningful reductions in NFE while maintaining or improving generation quality, addressing a key limitation of discrete diffusion models.
* The paper is well-written, clearly organized, and accessible to readers.


**Weakness**
Limited Scope of Experimental Evaluation

The paper lacks comprehensive analysis of how different solver configurations perform under varying conditions, limiting understanding of when and why higher-order methods excel. This limits drawing insights and adoption guidelines for readers and broader community

- Currently this work does not compare against several important recent methods and how effective higher order solvers are across these settings.
- While the appendix mentions predictor-corrector solvers, the authors neither evaluate them experimentally nor discuss how higher-order methods could be integrated within predictor-corrector frameworks.
- Remasking during sampling shows enhanced generation quality. Would be interesting to see higher order solvers with Remasking like in works like ReMDM
- Absence of semantic quality metrics: No evaluation using ModernBERT-based MAUVE scores or similar metrics that assess semantic coherence and diversity (entropy)
- Minimal Performance gain in case of Image Generation (but didn't consider this much for rating)

---

> ### Author Rebuttal · Authors · 2025-07-31
>
> Thank you for your constructive review and recognition of our mathematical foundations, meaningful NFE reductions while maintaining generation quality, and clear presentation. We're particularly encouraged by your acknowledgment that our work addresses key limitations in discrete diffusion models and provides accessible theoretical insights.
>
> Beyond empirical results, our work establishes the first rigorous mathematical framework for high-order numerical methods in discrete diffusion inference, achieving provably second-order convergence guarantees that were previously unknown in this domain. Our $\theta$-trapezoidal method maintains unconditional second-order accuracy across all $\theta \in (0,1]$, representing a fundamental theoretical advance over existing first-order $\tau$-leaping methods. These theoretical contributions are validated through empirical evaluation across multiple domains, demonstrating that rigorous mathematical foundations translate directly into practical performance gains and open new research directions for discrete diffusion acceleration.
>
> ---
>
> ### Regarding the evaluation scope and integration with recent methods
>
> Thank you for raising this important point about expanding experimental scope. We acknowledge that more comprehensive analysis across different solver configurations and integration with recent techniques would strengthen our work's practical impact. Our current work establishes fundamental theoretical frameworks for high-order methods in discrete diffusion models, with experiments strategically designed to validate theoretical predictions across multiple domains.
>
> Regarding the specific integration suggestions:
> - *Predictor-corrector frameworks*: Our high-order methods can naturally serve as enhanced predictor steps, with $\theta$-trapezoidal's extrapolation nature particularly well-suited for this integration. The theoretical benefits are also interesting directions, similar to continuous counterparts [1].
> - *Remasking techniques*: ReMDM's remasking technique is intended as inference-time scaling for generation quality enhancement, where computation budget such as NFE is not a major concern. Our high-order methods serve a completely different purpose: reducing inference NFE while maintaining generation quality. Since ReMDM is concurrent with our submission, we'll investigate this integration in future work.
> - *Temperature and confidence*: Temperature is orthogonal to our main algorithm design and typically depends heavily on specific models and targeted tasks. For example, LLaDA [2] performs well under low sampling temperature, while DiffuCoder [3] works better under high temperature. We select best-performing temperatures for each model rather than ablating this choice. Confidence is also tangent to our approach as we don't adopt heuristics for determining token generation order.
> - *Multi-token unmasking*: Regarding advanced acceleration techniques such as those in fast-dLLM, incorporating KV-cache could further improve our algorithm's inference FLOPS. However, our focus is restricted to algorithmic level with efficiency quantified by number of function evaluations (NFEs), making fast-dLLM's techniques beyond our current scope. Since fast-dLLM is concurrent with our submission, we'll investigate this integration in future work.
> - *Distilliation*: Our high-order numerical schemes operate at inference level and should be compatible with various architectures and training approaches, including distilled models, as long as the distilled models satisfy our analytical assumptions.
>
> We appreciate these diverse suggestions and agree they demonstrate intriguing directions for future exploration. We will include discussion of these relevant references and potential integration approaches in our revision to provide a more comprehensive view of the research landscape.
>
>
> ### Regarding semantic quality metrics and evaluation depth
>
> Thank you for pointing out this important issue. We adopted your advice and use Llama 3 for generative complexity computation, also reporting unigram entropy as a diversity metric. We re-benchmarked RADD performance under this evaluation framework and included LLaDA's semi-AR sampler as an additional benchmark.
>
> Due to time constraints, we report NFE up to 512 with each metric computed based on 512 samples. We will include full experimental results in the paper revision. For the semi-autoregressive (Semi-AR) method, we follow the original implementation proposed in LLaDA using the random remasking strategy. We found that when choosing confidence-remasking strategy for RADD, the generated samples consist of repeated sequences of most frequent tokens with extremely small entropy, so we omit those results.
>
> Our proposed algorithms perform best across nearly all NFEs under the new evaluation metrics. Other benchmarks such as Semi-AR or FHS either produce texts with higher perplexities or unnatural sample entropy, making them inferior in performance compared to $\theta$-Trapezoidal. This again suggests the empirical advantage of high-order solvers in practice.
>
>
> | Generative ppl with Llama3                 | NFE = 16  | NFE = 32  | NFE = 64 | NFE = 128 | NFE = 256 | NFE = 512 |
> |:----------------------:|:---------:|:---------:|:--------:|:---------:|:---------:|:---------:|
> | FHS                    |342.498    |210.742    |155.258   |132.135    |127.526    |123.013    |
> | Euler                  |318.413    |175.555    |125.955   |91.051     |75.245     |59.971     |
> | Tweedie $\tau$-leaping |316.744    |172.941    |121.248   |94.253     |75.403     |59.943     |
> | $\tau$-leaping         |152.867    |117.930    |86.980    |68.090     |**53.664** |44.676     |
> | **$\theta$-RK-2**          |150.439    |132.090    |107.066   |80.742     |63.277     |52.563     |
> | **$\theta$-Trapezoidal**   |**146.027**|**113.260**|**83.456**|**66.071** |54.307     |**44.293** |
> | Semi-AR                |2696.883   |1684.973   |829.391   |410.177    |251.963    |183.599    |
>
> | Unigram entropy                 | NFE = 16 | NFE = 32 | NFE = 64 | NFE = 128 | NFE = 256 | NFE = 512 |
> |:----------------------:|:--------:|:--------:|:--------:|:---------:|:---------:|:---------:|
> | FHS                    |7.843     |7.793     |7.748     |7.712      |7.714      |7.716      |
> | Euler                  |7.785     |7.677     |7.594     |7.446      |7.343      |7.158      |
> | Tweedie $\tau$-leaping |7.786     |7.675     |7.564     |7.453      |7.345      |7.151      |
> | $\tau$-leaping         |7.048     |7.122     |7.016     |6.890      |6.706      |6.537      |
> | **$\theta$-RK-2**          |6.772     |7.017     |7.085     |7.010      |6.831      |6.682      |
> | **$\theta$-Trapezoidal**   |7.126     |7.163     |7.033     |6.919      |6.740      |6.532      |
> | Semi-AR                |8.019     |8.056     |7.994     |7.908      |7.836      |7.771      |
>
> Additionally, we benchmarked our high-order solver on GSM8K with LLaDA to enhance empirical validation. Please refer to our response to Reviewer 5vgu for detailed results. We hope these additional experiments could alleviate your concerns about evaluation scope.
>
> ### Regarding performance gains in image generation
>
> The smaller gains in image generation compared to text reflect different domain characteristics. However, $\theta$-trapezoidal consistently outperforms existing approaches across nearly all tested NFE budgets. Our theoretical analysis suggests high-order methods' benefits become more pronounced with accurate score estimation and longer time horizons, potentially explaining domain-specific improvement variations.
>
> ---
>
> We sincerely appreciate your detailed feedback and constructive suggestions. Our work provides a two-fold contribution to the field: establishing rigorous theoretical foundations for high-order methods in discrete diffusion models and demonstrating their practical effectiveness across multiple domains. The theoretical framework we establish, combined with empirical validation using state-of-the-art evaluation models, opens new avenues for efficient discrete diffusion inference that complement existing acceleration approaches.
>
> The enhancements you suggest would indeed strengthen the practical applicability of our work, and we're committed to incorporating these improvements in future iterations. Given the novelty of our theoretical contributions, the consistent empirical improvements we demonstrate across different evaluation frameworks, and our willingness to address the evaluation scope concerns, we would be grateful if you would reconsider your evaluation of our work.
>
> ---
> ### References
>
> [1] Chen, Sitan, et al. "The probability flow ode is provably fast." Advances in Neural Information Processing Systems 36 (2023): 68552-68575.\
> [2] Nie, Shen, et al. Large Language Diffusion Models. arXiv:2502.09992.\
> [3] Gong, Shansan, et al. DiffuCoder: Understanding and Improving Masked Diffusion Models for Code Generation. arXiv:2506.20639

---

> > ### Comment · Reviewer_61BM · 2025-08-05
> > **Discussion**
> >
> > I appreciate the authors' thoughtful response and the additional benchmarking using a more robust model like LLaMA3, which helps validate the performance of the proposed approach.
> >
> > While I agree with the authors that a comprehensive evaluation of practical aspects is beyond the scope of the rebuttal, I believe it remains important to better understand the limitations and generality of the proposed solvers, especially in the context of the discrete diffusion framework and its inherent biases.
> >
> > The formulation of higher-order solvers for discrete diffusion is certainly valuable. However, while techniques such as re-masking and multi-token decoding may appear orthogonal to the solvers themselves, they do influence the overall sampling dynamics. Specifically, although we sample from the same marginal distribution at a given timestep or noise level, we deviate from the very low-rate regime assumed in discrete flow matching once we begin unmasking more tokens. This, in turn, can affect how well higher-order solvers perform in practice.
> >
> > I remain open to increasing my rating, but I would strongly encourage the authors to explore additional experimental setups that shed light on the limitations and generalization of higher-order solvers in discrete diffusion settings. Such insights would strengthen the overall contribution and practical relevance of the work.

---

> ### Author Response · Authors · 2025-08-06
> **Response to the Reply**
>
> Thank you for your thoughtful and technically insightful response. We deeply appreciate your recognition that our introduction of higher-order schemes to discrete diffusion models is "*certainly valuable*" and your constructive guidance on strengthening the practical aspects of our work.
>
> Before diving into discussion, we would like to further emphasize the **theoretical contributions of our work**, which is **a core component of the paper**. Our work establishes the *first* mathematical framework for high-order numerical methods in discrete diffusion model inference, which achieves second-order convergence guarantees. As far as we are aware, prior approximate methods like $\tau$-leaping were limited to be of first-order. We believe this theoretical foundation, combined with empirical validation across multiple domains, **opens new avenues for more efficient discrete diffusion model inference**.
>
> Regarding your concern about deviating from the low-rate regime when unmasking more tokens, we would like to share our understanding. To the best of our knowledge, when the rate is high and more tokens are unmasked at each step, the discretization error between consecutive timesteps may actually become larger due to more aggressive transitions. In such scenarios, we believe our higher-order schemes could be particularly effective compared to first-order methods, mainly because they introduce intermediate state approximations that better capture the underlying dynamics during these larger discrete jumps. If our understanding is incorrect, we would be grateful for your further clarification on how high unmasking rates specifically affect the performance of higher-order solvers.
>
> We fully acknowledge that a comprehensive and large-scale experimental validation of the proposed high-order solvers would greatly enhance the practical impact of our work. While we recognize that understanding how our solvers behave under different sampling regimes is crucial for practical adoption, we also would like to respectfully point out that re-masking and multi-token decoding techniques are public after March 1st, 2025, and therefore *concurrent* with our submission as per NeurIPS 2025 Official FAQ. Beyond the additional benchmarking with LLaMA-3 and enhanced evaluation metrics we have already conducted, we will be committed to enhancing our paper with a thorough analysis addressing the limitations and practical scope of high-order solvers in the setting of discrete diffusion models.
>
> Given the novel theoretical framework we have established, *i.e.*, **introducing rigorous second-order convergence guarantees to discrete diffusion models for the first time**, along with our commitment to addressing the practical questions raised by multiple reviewers, we would be deeply grateful if you would possibly re-consider your evaluation by taking both the fundamental theoretical contributions and the extra experimental results listed in our rebuttals into account. We will carefully incorporate suggestions from all reviewers in our revised manuscript. Thank you again for your insightful review and your dedicated contribution to the discrete diffusion community.
>
> Best regards,\
> Authors of Paper 13393

---

### Note · Authors · 2025-08-12

Dear Area Chair and Reviewers,

We would like to express our sincere gratitude for your time and effort in reviewing our work. Your insightful feedback has been invaluable in strengthening our paper.

We are confident that our work offers a significant two-fold contribution to the field of discrete diffusion models. Theoretically, we rigorously establish the first high-order convergence guarantee in KL divergence for discrete diffusion model inference. Empirically, we validate our theoretical claims across multiple domains, including image and text generation, demonstrating the practical effectiveness of our proposed methods in real applications. We believe these contributions lay a strong foundation for future research in accelerating discrete diffusion models.

We have taken the reviewers' primary concerns regarding the experimental scope and evaluation metrics very seriously. In response, we have conducted new experiments on the 8B-parameter LLaDA model and evaluated our methods using LLaMA-3-based evaluation and unigram entropy. The preliminary results, which we have shared in our rebuttal, further support our claims. We are committed to carrying out a more extensive empirical investigation on other datasets, models, and evaluation metrics, and we will ensure these results are incorporated into the final version of our paper. We hope these additions and our commitment to further validation address the reviewers' remaining concerns.

Thank you once again for your constructive engagement. We believe our paper presents substantial methodological and theoretical advances, supported by strong empirical evidence, and we are eager to share our findings with the NeurIPS community.

Sincerely,\
The Authors of Paper 13393

---

### Decision · Program_Chairs · 2025-09-17

**Decision:**

Accept (poster)

**Comment:**

The work proposes a higher-order numerical inference scheme for discrete diffusion models. The paper provides theoretical justifications for their proposed scheme and provides empirical tests of their method. The main concern from two of the reviewers is the limited scope of the experimental evaluations. Specifically, Reviewer 61BM wrote that the work would be improved by testing how the higher-order solver interacts with the following:
> - Effect of Temperature/Confidence
> - Multi-token unmasking heuristics like in FastDLLMs, etc.
> - ReMasking
> - Predictor-Corrector Solvers
> - How well higher order solvers work with distilled discrete diffusion models based on e.g., SDTT Checkpoints

In practice, it is important to test the effects of these interactions. However, I believe the theoretical contributions are strong enough to warrant acceptance without additional empirical validation.